# The success of artificial selection for collective composition hinges on initial and target values

Juhee Lee[1,2†], Wenying Shou[3]*, Hye Jin Park[1,2]*

[1]Department of Physics, Inha University, Incheon, Republic of Korea; [2]Asia Pacific Center for Theoretical Physics, Pohang, Republic of Korea; [3]Centre for Life's Origins and Evolution, Department of Genetics, Evolution and Environment, University College London, London, United Kingdom

## eLife Assessment

This **important** study of artificial selection in microbial communities shows that the possibility of selecting a desired fraction of slow and fast-growing types is impacted by their initial fractions. The evidence, which relies on mathematical analysis and simulations of a stochastic model, is **compelling**. It highlights the tension between selection at the strain and the community level. This study should be of interest to researchers interested in ecology, both theoretical and experimental.

*For correspondence:
wenying.shou@gmail.com (WS);
hyejin.park@inha.ac.kr (HJP)

Present address: †Integrated Science Lab, Department of Physics, Umeå university, Umeå, Sweden

**Abstract** Microbial collectives can perform functions beyond the capability of individual members. Enhancing collective functions through artificial selection is, however, challenging. Here, we explore the 'rafting-a-waterfall' metaphor where achieving a target population composition depends on both target and initial compositions. Specifically, collectives comprising fast-growing (F) and slow-growing (S) individuals were grown for 'maturation' time, and the collective with S-frequency closest to the target value is chosen to 'reproduce' via inoculating offspring collectives. During collective maturation, intra-collective selection acts like a waterfall, relentlessly driving the S-frequency to lower values, while during collective reproduction, inter-collective selection resembles a rafter striving to reach the target frequency. Using simulations and analytical calculations, we show that intermediate target S frequencies are the most challenging, akin to a target within the vertical drop of a waterfall, rather than above or below it. This arises because intra-collective selection is the strongest at intermediate S-frequencies, which can overpower inter-collective selection. While achieving a low target S frequencies is consistently feasible, attaining high target S-frequencies requires an initially high S-frequency — much like a raft that can descend but not ascend a waterfall. As Newborn size increases, the region of achievable target frequency is reduced until no frequency is achievable. In contrast, the number of collectives under selection plays a less critical role. In scenarios involving more than two populations, the evolutionary trajectory must navigate entirely away from the metaphorical 'waterfall drop.' Our findings illustrate that the strength of intra-collective evolution is frequency-dependent, with implications in experimental planning.

## Introduction

Microbial collectives can carry out functions that arise from interactions among member species. These functions, such as waste degradation (*Woo et al., 2020*; *Sun et al., 2022*), probiotics (*Bober et al., 2018*), and vitamin production (*Wang et al., 2016*), can be useful for human health and biotechnology. To improve collective functions, one can perform artificial selection (directed evolution) on

collectives (see *Figure 1*): Low-density 'Newborn' collectives are allowed to 'mature' during which cells proliferate and possibly mutate, and community function develops. 'Adult' collectives with high functions are then chosen to reproduce, each seeding multiple offspring Newborns. Artificial selection of collectives have been attempted both in experiments (*Goodnight, 1990*; *Swenson et al., 2000b*; *Swenson et al., 2000a*; *Blouin et al., 2015*; *Panke-Buisse et al., 2015*; *Panke-Buisse et al., 2017*; *Jochum et al., 2019*; *Wright et al., 2019*; *Raynaud et al., 2019*; *Arora et al., 2020*; *Chang et al., 2020*; *Mueller et al., 2021*; *Jacquiod et al., 2022*; *Raynaud et al., 2022*; *Arias-Sánchez et al., 2024*) and in simulations (*Penn, 2003*; *Penn and Harvey, 2004*; *Williams and Lenton, 2007*; *Xie et al., 2019*; *Doulcier et al., 2020*; *Xie and Shou, 2021*; *Chang et al., 2021*; *Fraboul et al., 2023*; *Lalejini et al., 2022*; *Zaccaria et al., 2023*; *Vessman et al., 2023*), often with unimpressive outcomes.

One of the major challenges in selecting collectives is to ensure the inheritance of a collective function (*Xie et al., 2023*; *Thomas et al., 2024*). Inheritance from a parent collective to offspring collectives can be compromised by changes in genotype and species compositions. During maturation of a collective, genotype compositions within each species can change due to intra-collective selection favoring fast-growing individuals (*Figure 1*, 'intra-collective' selection), while species compositions can change due to ecological interactions. Furthermore, during the reproduction of a collective, genotype and species compositions of offspring can vary stochastically from those of the parent (*Figure 1*, 'genetic drift').

Here, we consider the selection of collectives comprising two or three populations with different growth rates, and our goal is to achieve a target composition in the Adult collective. This is a common quest: whenever a collective function depends on both populations, the collective function is maximized, by definition, at an intermediate frequency (e.g. too little of either population will hamper function; *Xie et al., 2019*). Earlier work has demonstrated that nearly any target species composition can be achieved when selecting communities of two competing species with unequal growth rates (*Doulcier et al., 2020*; *Rainey, 2023*), so long as the shared resource is depleted during collective maturation (*Doulcier et al., 2020*). In this case, initially, both species evolved to grow faster, and the slower-growing species was preserved due to stochastic fluctuations in species composition during collective reproduction. Eventually, both species evolved to grow sufficiently fast to deplete the shared resource during collective maturation, and evolution in competition coefficients then acted to stabilize the species ratio to the target value (*Doulcier et al., 2020*). Regardless, earlier studies are often limited to numerical explorations, with prohibitive costs for a full characterization of the parameter space for such nested populations (population of collectives, and populations of variants within a collective).

We mathematically examine the selection of composition in collectives consisting of populations growing at different rates. We made simplifying assumptions so that we can analytically examine the evolutionary tipping point between intra-collective and inter-collective selection. We show that this tipping point creates a 'waterfall' effect which restricts not only which target compositions are achievable, but also the initial composition required to achieve the target. We also investigate how the range of achievable target composition is affected by the total population size in Newborns and the total number of collectives under selection. Finally, we show that the waterfall phenomenon extends to systems with more than 2 populations.

## Results and discussion

To enable the derivation of an analytical expression, we have made the following simplifying assumptions. First, growth is always exponential, without complications such as resource limitation, ecological interactions between the two populations, or density-dependent growth. Thus, the exponential growth equation can be used. Second, we initially consider only two populations (genotypes or species): the fast-growing F population with size $F$ and the slow-growing S population with size $S$. We do not consider a spectrum of mutants or species, since with more than two populations, an analytical solution becomes very difficult. Finally, the single top-functioning community is chosen to reproduce, which allows us to employ the simplest version of the extreme value theory (see section below for further justification).

Our goal is to select for collective composition in terms of F frequency $f = F/(S + F)$, or equivalently, S frequency $s = 1 - f$. More precisely, we want collectives such that after maturation time $\tau$, $f(\tau)$ is as close to the target value $\hat{f}$ as possible (*Figure 1*). Note that even if the target frequency has

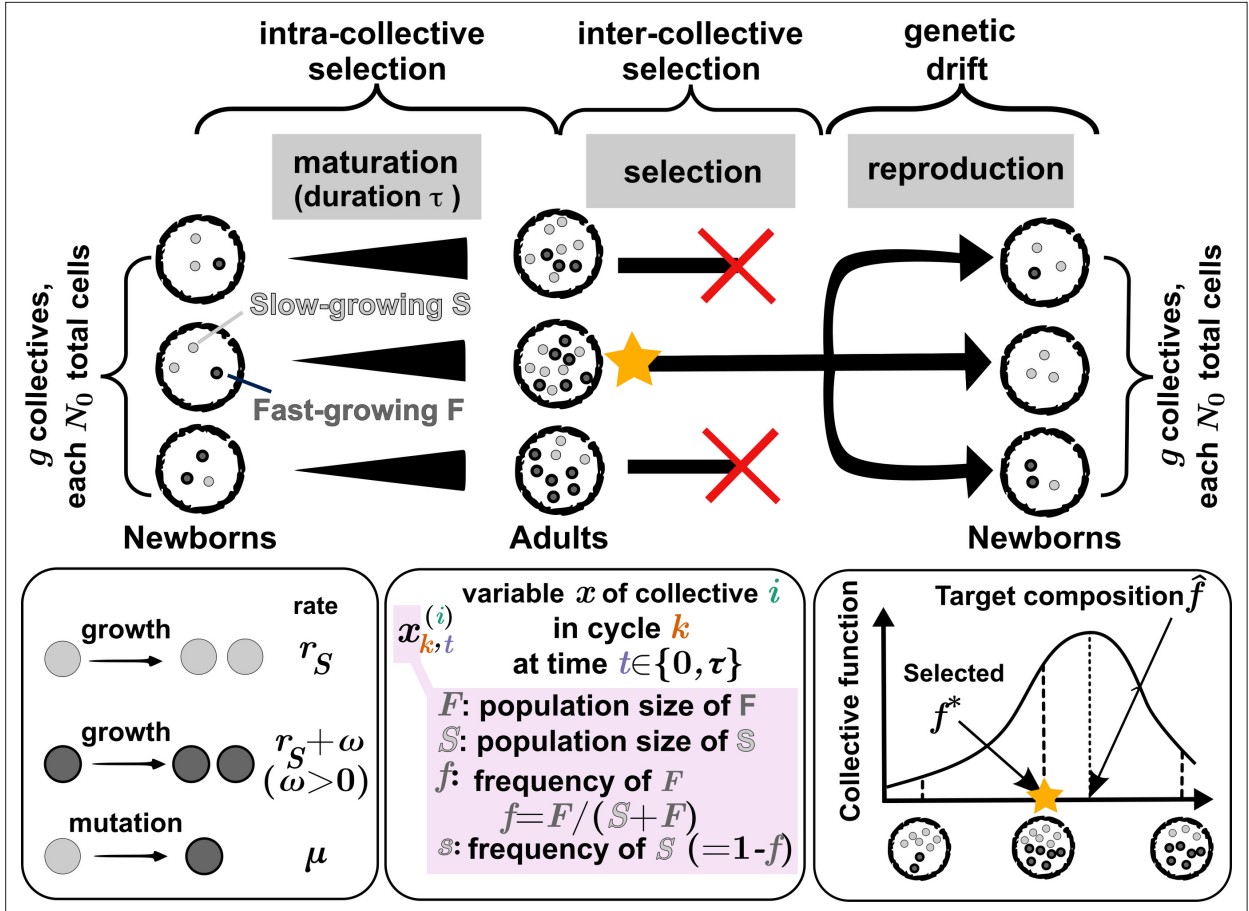

**Figure 1.** Schematic for artificial selection on collectives. Each selection cycle begins with a total of $g$ Newborn collectives, each with $N_0$ total cells of slow-growing S population (light gray dots) and fast-growing F population (dark gray dots). During maturation (over time $\tau$), S and F cells divide at rates $r_S$ and $r_S + \omega$ ($\omega > 0$), respectively, and S mutates to F at rate $\mu$. During inter-collective selection, the Adult collective with F frequency $f$ closest to the target composition $\hat{f}$ is chosen to reproduce $g$ Newborns for the next cycle. Newborns are sampled from the chosen Adult (yellow star) with $N_0$ cells per Newborn. The selection cycle is then repeated until the F frequency reaches a steady state, which may or may not be the target composition. To denote a variable $x$ of $i$-th collective in cycle $k$ at time $t$ ($0 \leq t \leq \tau$), we use notation $x_{k,t}^{(i)}$ where $x \in \{S, F, s, f\}$. Note that time $t = 0$ is for Newborns and $t = \tau$ is for Adults.

been achieved, since F frequency will always increase during maturation, inter-collective selection is required in each cycle to maintain the target frequency.

We will start with a complete model where S mutates to F at a nonzero mutation rate $\mu$. We made this choice because it is more challenging to attain or maintain the target frequency when the abundance of fast-growing F is further increased via mutations. This scenario is encountered in biotechnology: an engineered pathway will slow down cell growth, and breaking the pathway (and thus faster growth) is much easier than the other way around. When the mutation rate is set to zero, the same model can be used to capture collectives of two species with different growth rates. We show that intermediate F frequencies or equivalently, intermediate S frequencies, are the hardest targets to achieve. We then show using simulations that similar conclusions hold when selecting for a target composition in collectives of three populations.

## Model structure

A selection cycle (*Figure 1*; *Table 1*) starts with a total of $g$ Newborn collectives. At the beginning of cycle $k$ ($t = 0$), each Newborn collective has a fixed total cell number $N_0 = S_{k,0}^{(i)} + F_{k,0}^{(i)}$ where $S_{k,t}^{(i)}$ and $F_{k,t}^{(i)}$ denote the numbers of S and F cells in collective $i$ ($1 \leq i \leq g$) at time $t$ ($0 \leq t \leq \tau$) of cycle $k$. The average F frequency among the $g$ Newborn collectives in cycle $k$ is $\bar{f}_{k,0}$, such that the initial F cell number in each Newborn is drawn from the binomial distribution $\mathrm{Binom}(N_0, \bar{f}_{k,0})$.

**Table 1.** Nomenclature.

| Variables | Representing |
|---|---|
| $S$ | Number of slower-growing (S) cells |
| $F$ | Number of faster-growing (F) cells |
| $N$ | Total cell numbers in a collective, $N = S + F$ |
| $s$ | Frequency of S cells, $s = S/(S + F)$ |
| $f$ | Frequency of F cells, $f = F/(S + F) = 1 - s$ |
| $f^*$ | F frequency of the selected collective in a cycle |

| Parameters | Representing |
|---|---|
| $r_S$ | Growth rate of S |
| $\omega > 0$ | Growth rate advantage of F over S |
| $\mu$ | Mutation rate from S to F |
| $g$ | Total number of collectives |
| $\tau$ | Maturation time |
| $N_0$ | Total number of cells in Newborn, or Newborn size |
| | Target frequency in $s$ or $f$. |
| $f^L, f^H$ | *Low* and *High* thresholds of inaccessible $\hat{f}$ |
| $R_\tau$ | Fold-growth of S cells over time $\tau$, $R_\tau = e^{r_S \tau}$ |
| $W_\tau$ | Fold ratio change of F cells over S cells over time $\tau$, $W_\tau = e^{\omega \tau}$ |

Collectives are allowed to grow for time $\tau$ ('Maturation' in *Figure 1*). During maturation, S and F grow at rates $r_S$ and $r_S + \omega$ ($\omega > 0$), respectively. If maturation time $\tau$ is too small, a matured collective ('Adult') does not have enough cells to reproduce $g$ Newborn collectives with $N_0$ cells. On the other hand, if maturation time $\tau$ is too long, fast-growing F will take over. Hence, we set the maturation time $\tau = \ln(g + 1)/r_S$, which guarantees sufficient cells to produce $g$ Newborn collectives from a single Adult collective. At the end of a cycle, a single Adult with the highest function (with F frequency $f$ closest to the target frequency $\hat{f}$) is chosen to reproduce $g$ Newborn collectives, each with $N_0$ cells ('Selection' and 'Reproduction' in *Figure 1*). Note that even though S and F do not compete for nutrients, they compete for space: because the total number of cells transferred to the next cycle is fixed, an over-abundance of one population will reduce the likelihood of the other being propagated.

Collective function is dictated by the Adult's F frequency $f$. Among all Adult collectives, the selected Adult is the one whose F frequency is closest to the target value, $\hat{f}$. In contrast with findings from an earlier study (*Xie et al., 2019*), choosing top 1 is more effective than the less stringent 'choosing top 5%.' In the earlier study, variation in the collective trait is partly due to nonheritable factors such as random fluctuations in Newborn biomass. In that context, a less stringent selection criterion proved more effective, as it helped retain collectives with favorable genotypes that might have exhibited suboptimal collective traits due to unfavorable non-heritable factors. However, since this study excludes non-heritable variations in collective traits, selecting the top 1 collective is more effective than selecting the top 5% (see *Appendix 7—figure 1*).

The selected Adult, with F frequency denoted as $f^*$, is then used to reproduce $g$ offspring collectives, each with $N_0$ total cells. The number of F cells in a newborn follows a binomial distribution $B(N_0, f^*)$. By repeating the selection cycle, we aim to achieve and maintain the target composition $\hat{f}$.

Overall, our model considers mutational stochasticity, as well as demographic stochasticity in terms of stochastic birth and stochastic sampling of a parent collective by offspring collectives. Other types of stochasticity, such as environmental stochasticity and measurement noise, are not considered and require future research.

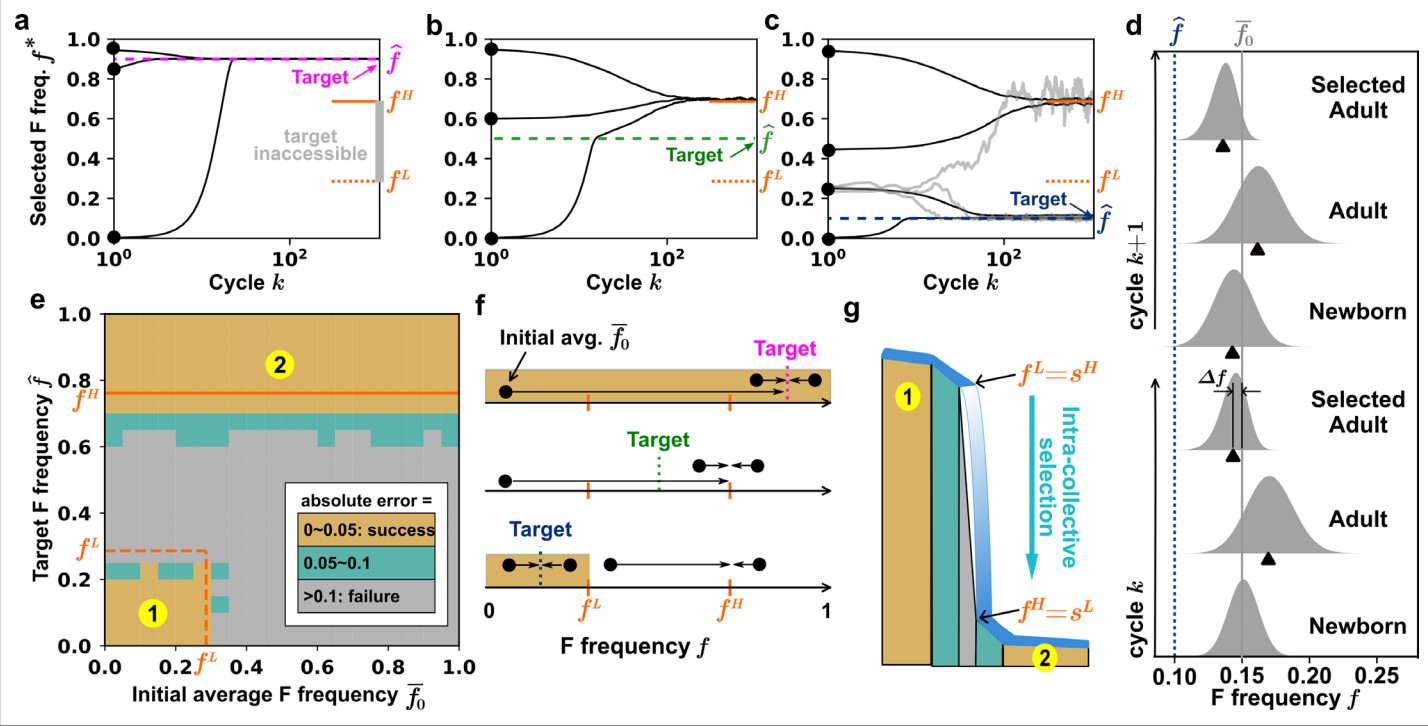

**Figure 2.** Initial and target compositions determine the success of artificial selection on collectives. (**a–c**) F frequency of the selected Adult collective ($f^*$) over cycles at different target $\hat{f}$ values (long dashed lines). $\hat{f}$ between $f^L$ and $f^H$ (orange dotted and solid line segments) is inaccessible where selection will fail. (**a**) A high target F frequency (e.g. $\hat{f} = 0.9 > f^H$; magenta) can be achieved from any initial frequency (black dots). (**b**) An intermediate target frequency (e.g. $f^L < \hat{f} = 0.5 < f^H$; green) is never achievable, as all initial conditions converge to $f^H$. (**c**) A low target frequency (e.g. $\hat{f} = 0.1 < f^L$; dark blue) is achievable, but only from initial frequencies below $f^L$. For initial frequencies at $f^L$, stochastic outcomes (gray curves) are observed: while some replicates reached the target frequency, others reached $f^H$. For parameters, we used S growth rate $r_S = 0.5$, F growth advantage $\omega = 0.03$, mutation rate $\mu = 0.0001$, maturation time $\tau \approx 4.8$, and $N_0 = 1000$. The number of collectives $g = 10$. Each black line is averaged from independent 300 realizations. (**d**) Inter-collective selection opposes intra-collective selection. We plot probability density distributions of F frequency $f$ during two consecutive cycles when selection is successful. Data correspond to cycles 31 and 32 from the second lowest initial point in c. $\Delta f$ is the selection progress within a cycle (see **Box 1**). Black triangle: median. (**e**) Two accessible regions (gold). Either high $\hat{f}$ ($\hat{f} > f^H$; region 2) or low $\hat{f}$ starting from low initial $f$ ($\hat{f} < f^L$ and $\bar{f}_{1,0} < f^L$; region 1) can be achieved. We theoretically predict (by numerically integrating **Equation 1**) $f^H$ (orange solid line) and $f^L$ (orange dotted line), which agree with simulation results (gold regions). (**f**) Example trajectories from initial compositions (black dots) to the target compositions (dashed lines). The gold areas indicate the region of initial frequencies where the target frequency can be achieved. (**g**) The tension between intra-collective selection and inter-collective selection creates a 'waterfall' phenomenon. See the main text for details.

## The success of collective selection is constrained by the target composition, and sometimes also by the initial composition

Since intra-collective selection favors F, we expect that a higher target $\hat{f}$ (a lower target $\hat{s}$) is easier to achieve. By 'achieve,' we mean that the absolute error $d$ between the target frequency $\hat{f}$ and the selected frequency averaged among independent simulations $\langle f^* \rangle$ is smaller than 0.05 (i.e. $d = |\langle f^* \rangle - \hat{f}| \leq 0.05$).

We fixed $N_0$, the total population size of a Newborn to 1000, and obtained selection dynamics for various initial and target F frequencies by implementing stochastic simulations (Appendix 1). If the target $\hat{f}$ is high (e.g. 0.9, **Figure 2a** magenta), selection is successful (computed absolute errors **Appendix 1—figure 4**): regardless of the initial frequency, $f^*$ of the chosen collective eventually converges to the target $\hat{f}$ and stays around it. In contrast, without collective-level selection (e.g. choosing a random collective to reproduce), F frequency increases until F reaches fixation (Supplementary information **Appendix 1—figure 3b**).

In contrast, an intermediate target frequency (e.g. $\hat{f} = 0.5$; **Figure 2b** green) is never achievable. High initial F frequencies (e.g. 0.95) decline toward the target but stabilize at the 'high-threshold' $f^H$ (~ 0.7, solid orange line segment in **Figure 2a-c**) above the target. Low initial F frequencies (e.g. 0) increase toward the target, but then overshoot and stabilize at the $f^H$ value.

If the target frequency is low (e.g. $\hat{f} = 0.1$; *Figure 2c* dark blue), artificial selection succeeds when the initial frequency is below the 'lower-threshold' $f^L$ (dotted orange line segment in *Figure 2a-c*). Initial F frequencies above $f^L$ (e.g. 0.45 and 0.95) converge to $f^H$ instead. Initial F frequencies near $f^L$ display stochastic trajectories, converging to either $f^H$ or $\hat{f}$.

To achieve target $\hat{f}$, inter-collective selection must overcome intra-collective selection. We can visualize the distributions of $f$ over two consecutive cycles (bottom to top, *Figure 2d*) where $f$ started above target $\hat{f}$. When newborns matured into adults, the distribution of $f$ up-shifted due to intra-collective selection. The distribution of $f$ was then down-shifted toward the target due to inter-collective selection. If the magnitude of down-shift exceeded that of up-shift, progress toward the target was made. During reproduction of collectives, the distribution of $f$ retained the same mean but became broader due to stochastic sampling by the Newborns from their parent.

In summary, two regions of target frequencies are 'accessible' (gold in *Figure 2e, f*; *Box 1*): (1) target frequencies above $f^H$ ($\hat{f} > f^H$) or (2) target frequencies below $f^L$ ($\hat{f} < f^L$) and starting at an average frequency below $f^L$ ($\bar{f}_{1,0} < f^L$).

## Intra-collective evolution is the fastest at intermediate F frequencies, creating the 'waterfall' phenomenon

To understand what gives rise to the two accessible regions, we calculated $\triangle f$, the selection progress in F frequency over two consecutive cycles (*Box 1*, *Equation 2*). The solution (*Figure 3a*, green) has the same shape as results from numerically integrating *Equation 1* (*Figure 3a*, orange) and from stochastic simulations (*Figure 3a*, blue).

If $\triangle f$ is negative, then inter-collective selection will succeed in countering intra-collective selection and reducing $f$ toward the target. $\triangle f$ is negative if the selected $f_k^*$ is low or high, but not if it is intermediate between $f^L$ and $f^H$ (*Figure 3a*). This is because the increase in $f$ during maturation is the most drastic when Newborn $f$ is intermediate (*Figure 3b*), for intuitive reasons: when Newborn $f$ is low, the increase in $f$ will be minor; when Newborn $f$ is high, the fitness advantage of F over the population average is small and hence the increase is also minor. Thus, when Newborn F frequency is intermediate, intra-collective selection is the strongest and may overwhelm inter-collective selection (*Figure 3b* and *Appendix 2—figure 2a*). Not surprisingly, similar conclusions are derived where S and F are slow-growing and fast-growing species which cannot be converted through mutations (Appendix 4 and *Appendix 4—figure 1*).

Thus, inter-collective selection is akin to a raftman rowing the raft to a target, while intra-collective selection is akin to a waterfall. This metaphor is best understood in terms of S frequency $s = 1 - f$. The lower-threshold $f^L$ corresponds to higher-threshold in $s^H = 1 - f^L$. Intra-collective selection is akin to a waterfall, driving the S frequency $s$ from high to low (*Figure 2g*). Intra-collective selection acts the strongest when $s$ is intermediate ($s^L < s < s^H$), similar to the vertical drop of the fall. Intra-collective selection acts weakly at high ($> s^H$) or low ($< s^L$) $s$, similar to the gentle sloped upper and lower pools of the fall (regions 1 and 2 of *Figure 2e and g*). Thus, an intermediate target frequency can be impossible to achieve: a raft starting from the upper pool will be flushed down to $s^L$ ($f^H$), while a raft starting from the lower pool cannot go beyond $s^L$ ($f^H$). In contrast, a low target S frequency (in the lower pool) is always achievable. Finally, a high target S frequency (in the upper pool) can only be achieved if starting from the upper pool (as the raft cannot jump to the upper pool if starting from below).

## Manipulating experimental setups to expand the achievable target region

In *Equation 2*; *Box 1*, selection progress $\triangle f$ depends on the total number of collectives under selection ($g$). $\triangle f$ also depends on the mean and the standard deviation of Adult F frequency — $\bar{f}(\tau)$ and $\sigma_f(\tau)$. *Equations 3 and 4* of *Box 1* provide simplified expressions of $\bar{f}(\tau)$ and $\sigma_f(\tau)$ when mutation rate $\mu$ has been set to 0. When the mutation rate $\mu$ is not zero (*Equations 48 and 49* in Appendix 2), selection progress is additionally influenced by $\frac{\mu}{\omega}$ (mutation rate $\mu$ scaled with fitness difference $\omega$).

Our goal is to make $\triangle f$ as negative as possible so that any increase in $f$ during collective maturation may be reduced. From *Equation 2* in *Box 1*, a small $\bar{f}(\tau)$ will facilitate collective-level selection. Additionally, a large $\sigma_f(\tau)$ will also facilitate collective-level selection due to negative $\Phi^{-1}\left(\frac{\ln 2}{g}\right)$. Note

# Box 1. Changes in the distribution of F frequency $f$ after one cycle

We consider the case where $f_k^*$, the F frequency of the selected Adult at cycle $k$, is above the target value ($f_k^* > \hat{f}$). This case is particularly challenging because intra-collective evolution favors fast-growing F and thus will further increase $f$ away from the target. From $f_k^*$, Newborns of cycle $k+1$ will have $f$ fluctuating around $f_k^*$, and after they mature, the minimum $f$ is selected ($f_{k+1}^* = \min\left[f_{k+1,\tau}^{(1)}, f_{k+1,\tau}^{(2)}, \cdots, f_{k+1,\tau}^{(g)}\right]$). If the selected composition at cycle $k+1$ can be reduced compared to that of cycle $k$ (i.e. $f_{k+1}^* < f_k^*$), the system can evolve to the lower target value.

To find $f_k^*$ values such that $f_{k+1}^* < f_k^*$, we used the median value of the conditional probability distribution $\Psi$ of $f_{k+1}^*$ given the selected $f_k^*$ at cycle $k$ (mathematical details in Appendix 2). If the median value ($\mathrm{Median}[\Psi(f_{k+1}^*|f_k^*)]$) is smaller than $f_k^*$, then selection will likely be successful since the selected Adult in cycle $k+1$ has more than 50% chance to have a reduced F frequency compared to cycle $k$.

There are two points where the median values are the same as $f_k^*$ (**Figure 3a**), which are assigned as lower-threshold ($f^L$) and higher-threshold ($f^H$).

Following the extreme value theory, the conditional probability density function $\Psi(f_{k+1}^* = f|f_k^*)$ is

$$\Psi(f_{k+1}^* = f|f_k^*) = gP_{f_{k+1,\tau}}(f|f_k^*)\left[1 - \int_0^f df' P_{f_{k+1,\tau}}(f'|f_k^*)\right]^{g-1}.$$

(1)

**Equation 1** can be described as the product between two terms related to probability: (i) $gP_{f_{k+1,\tau}}(f|f_k^*)$ describes the probability density that any one of the $g$ Adult collectives achieves $f$ given $f_k^*$, and (ii) $\left[1 - \int_0^f df' P_{f_{k+1,\tau}}(f'|f_k^*)\right]^{g-1}$ describes the probability that all other $g-1$ collectives achieve frequencies above $f$ and thus not selected.

Since computing the exact formula of Adults' $f$ distribution in cycle $k+1$ is hard, we approximate it as Gaussian with mean $\bar{f}(\tau)$ and variance $\sigma_f^2(\tau)$. The Gaussian approximation on **Equation 1** requires sharp Gaussian distributions of $S(\tau)$ and $F(\tau)$ (i.e. $\bar{S}(\tau) \gg \sigma_s(\tau)$ and $\bar{F}(\tau) \gg \sigma_F(\tau)$). Compared to Gaussian, the exact $S(\tau)$ (negative binomial) distribution and $F(\tau)$ (Luria-Delbrück) distribution are right-skewed and heavy-tailed. However, these problems are alleviated when the initial numbers of $S$ and $F$ cells are not small (on the order of 100). Indeed, the sharpness of distributions could be achieved (see **Appendix 1—figure 1**).

To obtain an analytical solution of the change in $f$ over one cycle, we first assume that in a Newborn collective, the number of S cells is distributed as Gaussian with mean $\bar{S}_0 = N_0(1 - f_k^*)$ and variance $\sigma_{S,0}^2 = N_0 f_k^*(1 - f_k^*)$. Then, the number of F cells, $F_0 = N_0 - S_0$, is distributed as Gaussian with mean $\bar{F}_0 = N_0 f_k^*$ and variance $\sigma_{f,0}^2 = N_0 f_k^*(1 - f_k^*)$. From these, we can calculate for Adult collectives the mean and variance of population sizes $F(\tau)$ (i.e. $\bar{F}(\tau)$, $\sigma_F^2(\tau)$) and $S(\tau)$ (i.e. $\bar{S}(\tau)$, $\sigma_S^2(\tau)$) (mathematical details in Appendix 1). This task is simplified by the exponential growth of S and F: $R_\tau = e^{r_s\tau}$ describes the fold growth of S over maturation time $\tau$, and since $\omega$ is the fitness advantage of F over S, $W_\tau = e^{\omega\tau}$ describes the fold change of F/S over time $\tau$. From $R_\tau$, $W_\tau$, $\frac{\mu}{\omega}$ (mutation rate scaled with the fitness difference), $f_k^*$ (F frequency in the selected collective at cycle $k$), $N_0$ (Newborn size), $\frac{\omega}{r_s}$ (relative fitness advantage), we can calculate the mean and variance of F frequency among the Adults of $k+1$ cycle ($\bar{f}(\tau)$; $\sigma_f^2(\tau)$, detailed formula in **Equations 48 and 49**).

Selection progress - the difference between the median value of the conditional probability distribution $\Psi(f_{k+1}^*|f_k^*)$ and the selected frequency of $f_k^*$ (Appendix 2) - can be expressed as:

$$\triangle f = \mathrm{Median}[\Psi(f_{k+1}^*|f_k^*)] - f_k^* = \bar{f}(\tau) + \left[\Phi^{-1}\left(\frac{\ln 2}{g}\right)\right]\sigma_f(\tau) - f_k^*,$$

(2)

where $\Phi^{-1}(\dots)$ is the inverse cumulative function of standard normal distribution (see main text for an example). We chose the median because compared to the mean, it is easier to get an analytical expression since $\Phi^{-1}(\dots)$ is known in a closed form. Regardless, using median generated results similar to simulations (**Appendix 2—figure 3**). As expected, selection progress $\triangle f$ is governed by both the mean ($\bar{f}(\tau)$) and the variation ($\sigma_f(\tau)$) in $f$ among Adults. When the mutation rate $\mu = 0$, $\bar{f}(\tau)$ and $\sigma_f(\tau)$ can be simplified to:

$$\bar{f}(\tau) = \frac{f_k^*}{\frac{1 - f_k^*}{W_\tau} + f_k^*},$$

(3)

and

$$\sigma_f{}^2(\tau) = \frac{1}{N_0 W_\tau^2} \frac{f_k^* \left(1 - f_k^*\right) \left(2 - 2f_k^* + 2f_k^{*2} - \frac{1 - f_k^*}{R_\tau W_\tau} - \frac{f_k^*}{R_\tau}\right)}{\left(\frac{1 - f_k^*}{W_\tau} + f_k^*\right)^4}.$$

(4)

In the limit of small $f_k^*$, **Equation 3** becomes $\bar{f}(\tau)|_{f_k^* \ll 1} \approx f_k^* W_\tau$ while **Equation 4** becomes $\sigma_f^2(\tau)|_{f_k^* \ll 1} = (2 - \frac{1}{R_\tau W_\tau}) f_k^* W_\tau^2 / N_0$. Thus, both Newborn size ($N_0$) and fold-change in F/S during maturation ($W_\tau$) are important determinants of selection progress.

that since $\frac{\ln 2}{g} < 0.5$ for $g \geq 2$, $\Phi^{-1}\left(\frac{\ln 2}{g}\right)$ — corresponding to the number $y$ such that the probability of a standard normal random variable being less than or equal to $y$ is $\frac{\ln 2}{g}$ — is negative.

From **Equation 4** in **Box 1**, $\sigma_f(\tau)$ will be large if Newborn size $N_0$ is small. Indeed, as Newborn size $N_0$ declines, the region of achievable target frequency expands (gold area in **Figure 4a**). If the Newborn size $N_0$ is sufficiently small (e.g. $\leq 700$ in our parameter regime), any target frequency can be reached. An analytical approximation of the maximal Newborn size permissible for all target frequencies is given in Appendix 3.

From **Equations 3 and 4** in **Box 1**, maturation time $\tau$ affects $\bar{f}(\tau)$ and $\sigma_f(\tau)$ through $W_\tau = e^{\omega\tau}$ (the fold change in F/S over $\tau$), and affects $\sigma_f(\tau)$ additionally through $R_\tau = e^{r_S\tau}$ (fold-growth of S over $\tau$). Longer $\tau$ increases $\bar{f}(\tau)$ and is thus detrimental to selection progress. The relationship between $\sigma_f(\tau)$ and $\tau$ is not monotonic (**Appendix 2—figure 2c**), meaning that an intermediate value of $\tau$ is the best for achieving large $\sigma_f(\tau)$. However, the effect of $\bar{f}(\tau)$ dominates that of $\sigma_f(\tau)$ and therefore, the region of success monotonically reduces with longer maturation time (**Figure 4c**). Similarly, $\bar{f}(\tau)$ will be small if $\omega$ (fitness advantage of F over S) is small. Indeed, as $\omega$ becomes larger, the region of success becomes smaller (**Appendix 5—figure 1**).

$g$, the number of collectives under selection, also affects selection outcomes. As $g$ increases, the value of $\Phi^{-1}\left(\frac{\ln 2}{g}\right)$ becomes more negative, and so does $\triangle f$ — meaning collective-level selection will be more effective. Intuitively, with more collectives, the chance of finding a $f$ closer to the target is more likely. Thus, a larger number of collectives broadens the region of success (**Figure 4b**). However, the effect of $g$ is not dramatic. To see why, we note that the only place that $g$ appears is **Equation 2** in $\Phi^{-1}\left(\frac{1}{g}\right)$. When $g$ becomes large, $\Phi^{-1}\left(\frac{1}{g}\right)$ is asymptotically expressed as $\Phi^{-1}\left(\frac{1}{g}\right) \approx -\sqrt{2\ln g - \ln[\ln g] + \cdots}$ (Appendix 2) (**Phllip, 1960**), and thus does not change dramatically as $g$ varies.

## The waterfall phenomenon in a higher dimension

To examine the waterfall effect in a higher dimension, we investigate a three-population system where a faster-growing population (FF) grows faster than the fast-growing population (F) which grows faster than the slow-growing population (S) (**Figure 5a** and **Appendix 8—figure 1**). In the three-population case, the evolutionary trajectory travels in a two-dimensional plane. A target population composition

can be achieved if inter-collective selection can sufficiently reduce the frequencies of F as well as FF (accessible regions, gold in *Figure 5b*).

From numerical simulations, we identified two accessible regions: a small region near FF and a band region spanning from S to F (gold in *Figure 5b i*). Intuitively, the rate at which FF grows faster than S+F is greater than the rate at which F grows faster than S (see Appendix 8). Thus, the problem can initially be reduced to a two-population problem (i.e. FF versus F+S; *Figure 5c* left), and then expanded to a three-population problem (*Figure 5c* right).

Similar to the two-population case, targets in the inaccessible region are never achievable (*Figure 5b ii*), while those in the FF region are always achievable (*Figure 5b i*). Strikingly, a target composition in an accessible region may not be achievable even when the initial composition is within the same region: once the composition escapes the accessible region, the trajectory cannot return back to the accessible region (*Figure 5biii*, the leftmost initial condition). However, if the initial position is closer to the target in the accessible region, the target becomes achievable (*Figure 5b iii*, initial condition near the bottom). Note that here, the selection outcome is path-dependent in the sense of being sensitive to initial conditions. This phenomenon is distinct from hysteresis, where path-dependence results from whether a tuning parameter is increased or decreased.

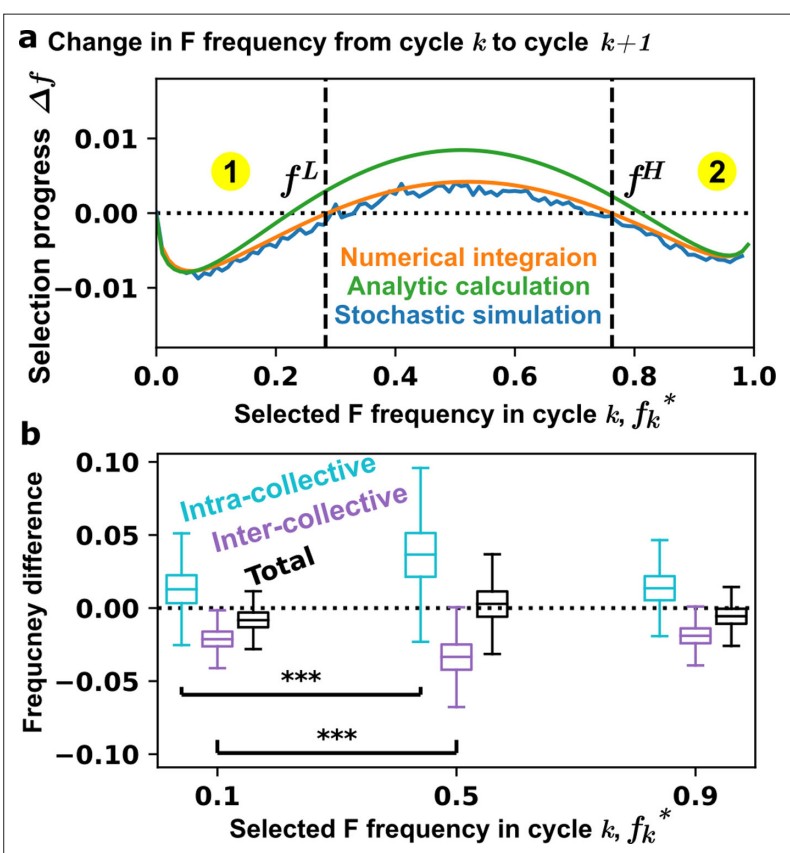

**Figure 3.** Intra-collective selection and inter-collective selection jointly set the boundaries for selection success. (**a**) The change in F frequency over one cycle. When $f_k^*$ is sufficiently low or high, inter-collective selection can lower the F frequency to below $f_k^*$ ($\Delta f < 0$). The points where $\Delta f = 0$ (in the orange line) are denoted as $f^L$ and $f^H$, corresponding to the boundaries in *Figure 2*. (**b**) The distributions of frequency differences obtained by 1000 numerical simulations. The cyan, purple, and black box plots respectively indicate the changes in F frequency after intra-collective selection (the mean frequency among the 100 Adults minus the mean frequency among the 100 Newborns during maturation), after inter-collective selection (the frequency of the 1 selected Adult minus the mean frequency among the 100 Adults), and over one selection cycle (the frequency of the selected Adult of one cycle minus that of the previous cycle). The box ranges from 25% to 75% of the distribution, and the median is indicated by a line across the box. The upper and lower whiskers indicate maximum and minimum values of the distribution. ***p<0.001 in an unpaired $t$-test.

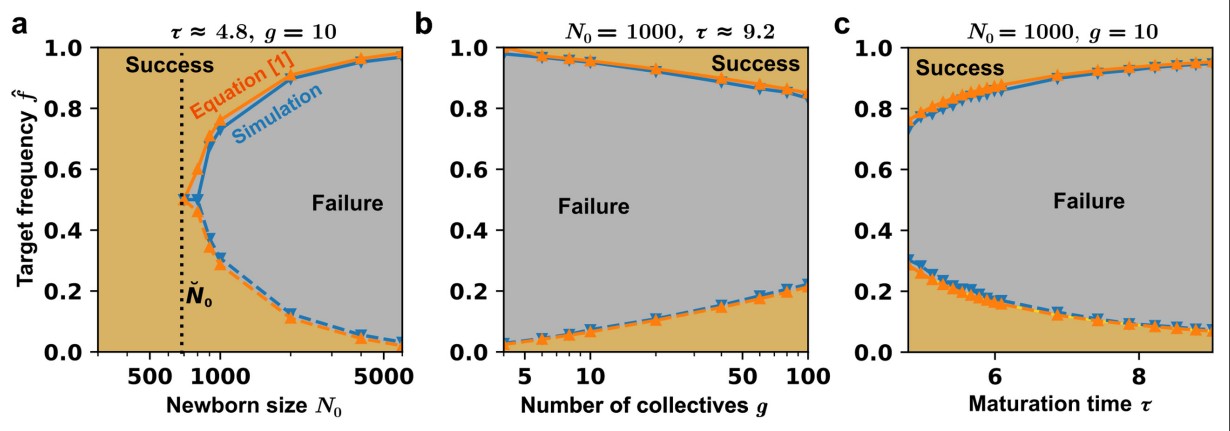

**Figure 4.** Expanding the region of success for artificial collective selection. (**a**) Reducing the population size in Newborn $N_0$ expands the region of success. In the gold area, the probability that $f^*_{k+1}$ becomes smaller than $f^*_k$ in a cycle is more than 50%. We used $g = 10$ and $\tau \approx 4.8$. *Figures 2–3* correspond to $\check{N}_0 = 1000$ in this graph. Black dotted line indicates the critical Newborn size below which all target frequencies can be achieved. (**b**) Increasing the total number of collectives $g$ also expands the region of success, although only slightly. We used a fixed Newborn size $N_0 = 1000$. The maturation time $\tau = \log(100)/r_S \approx 9.2$ is set to be long enough so that an Adult can generate at least 100 Newborns. (**c**) Increasing the maturation time shrinks the region of success. We used a fixed Newborn size $N_0 = 1000$ and number of collectives $g = 10$.

In conclusion, we have investigated the evolutionary trajectories of population compositions in collectives under selection, which are governed by intra-collective selection (which favors fast-growing populations) and inter-collective selection (which, in our case, strives to counter fast-growing populations). Intra-collective selection has the strongest effect at intermediate frequencies of faster-growing populations, potentially creating an inaccessible region of target frequency analogous to the vertical drop of a waterfall. High and low target frequencies are both accessible, analogous to the lower and the upper pools of a waterfall, respectively. A less challenging target (high $\hat{f}$; low $\hat{s}$) is achievable from any initial position. In contrast, a more challenging target (low $\hat{f}$; high $\hat{s}$) is only achievable if the entire trajectory is contained within the region, similar to a raft striving to reach a point in the upper pool must start at and remain in the upper pool. Our work suggests that the strength of intra-collective selection is not constant, and that strategically choosing an appropriate starting point can be essential for successful collective selection.

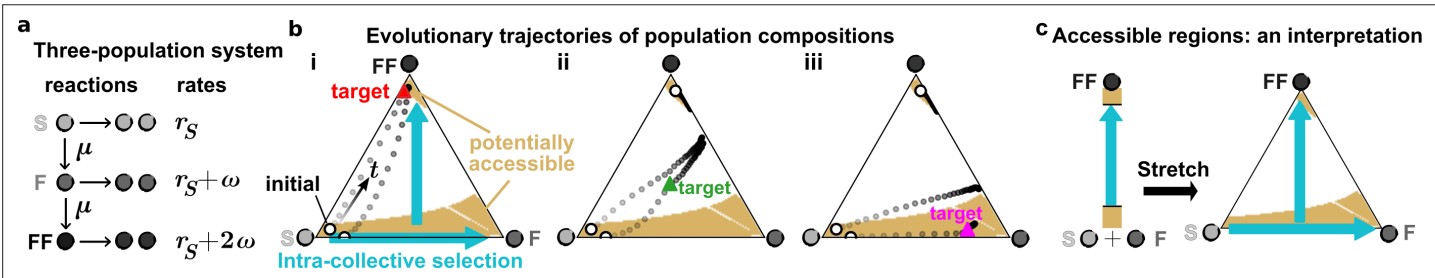

**Figure 5.** In higher dimensions, the success of artificial selection requires the entire evolutionary trajectory remaining in the accessible region. (**a**) During collective maturation, a slow-growing population (**S**) (with growth rate $r_S$; light gray) can mutate to a fast-growing population (**F**) (with growth rate $r_S + \omega$; medium gray), which can mutate further into a faster-growing population (**FF**) (with growth rate $r_S + 2\omega$; dark gray). Here, the rates of both mutational steps are $\mu$, and $\omega > 0$. (**b**) Evolutionary trajectories from various initial compositions (open circles) to various targets (filled triangles). Intra-collective evolution favors FF over F (vertical blue arrow) over S (horizontal blue arrow). The accessible regions are marked gold (see Appendix 1). We obtain final compositions starting from several initial compositions while aiming for different target compositions in i, ii, and iii. The evolutionary trajectories are shown in dots with color gradients from initial time (light grey) to final time (dark grey). (i) A target composition with a high FF frequency is always achievable. (ii) A target composition with intermediate FF frequency is never achievable. (iii) A target composition with low FF frequency is achievable only if starting from an appropriate initial composition such that the entire trajectory never meanders away from the accessible region. The figures are drawn using the mpltern package (*Ikeda et al., 2019*). (**c**) The accessible region in the three-population problem is interpreted as an extension of the two-population problem. First, the accessible region between FF and S+F is given, and then the S+F region is stretched into S and F.

# Materials and methods

## Stochastic simulations

A selection cycle is composed of three steps: maturation, selection, and reproduction. At the beginning of the cycle $k$, a collective $i$ has $S_{k,0}^{(i)}$ slow-growing cells and $F_{k,0}^{(i)}$ fast-growing cells. At the first cycle, the mean F frequency of collectives is set to be $\bar{f}_{1,0}$. $F_{1,0}^{(i)}$ is sampled from the binomial distribution with mean $N_0\bar{f}_{1,0}$. Then, $S_{1,0}^{(i)}(= N_0 - F_{1,0}^{(i)})$ S cells are in the collective $i$. In the maturation step, we calculate $S_{k,\tau}^{(i)}$ and $F_{k,\tau}^{(i)}$ by using stochastic simulation. We can simulate the division and mutation of each individual cell stochastically by using the tau-leaping algorithm (**Gillespie, 2001**; **Cao et al., 2006**; see **Appendix 1—figure 3**). However, individual-based simulations require a long computing time. Instead, we randomly sample $S_{k,\tau}^{(i)}$ and $F_{k,\tau}^{(i)}$ from the joint probability density distribution $P(S_{k,\tau}^{(i)}, F_{k,\tau}^{(i)})$. To obtain $P(S_{k,\tau}^{(i)}, F_{k,\tau}^{(i)})$, we solve the master equation which describes the time evolution of the probability distribution $P(S_{k,t}^{(i)}, F_{k,t}^{(i)})$ under the random processes (see Appendix 1). We assumed that $S_{k,\tau}^{(i)}$ and $F_{k,\tau}^{(i)}$ are independent (as S and F populations grow independently without ecological interactions), and thus $P(S_{k,\tau}^{(i)}, F_{k,\tau}^{(i)})$ is product of two probability density functions $P(S_{k,\tau}^{(i)})$ and $P(F_{k,\tau}^{(i)})$. Each distribution follows a Gaussian distribution, with the mean and variance numerically obtained from ordinary differential equations derived from the master equation (see Appendix 1). We choose the collective with the closest frequency to the target $\hat{f}$ to generates $g$ Newborns. The number of F cells is sampled from the binomial distribution with the mean of $N_0 f_k^*$. We start a new cycle with those Newborn collectives. Then, the number of S cells in a collective $i$ is $S_{k+1,0}^{(i)} = N_0 - F_{k+1,0}^{(i)}$.

## Analytical approach to the conditional probability

The conditional probability distribution $\Psi(f_{k+1}^*|f_k^*)$ of observing $f_{k+1}^*$ at a given $f_k^*$ is calculated by the following procedure. Given the selected collective in cycle $k$ with $f_k^*$, the collective-level reproduction proceeds by sampling $g$ Newborn collectives with $N_0$ cells in cycle $k + 1$. Each Newborn collective contains certain F numbers $F_{k+1,0}^{(1)}, \cdots, F_{k+1,0}^{(g)}$ at the beginning of the cycle $k + 1$, which can be mapped into $f_{k+1,0}^{(1)}, \cdots, f_{k+1,0}^{(g)}$ with the constraint of $N_0$ cells. If the number of cells in the selected collective is large enough, the joint conditional distribution function $P\left(f_{k+1,0}^{(1)}, \cdots, f_{k+1,0}^{(g)}|f_k^*\right)$ is well described by the product of $g$ independent and identical Gaussian distribution $\mathcal{N}(\mu, \sigma^2)$. So we consider the frequencies of $g$ Newborn collectives as $g$ identical copies of the Gaussian random variable $f_{k+1,0}$. The mean and variance of $f_{k+1,0}$ are given by $m = f_k^*$ and $\sigma^2 = f_k^*(1 - f_k^*)/N_0$. Then, the conditional probability distribution function of $f_{k+1,0}$ being $\zeta$ is given by

$$P_{f_{k+1,0}}(\zeta|f_k^*) = \frac{1}{\sqrt{2\pi}} \exp\left(-\frac{(\zeta - m)^2}{2\sigma^2}\right). \tag{5}$$

After the reproduction step, the Newborn collectives grow for time $\tau$. The frequency is changed from the given frequency $\zeta$ to $f$ by division and mutation processes. We assume that the frequency $f$ of an Adult is also approximated by a Gaussian random variable $\mathcal{N}(\bar{f}(\tau), \sigma_f^2(\tau))$. The mean $\bar{f}(\tau)$ and variance $\sigma_f^2(\tau)$ are calculated by using means and variances of $S$ and $F$ (see Appendix 2). Since $\bar{f}(\tau)$ and $\sigma_f^2(\tau)$ also depend on $\zeta$, the conditional probability distribution function of $f_{k+1,\tau}$ being $f$ is given by

$$P_{f_{k+1,\tau}}(f|\zeta) = \frac{1}{\sqrt{2\pi}} \exp\left(-\frac{(f - \bar{f}(\tau))^2}{2\sigma_f^2(\tau)}\right). \tag{6}$$

The conditional probability distribution of an Adult collective in cycle $k + 1$ ($f_{k+1,\tau}$) to have frequency $f$ at a given $f_k^*$ is calculated by multiplying two Gaussian distribution functions and integrating overall $\zeta$ values, which is given by

$$P_{f_{k+1,\tau}}(f|f_k^*) = \int_0^1 d\zeta\, P_{f_{k+1,\tau}}(f|\zeta)\, P_{f_{k+1,0}}(\zeta|f_k^*). \tag{7}$$

Since we select the minimum frequency $f_{k+1}^{\min}$ among $g$ identical copies of $f_{k+1,\tau}$, the conditional probability distribution function of $f_{k+1}^{\min}$ follows a minimum value distribution, which is given in

*Equation 1*. Here, for the case of $\hat{f} < f_k^*$, the selected frequency $f_{k+1,0}$ is the minimum frequency $f_{k+1}^{\min}$. So we have $\Psi(f_{k+1}^*|f_k^*)$ by replacing $f_{k+1}^{\min}$ with $f_{k+1}^*$.

We assume that the conditional probability distribution in *Equation 7* follows a normal distribution, whose mean and variances are described by *Equation 48* and *Equation 49*. Then, the extreme value theory (*Gumbel, 1958*) estimates the median of the selected Adult by

$$\text{Median}(f_{k+1}^*) = \bar{f}(\tau) + \left[ \Phi^{-1}\left(\frac{\ln 2}{g}\right) \right] \sigma_f(\tau). \tag{8}$$

The selection progress $\Delta f$ in *Equation 2* is obtained by subtracting $f_k^*$ from *Equation 8*.

# Acknowledgements

J Lee and HJ Park were supported by the National Research Foundation of Korea grant funded by the Korean government (MSIT), Grant No. RS-2023–00214071 and RS-2024–00460958 and by an appointment to the JRG Program at the APCTP through the Science and Technology Promotion Fund and the Lottery Fund of the Korean Government. W Shou was supported by the Academy of Medical Sciences Professorship and a Royal Society Wolfson Fellowship. This was also supported by the Korean Local Governments–Gyeongsangbuk-do Province and Pohang City and INHA UNIVERSITY Research Grant. We thank Su-Chan Park, Li Xie, Alex Yuan, and Botond Major for constructive comments and discussions.

# Additional information

## Competing interests

Wenying Shou: Reviewing editor, eLife. The other authors declare that no competing interests exist.

## Funding

| Funder | Grant reference number | Author |
| --- | --- | --- |
| National Research Foundation of Korea | RS-2023-00214071 | Juhee Lee<br>Hye Jin Park |
| National Research Foundation of Korea | RS-2024-00460958 | Juhee Lee<br>Hye Jin Park |
| Asia Pacific Center for Theoretical Physics | JRG Program | Juhee Lee<br>Hye Jin Park |
| Academy of Medical Sciences | Professorship | Wenying Shou |
| Royal Society | Wolfson Fellowship | Wenying Shou |
| Inha University | Research grant | Hye Jin Park |

The funders had no role in study design, data collection and interpretation, or the decision to submit the work for publication.

## Author contributions

Juhee Lee, Conceptualization, Formal analysis, Validation, Investigation, Visualization, Methodology, Writing – original draft, Writing – review and editing; Wenying Shou, Conceptualization, Supervision, Writing – original draft, Writing – review and editing; Hye Jin Park, Conceptualization, Supervision, Validation, Writing – original draft, Writing – review and editing

## Author ORCIDs

Juhee Lee ⓘ https://orcid.org/0000-0003-3318-6377
Wenying Shou ⓘ https://orcid.org/0000-0001-5693-381X
Hye Jin Park ⓘ https://orcid.org/0000-0003-3552-6275

Reviewer #1 (Public review): https://doi.org/10.7554/eLife.97461.3.sa1

Reviewer #3 (Public review): https://doi.org/10.7554/eLife.97461.3.sa2
Author response https://doi.org/10.7554/eLife.97461.3.sa3

## Additional files

### Supplementary files
MDAR checklist

### Data availability
Data and source code of stochastic simulations are available in https://github.com/schwarzg/artificial_selection_collective_composition (copy archived at *Lee, 2025*).

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

# Appendix 1

## Stochastic simulation of the selection cycle

In the main text, we design a simple model of artificial selection on collectives. The selection cycle starts with $g$ 'Newborn' collectives which consist of two populations - slow-growing population (S) and fast-growing population (F). S mutates to F at a rate $\mu$. The newborns mature for a fixed time $\tau$. The matured collective ('Adult') with the highest function (with F frequency $f$ closest to the target $\hat{f}$) is chosen to reproduce $g$ Newborn collectives, each with $N_0$ cells.

In our selection cycle, variation among collectives mainly resulted from demographic noises during cell birth, cell mutation, and collective reproduction. In this section, we provide details of the simulation.

## Maturation

Here, we calculate the cell numbers during maturation. Each collective $i$ ($i = 1, \ldots, g$) has $S_{k,t}^{(i)}$ S cells and $F_{k,t}^{(i)}$ F cells where $k$ is the cycle number and $t$ indicates time ($0 \leq t \leq \tau$). At the beginning of cycle $k$ ($t = 0$), each Newborn collective has a total of $N_0 = S_{k,0}^{(i)} + F_{k,0}^{(i)}$ cells. The collectives are allowed to 'mature' for $t = \tau$ during which S and F grow at rates $r_S$ and $r_S + \omega$ ($\omega > 0$), respectively. In this subsection, we ignore the cycle number index $k$ and the collective index $k$ for convenience. That is, we denote $S_{k,t}^{(i)}$ and $F_{k,t}^{(i)}$ as $S(t)$ and $F(t)$, respectively.

We describe cell divisions of S and F cells and mutation from S to F with the following chemical reaction rules:

$$S \xrightarrow{r_S} S + S, \tag{9}$$

$$F \xrightarrow[\omega > 0]{r_S + \omega} F + F, \tag{10}$$

$$S \xrightarrow{\mu} F, \tag{11}$$

One can run an individual-based simulation by counting the number of events occurring during collective maturation via the tau-leaping algorithm (*Gillespie, 2001*; *Cao et al., 2006*) to generate a sample trajectory of $S(t)$ and $F(t)$ for each collective. However, the individual-based simulation requires long computing times due to a large number of random events to be counted. Hence, we used a 'sampling method' by sampling the numbers of S and F cells in collectives from a joint probability density distribution (jpdf) $P(S, F, t)$ which denotes the probability density to have $S$ number of S cells and $f_{k+1,0}$ number of F cells at time $t$ in the cycle. To do so, we require an analytical expression of $P(S, F, t)$.

First, we assume that the chemical reactions in *Equations 9–11* occur independently, and never occur simultaneously within a short time interval $[t, t + dt)$. Then, the differential of $P(S, F, t)$ with respect to time is given by

$$\begin{aligned}
\frac{dP(S, F, t)}{dt} &= r_S(S - 1)P(S - 1, F, t) \\
&\quad + (r_S + \omega)(F - 1)P(S, F - 1, t) \\
&\quad + \mu(S + 1)P(S + 1, F - 1, t) \\
&\quad - (r_S S + (r_S + \omega)F + \mu S)P(S, F, t).
\end{aligned} \tag{12}$$

This master equation describes a probability density 'flux' at the state $(S, F)$. The first term describes the scenario where a single birth event of a S cell happens during time interval $[t, t + dt)$, which changes the collective's composition from $(S - 1, F)$ to $(S, F)$. Similarly, the second term comes from a birth event of an F cell. The third term indicates the mutation event from $(S + 1, F - 1)$ to $(S, F)$. The last term corresponds to the outflow of probability density by birth and mutation processes, which describes the changes from $(S, F)$ to any other states.

Calculating the exact form of $P(S, F, t)$ is not simple. Instead, we assume that the mutation rate is much smaller than the growth rates, and hence the correlation between $S$ and $F$ is sufficiently small. Additionally, S and F do not interact ecologically. Then, we can express $P(S, F, t)$ as a product of two probability density functions (pdf) of $S(t)$ and $F(t)$, $P(S, F, t) = P(S, t)P(F, t)$. We assume that each pdf

of $S$ and $F$ can be approximated as Gaussian ($\mathcal{N}$), which is supported by the Central Limit Theorem and **Appendix 1—figure 1**. In more detail, the cell numbers $S$ and $F$ are mainly determined by growth (**Equations 9; 10**), and also mutations (**Equation 11**). Even though the number of events would be different among different realizations, the mean numbers of events will follow Gaussian distributions. So, we can simply assume that the distributions of cell numbers also follow Gaussian distributions. This assumption requires that the distributions have insignificant skewness and no heavy tails, which we will numerically check afterwards. The pdfs of $S(t)$ and $F(t)$ are given by

$$P(S,t) = \frac{1}{\sqrt{2\pi\sigma_S^2(t)}}e^{-\frac{(S(t)-\overline{S}(t))^2}{2\sigma_S^2(t)}} \tag{13}$$

and

$$P(F,t) = \frac{1}{\sqrt{2\pi\sigma_F^2(t)}}e^{-\frac{(F(t)-\overline{F}(t))^2}{2\sigma_F^2(t)}} . \tag{14}$$

That is, $P(S,F,t)$ is written as

$$P(S,F,t) = \frac{1}{\sqrt{2\pi\sigma_S^2(t)}}\frac{1}{\sqrt{2\pi\sigma_F^2(t)}}e^{-\frac{(S(t)-\overline{S}(t))^2}{2\sigma_S^2(t)}-\frac{(F(t)-\overline{F}(t))^2}{2\sigma_F^2(t)}} . \tag{15}$$

Now we need means ($\overline{S}(t)$ and $\overline{F}(t)$) and variances ($\sigma_S^2(t)$ and $\sigma_F^2(t)$) of S and F cell numbers to express the distribution analytically.

The means are defined by $\overline{S}(t) = \sum_{S=0}^{\infty}\sum_{F=0}^{\infty}S\,P(S,F,t)$ and $\overline{F}(t) = \sum_{S=0}^{\infty}\sum_{F=0}^{\infty}F\,P(S,F,t)$. The differential equations for means are obtained by applying the definition to the master equation in **Equation 12**, as

$$\frac{d\overline{S}(t)}{dt} = r_S\overline{S}(t) - \mu\overline{S}(t), \tag{16}$$

$$\frac{d\overline{F}(t)}{dt} = (r_S + \omega)\overline{F}(t) + \mu\overline{S}(t). \tag{17}$$

We assume that the mutation rate $\mu$ is much smaller than $r_S$ and $\omega$. By solving **Equation 16** and **Equation 17**, the means $\overline{S}(t)$ and $\overline{F}(t)$ are given by

$$\overline{S}(t) = \overline{S}_o e^{(r_S-\mu)t} \approx \overline{S}_o e^{r_St}, \tag{18}$$

$$\overline{F}(t) = \overline{F}_o e^{(r_S+\omega)t} + \frac{\mu\overline{S}_o}{\omega+\mu}\left(e^{(r_S+\omega)t} - e^{(r_S-\mu)t}\right) \approx \overline{F}_o e^{(r_S+\omega)t} + \frac{\mu\overline{S}_o}{\omega}\left(e^{(r_S+\omega)t} - e^{r_St}\right), \tag{19}$$

where $\overline{S}_o \equiv \overline{S}(0)$ and $\overline{F}_o \equiv \overline{F}(0)$ are the mean numbers of S and F cells at the beginning of cycle, $t = 0$. Note that the second term of **Equation 19** is consistent with previous studies (**Zheng, 1999**). Now we introduce factors $R_t = e^{r_St}$ and $W_t = e^{\omega t}$ in **Equations 18; 19** in order to simplify the formula. $R_t$ is the multiplying factor by which the S cell number increases after time $t$. $W_t$ is the fold change in $F/S$. Then, we can rewrite

$$\overline{S}(t) = \overline{S}_o R_t, \tag{20}$$

$$\overline{F}(t) = \overline{F}_o R_t W_t + \frac{\mu\overline{S}_o}{\omega}R_t(W_t - 1). \tag{21}$$

We define the second momenta of $S$ and $F$ as

$$\overline{S^2} = \sum_{S=0}^{\infty} \sum_{F=0}^{\infty} S^2 P(S,F,t), \tag{22}$$

$$\overline{F^2} = \sum_{S=0}^{\infty} \sum_{F=0}^{\infty} F^2 P(S,F,t). \tag{23}$$

Then, the corresponding differential equations are given by

$$\frac{d\overline{S^2}}{dt} = 2(r_S - \mu)\overline{S^2} + (r_S + \mu)\overline{S}, \tag{24}$$

$$\frac{d\overline{F^2}}{dt} = 2(r_S + \omega)\overline{F^2} + (r_S + \omega)\overline{F} + 2\mu\overline{SF} + \mu\overline{S}. \tag{25}$$

The solution of **Equation 24** is

$$\overline{S^2}(t) = \frac{r_S + \mu}{\mu - r_S}\overline{S}_o \left[e^{(\mu - r_S)t} - 1\right] e^{2(r_S - \mu)t} + \overline{S^2}_o e^{2(r_S - \mu)t} \tag{26}$$

$$\approx \overline{S_o^2} e^{2r_S t} + \overline{S}_o e^{r_S t}(e^{r_S t} - 1), \tag{27}$$

where $\overline{S_o^2} \equiv \overline{S^2}(0)$ is the second moment of initial values. Thus, the variance $\sigma_S^2(t) = \overline{S^2}(t) - [\overline{S}(t)]^2$ is

$$\sigma_S^2(t) = \sigma_{S,0}^2 e^{2r_S t} + \overline{S}_o e^{r_S t}(e^{r_S t} - 1) \tag{28}$$

$$= \sigma_{S,0}^2 R_t^2 + \overline{S}_o R_t(R_t - 1) \tag{29}$$

where $\sigma_{S,0}^2 \equiv \sigma_S^2(0)$ is a variance of S cell numbers at $t = 0$. In **Equation 25**, we require $\overline{SF}(t) = \sum_{S=0}^{\infty} \sum_{F=0}^{\infty} SFP(S,F,t)$ to calculate $\overline{F^2}(t)$. **Equation 12** provides a differential equation for $\overline{SF}(t)$ as

$$\frac{d\overline{SF}}{dt} = (2r_S + \omega - \mu)\overline{SF} + \mu\left(\overline{S^2} - \overline{S}\right). \tag{30}$$

The solution of **Equation 30** is given by

$$\begin{aligned}\overline{SF}(t) = {} & \overline{SF}_o e^{(2r_S + \omega)t} \\ & + \frac{\mu}{\omega}\left(\overline{S^2}_0 + \overline{S}_0\right)\left(e^{(2r_S + \omega)t} - e^{2r_S t}\right) \\ & - \frac{2\mu\overline{S}_0}{r_S + \omega}\left(e^{(2r_S + \omega)t} - e^{r_S t}\right).\end{aligned} \tag{31}$$

By using **Equation 31**, the solution of **Equation 25** is given by

$$\begin{aligned}\overline{F^2}(t) = {} & \overline{F^2}_0 e^{2(r_S + \omega)t} + \overline{F}_0 e^{(r_S + \omega)t}\left(e^{(r_S + \omega)t} - 1\right) \\ & + \frac{\mu\overline{S}_o}{\omega}\left(\frac{2\omega}{r + 2\omega}e^{2(r_S + \omega)t} - e^{(r_S + \omega)t} + \frac{r}{r + 2\omega}e^{r_S t}\right) \\ & + \frac{2\mu\overline{SF}_0}{\omega}e^{(r_S + \omega)t}\left(e^{(r_S + \omega)t} - e^{r_S t}\right) \\ & + O(\mu^2),\end{aligned} \tag{32}$$

where $\overline{F_o^2} \equiv \overline{F^2}(0)$ is the second moment of initial values. Thus, the variance $\sigma_F^2(t) = \overline{F^2}(t) - [\overline{F}(t)]^2$ is given, up to the order of $\mu$, by

$$\sigma_F{}^2(t) = \sigma_{F,0}^2 e^{2(r_S+\omega)t} + \overline{F}_0 e^{(r_S+\omega)t}\left(e^{(r_S+\omega)t} - 1\right)$$

$$+ \frac{\mu \overline{S}_o}{\omega}\left(\frac{2\omega}{r_S+2\omega}e^{2(r_S+\omega)t} - e^{(r_S+\omega)t} + \frac{r_S}{r_S+2\omega}e^{r_S t}\right)$$

$$+ \frac{2\mu\left(\overline{SF}_0 - \overline{S}_0\,\overline{F}_0\right)}{\omega}e^{(r_S+\omega)t}\left(e^{(r_S+\omega)t} - e^{r_S t}\right)$$

$$+ O\left(\mu^2\right).$$

(33)

Using $R_\tau = e^{r_S \tau}$ and $W_\tau = e^{\omega \tau}$, we rewrite

$$\sigma_F^2(t) = \sigma_{F,0}^2 R_t^2 W_t^2 + \overline{F}_0 R_t W_t (R_t W_t - 1)$$

$$+ \frac{\mu \overline{S}_o R_t}{\omega}\left(\frac{2\omega}{r_S+2\omega}R_t W_t^2 - W_t + \frac{r_S}{r_S+2\omega}\right)$$

$$+ \frac{2\mu(\overline{SF}_0 - \overline{S}_0\overline{F}_0)}{\omega}R_t^2 W_t(W_t - 1)$$

$$+ O(\mu^2).$$

(34)

Using *Equations 18; 19; 28; 33*, we construct pdfs for $S(t)$ and $F(t)$ at the end of cycle $t = \tau$. Then, we randomly sample a number from $P(S, \tau)$ for $S(\tau)$ and another number from $P(F, \tau)$ for $F(\tau)$. Those two numbers are cell numbers in a single Adult. We repeat this process for each Newborn to get cell numbers of all Adults. Note that the initial values for the Newborn are $\overline{S}_0 = S_{k,0}^{(i)}$, $\overline{F}_0 = F_{k,0}^{(i)}$, $\sigma_{S,0}^2 = 0$, and $\sigma_{F,0}^2 = 0$. This process only requires two random numbers per collective, while the result is consistent with the individual-based simulation.

Now, we check the validity of the Gaussian approximation for probability density functions of S and F populations. If we consider mutation from S to F as death in the S population, then the process in S corresponds to a branching process with death. Also, the birth process in F, including mutation, results in a Luria-Delbrück distribution (*Zheng, 1999*). Thus, the distributions of Adults' S and F numbers are more skewed and heavy-tailed than Gaussian. But this problem is alleviated by larger initial S and F numbers and when the maturation time $\tau$ is not very long (see *Appendix 1—figure 1*). Since we usually consider larger initial cell numbers, we use the Gaussian approximation on S and F populations in further calculations.

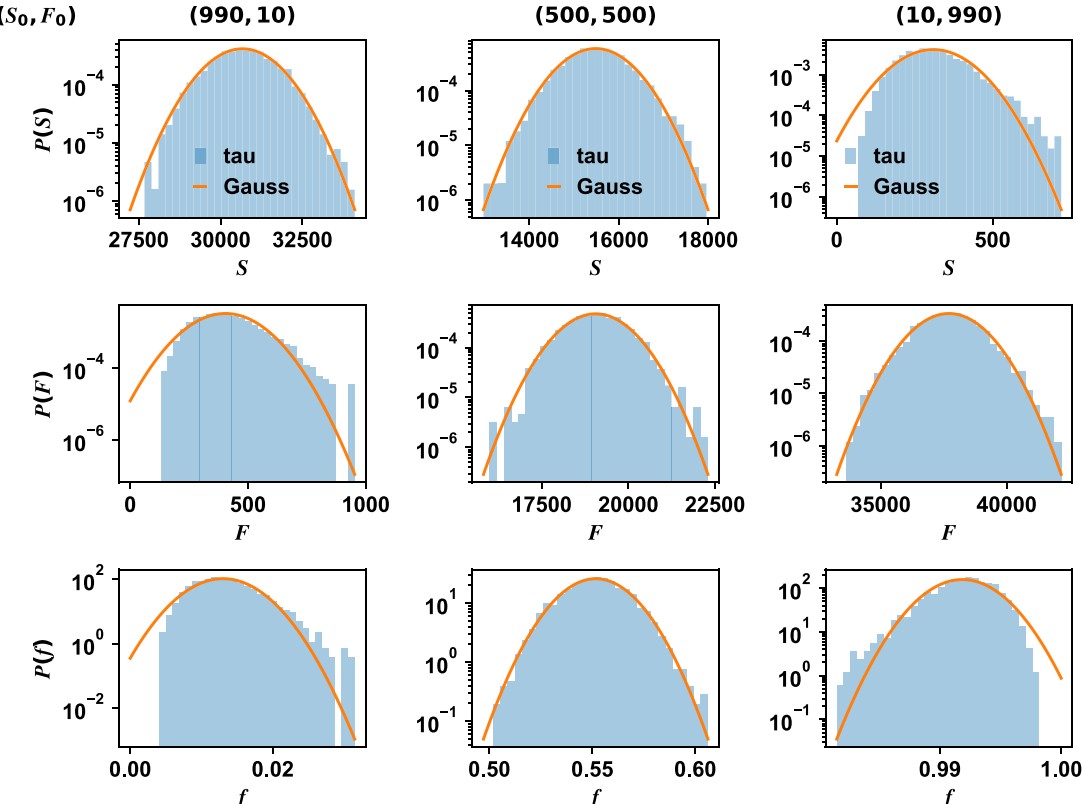

**Appendix 1—figure 1.** Comparison between the calculated Gaussian distribution ('Gauss,' with the mean and variances computed from *Equations 18; 19; 28; 33*) and simulations using tau-leaping ('tau'). The simulations run 3000 times. The initial number of cells are $(S_0, F_0) = (990, 10)$, $(500, 500)$, and $(10, 990)$ for each column. The parameters $r = 0.5$, $\omega = 0.03$, $\mu = 0.0001$, and $\tau = 4.8$ are used.

## Selection

After sampling cell numbers of each Adult in the maturation step, we compute the F frequencies in each collectives $\{f_{k,\tau}^{(1)}, \cdots, f_{k,\tau}^{(g)}\}$. We denote the F frequency of collective $i$ at time $t = \tau$ in cycle $k$ as $f_{k,\tau}^{(i)}$. Among the $g$ Adults, we select one collective with the F frequency which is the closest value to the target frequency $\hat{f}$. The selected Adult's F frequency value is denoted by $f_k^*$. In mathematical expression, the selected frequency is defined by

$$f_k^* = \mathrm{argmin}_{f_{k,\tau}^{(i)}, i \in \{1, \cdots, g\}} |f_{k,\tau}^{(i)} - \hat{f}|. \tag{35}$$

## Reproduction

Using the chosen Adult, we generate $g$ Newborn collectives for the next cycle $k + 1$. The most natural way is consecutive random sampling $N_0$ cells from the selected Adult without replacement. In the mathematical expression, we first randomly sample $F_{k+1,0}^{(1)}$ F cells and draw $S_{k+1,0}^{(1)} = N_0 - F_{k+1,0}^{(1)}$ S cells from the selected Adult. Next, we sample $F_{k+1,0}^{(2)}$ F cells and $S_{k+1,0}^{(2)} = N_0 - F_{k+1,0}^{(2)}$ S cells from the remaining cells in the Adult. We repeat the process $g$ times. Then the jpdf to choose $F_{k+1,0}^{(1)}, \cdots, F_{k+1,0}^{(g)}$ F cells, $P(F_{k+1,0}^{(1)}, \cdots, F_{k+1,0}^{(g)})$, follows a multivariate hypergeometric distribution.

If we assume that the selected Adult size $N_k^* = S_k^* + F_k^*$ is large enough compared to Newborn size $N_0$, the consecutive sampling is well approximated to the independent binomial sampling (see *Appendix 1—figure 2*). Thus, we independently sample $g$ numbers of F cells, $\{F_{k+1,0}^{(1)}, \cdots, F_{k+1,0}^{(g)}\}$, from the binomial distribution. The probability mass function of each $F_{k+1,0}^{(i)}$ is given by

$$P_{F_{k+1,0}^{(i)}}(F) = \frac{N_0!}{(N_0 - F)!F!}(f_k^*)^F (1 - f_k^*)^{N_0 - F}. \tag{36}$$

After sampling, the numbers of S cells are set to be $S_{k+1}^{(i)}(0) = N_0 - F_{k+1,0}^{(i)}$ for each collective. We can now start cycle $k+1$ with these Newborn collectives. By repeating the above three steps (maturation, selection, and reproduction), we run the simulation until F frequency reaches a stationary state.

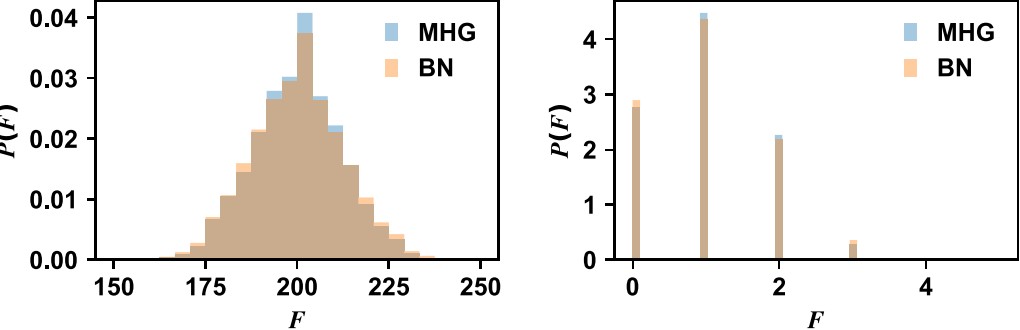

**Appendix 1—figure 2.** Congruence between consecutive sampling (MHG for multivariate hypergeometric distribution) and independent binomial (BN) sampling. The initial number of cells are $S = 8000$ and $F = 2000$ for the left panel, and $S = 20$ and $F = 5$ for the right panel. 10,000 samples are drawn for each distribution. Here, a parent collective is divided into 10 collectives.

## Simulation result

*Appendix 1—figure 3* presents the composition trajectories of all collectives using the tau-leaping algorithm in the maturation step. The selected adults have the closest composition to the target composition $\hat{f}$. The selected Adult can have smaller F frequency than its parent Adult, so F frequency can be lowered after cycles.

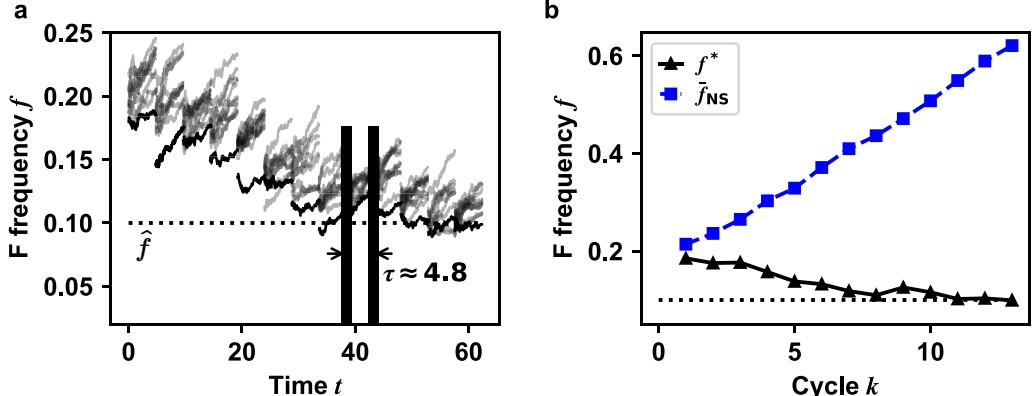

**Appendix 1—figure 3.** Trajectories of F frequency for 10 collectives ($g = 10$) over time. (**a**) The collective whose frequency is closest to the target value is selected in every cycle (black lines). The gray lines denote the other collectives. For parameters, we used S growth rate $r_S = 0.5$, F growth advantage $\omega = 0.03$, mutation rate $\mu = 0.0001$, maturation time $\tau \approx 4.8$, and $N_0 = 1000$. (**b**) Comparison between frequency trajectories with selection (the chosen one Adult producing all offspring; black) and without selection (each Adult producing one offspring; blue) clearly shows the effect of artificial selection. The black line indicates F frequency of the selected collective $f_k^*$ at each cycle in (**a**). The blue line indicates the average trajectory without selection $\bar{f}_{k+1}^*$ (the average of $g = 10$ individual lineages without inter-collective selection at the end of each cycle).

In *Appendix 1—figure 4*, we plot the absolute error $d$ between the target frequency $\hat{f}$ and $\langle f^* \rangle$ (i.e. $d = |\langle f^* \rangle - \hat{f}|$) at the end of simulations (1000 cycles). Since the computing time for the Tau-leaping algorithm (individual-based simulation) to reach 1000 cycles is very long, we used the sampling scheme in the above subsection. In the colormap, errors higher than 0.15 are marked with

gray, which indicates selection failure. The dashed lines indicate the same boundary in *Figure 2e* in the main text.

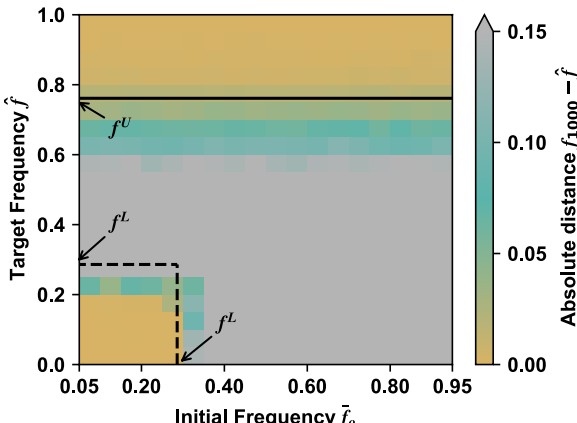

**Appendix 1—figure 4.** Color map of the absolute error $d = |\langle f^* \rangle - \hat{f}|$ averaged selected collectives at the end of simulations ($k = 1000$) and the target frequency $\hat{f}$. The solid and dashed lines are drawn by the arguments in the main text. For parameters, we used $r_S = 0.5$, $\omega = 0.03$, $\mu = 0.0001$, $N_0 = 1000$, $g = 10$ and $\tau \approx 4.8$. The result is the average of 300 independent simulations. Compared to *Figure 2e*, this figure has a higher resolution.

# Appendix 2

## Conditional probability distribution of the selected collective frequency $f^*$ and selection progress $\Delta f$

In the main text, we identify the region of success by using selection progress $\Delta f = \text{Median}[\Psi(f^*_{k+1}|f^*_k)] - f^*_k$, which is obtained from the conditional pdf of $f^*_{k+1}$ (the F frequency of the selected Adult at cycle $k+1$) given the selected $f^*_k$ at cycle $k$, written as $\Psi(f^*_{k+1} = f|f^*_k)$. We consider the challenging case where $f^*_k$ is above the target value ($f^*_k > \hat{f}$), and therefore the Adult with minimum F frequency will be selected. To get an analytical expression of $\Psi(f^*_{k+1} = f|f^*_k)$, we first find the conditional pdf of $f$ of Adults in cycle $k+1$ given $f^*_k$ at cycle $k$. Then, we find $\Psi(f^*_{k+1}|f^*_k)$ from the minimum value distribution of F frequencies among $G_S(\tau)$ Adults. Below, we describe the mathematical details of this process.

Let us start from the reproduction step from the selected Adult in cycle $k$. We reproduce $g$ Newborns in the next cycle $k+1$. Then the probability distribution of the F cell numbers in Newborn collectives is given in *Equation 36*. If the total number of cells in a Newborn collective $N_0$ is large enough, *Equation 36* is approximated by the Gaussian distribution $F^{(i)}_{k+1,0} \sim \mathcal{N}(N_0 f^*_k, N_0 f^*_k(1-f^*_k))$. Then, the probability density function that $f^{(i)}_{k+1,0}$ to be $\zeta$ in Newborn collective $i$ is

$$P_{f^{(i)}_{k+1,0}}(\zeta|f^*_k) = \frac{\sqrt{N_0}}{\sqrt{2\pi f^*_k(1-f^*_k)}} e^{-\frac{N_0(\zeta-f^*_k)^2}{2f^*_k(1-f^*_k)}}. \tag{37}$$

The Newborn collective $i$ has initial cell numbers $S^{(i)}_{k+1,0} = N_0(1-\zeta)$ and $F^{(i)}_{k+1,0} = N_0\zeta$. From here, we ignore cycle index $k+1$ in subscript and $i$ superscript for convenience.

Next, we write the conditional pdf of Adults' F frequency with given Newborn F frequency $\zeta$. We assume that cell numbers in Adult $S(\tau)$ and $F(\tau)$ follow Gaussian distributions as in *Equations 13; 14*. Based on *Equations 18; 19; 28; 33*, we have

$$S(\tau) \approx \overline{S}(\tau) + \sigma_S(\tau)G_S(\tau), \tag{38}$$

$$F(\tau) \approx \overline{F}(\tau) + \sigma_F(\tau)G_F(\tau), \tag{39}$$

where $G_S(\tau)$ and $G_F(\tau)$ are random variables following the standard distribution $\mathcal{N}(0,1)$. Note that each Gaussian is sharp if Newborn size $N_0$ is sufficiently large ($\overline{S}(\tau) \gg \sigma_S(\tau)$ and $\overline{F}(\tau) \gg \sigma_F(\tau)$). Then, we can approximately write $f(\tau)$ as

$$f(\tau) \approx \frac{\overline{F}(\tau) + \sigma_F(\tau)G_F(\tau)}{\overline{S}(\tau) + \sigma_S(t)G_S(\tau) + \overline{F}(\tau) + \sigma_F(\tau)G_F(\tau)}. \tag{40}$$
$$\approx \bar{f}(\tau) + \sigma_f(t)G_f(\tau)$$

The mean of $f$ is given by

$$\bar{f}(\tau) = \frac{\overline{F}(\tau)}{\overline{S}(\tau) + \overline{F}(\tau)} = \frac{\zeta W_\tau + \frac{\mu}{\omega}(1-\zeta)(W_\tau - 1)}{1 - \zeta + \zeta W_\tau + \frac{\mu}{\omega}(1-\zeta)(W_\tau - 1)}, \tag{41}$$

and the variance is

$$\begin{aligned}
\sigma_f^2(\tau) &= \frac{(1-\bar{f}(\tau))^2\sigma_F^2(\tau) + \bar{f}(\tau)^2\sigma_S^2(\tau)}{\overline{N}(\tau)^2} \\
&= \frac{1}{N_0 R_\tau \left((1-\zeta) + \zeta W_\tau + \frac{\mu}{\omega}(1-\zeta)(W_\tau - 1)\right)^4} \times \\
&\quad \left[(1-\zeta)^2 \left(\zeta W_\tau(R_\tau W_\tau - 1) + \frac{\mu}{\omega}(1-\zeta)\left(\frac{2\omega/r_S}{1+2\omega/r_S}R_\tau W_\tau^2 - W_\tau + \frac{1}{1+2\omega/r_S}\right)\right) \right. \\
&\quad \left. + \left(\zeta W_\tau + \frac{\mu}{\omega}(1-\zeta)(W_\tau - 1)\right)^2(1-\zeta)(R_\tau - 1)\right].
\end{aligned} \tag{42}$$

where $R_\tau = e^{r_S \tau}$ and $W_\tau = e^{\omega \tau}$. The average Adult size is $\overline{N}(\tau) = \overline{S}(\tau) + \overline{F}(\tau)$. Thus, the Adult's F frequency $f(\tau) = f_\tau$ follows the Gaussian distribution $\mathcal{N}(\bar{f}_\tau, \sigma_{f_\tau}^2)$ whose pdf is given by

$$P_{f_\tau}(f|\zeta) = \frac{1}{\sqrt{2\pi}\sigma_{f_\tau}} e^{-\frac{(f - \overline{f_\tau})^2}{2\sigma_{f_\tau}^2}}. \tag{43}$$

Next, we get the conditional pdf of $f_{k+1,\tau}$ (offspring Adult's F frequency in cycle $k+1$) given $f_k^*$. We multiply **Equations 37 and 43** and take the integral over $\zeta$:

$$P_{f_{k+1,\tau}}(f|f_k^*) = \int_0^1 d\zeta\, P_{f_{k+1,\tau}}(f|\zeta)\, P_{f_{k+1,0}}(\zeta|f_k^*). \tag{44}$$

After maturation in cycle $k+1$, the Adult with the smallest frequency is selected among $g$ Adult collectives, denoted as $f_{k+1}^{\min}$. The pdf of $f_{k+1}^{\min}$ is obtained by the theory of extreme value statistics (**Gumbel, 1958**). The cumulative distribution function (cdf) of the minimum value $f_{k+1}^{\min}$ is given by

$$\begin{aligned}
C_{f_{k+1}^{\min}}(f|f_k^*) &= \mathrm{Prob}\left[f_{k+1}^{\min} \leq f|f_k^*\right] \\
&= 1 - \mathrm{Prob}\left[f_{k+1,\tau}^{\min} \geq f|f_k^*\right] \\
&= 1 - \mathrm{Prob}\left[(f_{k+1,\tau}^{(1)} \geq f) \wedge (f_{k+1,\tau}^{(2)} \geq f) \wedge \cdots \wedge (f_{k+1,\tau}^{(g)} \geq f)|f_k^*\right].
\end{aligned} \tag{45}$$

Since frequencies are independent and identically distributed, $C_{f_{k+1}^{\min}}(f|f_k^*) = 1 - [\mathrm{Prob}(f_{k+1,\tau} \geq f)|f_k^*]^g$. Note that $\mathrm{Prob}[f_{k+1,\tau} \geq f|f_k^*] = \int_f^1 df'\, P_{f_{k+1,\tau}}(f'|f_k^*) = 1 - \int_0^f df'\, P_{f_{k+1,\tau}}(f'|f_k^*) = 1 - C_{f_{k+1,\tau}}(f|f_k^*)$, and **Equation 45** becomes

$$C_{f_{k+1}^{\min}}(f|f_k^*) = 1 - \left(1 - C_{f_{k+1,\tau}}(f|f_k^*)\right)^g. \tag{46}$$

Then, the probability density function $\Psi(f_{k+1}^*|f_k^*)$ is obtained by differentiating **Equation 46** with respect to $f$ and replacing $f \longrightarrow f_{k+1}^*$,

$$\Psi(f_{k+1}^*|f_k^*) = g \left(1 - C_{f_{k+1,\tau}}(f_{k+1}^*|f_k^*)\right)^{g-1} P_{f_{k+1,\tau}}(f_{k+1}^*|f_k^*). \tag{47}$$

We compute the probability density function **Equation 47** by using numerical integration and compare it with the stochastic simulation results in **Appendix 2—figure 1**. The two distributions are similar.

To get the analytic approximation of the median of **Equation 47**, we assume that the Adult's F frequency distribution is Gaussian. Then we only need to calculate the mean $\bar{f}(\tau)$ and variance $\sigma_f^2(\tau)$ of Adult's F frequency. Instead of calculating the integral with respect to $\zeta$ in **Equation 44**, we put a set of initial values from Newborn's F frequency distribution $\mathcal{N}(f_k^*, f_k^*(1 - f_k^*)/N_0)$ in **Equations 20, 21, 28 and 34**: $\overline{S}_0 = N_0(1 - f_k^*)$, $\overline{F}_0 = N_0 f_k^*$, $\sigma_{S,0}^2 = N_0 f_k^*(1 - f_k^*)$, $\sigma_{F,0}^2 = N_0 f_k^*(1 - f_k^*)$, and $\overline{SF}_0 - \overline{S}_0\overline{F}_0 = -N_0 f_k^*(1 - f_k^*)$. Then we have

$$\bar{f}(\tau) = \frac{\overline{F}(\tau)}{\overline{S}(\tau) + \overline{F}(\tau)} = \frac{f_k^* W_\tau + \frac{\mu}{\omega}(1 - f_k^*)(W_\tau - 1)}{1 - f_k^* + f_k^* W_\tau + \frac{\mu}{\omega}(1 - f_k^*)(W_\tau - 1)}, \tag{48}$$

$$\sigma_f^2(\tau) = \frac{(1 - f_k^*)}{N_0 R_\tau \left(1 - f_k^* + f_k^* W_\tau + \frac{\mu}{\omega}(1 - f_k^*)(W_\tau - 1)\right)^4} \times$$

$$\left[ f_k^* W_\tau \left\{ (2 - 2f_k^* + 2f_k^{*2}) R_\tau W_\tau - (1 - f_k^*) - f_k^* W_\tau \right\} \right.$$

$$+ \frac{\mu}{\omega}(1 - f_k^*) \left\{ (1 - f_k^*) \left( \left( \frac{2\omega}{r_S + 2\omega} - 2f_k^* \right) R_\tau W_\tau^2 - W_t + \frac{r_S}{r_S + 2\omega} + 2f_k^* R_\tau W_\tau \right) \right.$$

$$\left. + 2f_k^* \left( (1 + f_k^*) R_\tau - 1 \right) W_\tau (W_\tau - 1) \right\} + O(\mu^2) \bigg], \tag{49}$$

which give rise to **Equation 3** and **Equation 4** in the main text, respectively. The functional form of **Equations 48; 49** are plotted in **Appendix 2—figure 2a**.

The median $(\mathrm{Median}[\Psi(f_{k+1}^*|f_k^*)] \equiv \tilde{f})$ of **Equation 47** satisfies $C_{f_{k+1}}^{\min}(\tilde{f}|f_k^*) = \frac{1}{2}$, which means $\tilde{f} = C_{f_{k+1},\tau}^{-1}\left(\frac{\ln 2}{g}\right)$. If we assume that the distribution **Equation 47** is Gaussian, then the inverse function $C_{f_{k+1},\tau}^{-1}\left(\frac{\ln 2}{g}\right)$ can be written as

$$\mathrm{Median}[\Psi(f_{k+1}^*|f_k^*)] = C_{f_{k+1},\tau}^{-1}\left(\frac{\ln 2}{g}\right) = \bar{f}(\tau) + \left[\Phi^{-1}\left(\frac{\ln 2}{g}\right)\right]\sigma_f(\tau), \tag{50}$$

where $\Phi^{-1}(y)$ is an inverse cumulative density function (CDF) of the normal distribution with mean $i$ in **Equation 49** and standard deviation $\sigma_f(\tau)$, a square root of **Equation 49**. Subtracting $f_k^*$ from **Equation 50** gives the selection progress

$$\Delta f = \mathrm{Median}[\Psi(f_{k+1}^*|f_k^*)] - f_k^* = \bar{f}(\tau) + \left[\Phi^{-1}\left(\frac{\ln 2}{g}\right)\right]\sigma_f(\tau) - f_k^* \tag{51}$$

which is **Equation 2** in the main text.

Furthermore, we get an asymptotic expression of $\Phi^{-1}(\ln 2/g)$ when $g$ is large (or $\Phi^{-1}(y)$ with small $y$). Here, we introduce a method from **Phllip, 1960**. We start from the CDF of the standard normal distribution, $\Phi(x) = \mathrm{erfc}(-x/\sqrt{2})/2$ where the function $x$ is the complementary error function. To get the expression of $\Phi^{-1}$, we need an asymptotic expression of the inverse of $y = \mathrm{erfc}(x)$ function $(x = \mathrm{erfc}^{-1}(y))$ as the inverse CDF $\Phi^{-1}(y) = -\sqrt{2}\,\mathrm{erfc}^{-1}(2y)$. The known asymptotic expansion of $y = \mathrm{erfc}(x)$ for large $\Phi^{-1}(\ln 2/g)$ is $\mathrm{erfc}(x) \approx e^{-x^2}/x\sqrt{\pi}$. By taking the logarithm of both sides, we have

$$x^2 \approx -\ln y - \frac{1}{2}\ln \pi x^2. \tag{52}$$

Replacing $x^2$ on the right-hand side in **Equation 52** into the expression itself, we get a continued logarithmic form of

$$x^2 \approx -\ln y - \frac{1}{2}\ln \pi \left(-\ln y - \frac{1}{2}\ln \pi \left(-\ln y - \frac{1}{2}\ln \cdots\right)\right). \tag{53}$$

Inserting $x = \mathrm{erfc}^{-1}(y)$ (square root of **Equation 53**) into the inverse CDF $\Phi^{-1}(y) = -\sqrt{2}\,\mathrm{erfc}^{-1}(2y)$, we have $\Phi^{-1}(y) \approx -\sqrt{-2\ln 2y - \ln\pi(-\ln 2y - \cdots)}$. So, the asymptotic expression of $\mathcal{N}(f_k^*, f_k^*(1 - f_k^*)/N_0)$ is given by

$$\Phi^{-1}\left(\frac{\ln 2}{g}\right) \approx -\sqrt{2\ln g - \ln[\ln g] + \cdots}. \tag{54}$$

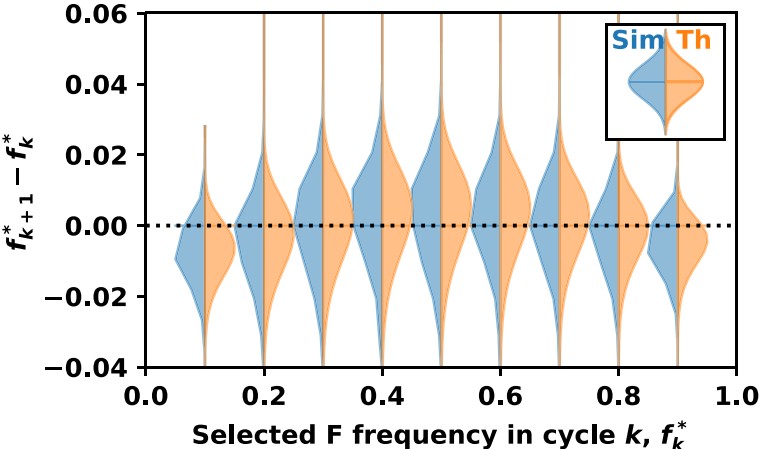

**Appendix 2—figure 1.** The probability density functions of the selected Adult's F frequency $f_{k+1}^*$ subtracted by $f_k^*$. For simulations (blue), at each $f_k^*$, we performed 1000 stochastic simulations. The orange distribution represents *Equation 47* computed by numerical integration. The median values of the distributions are shown in *Figure 3a* in the main text.

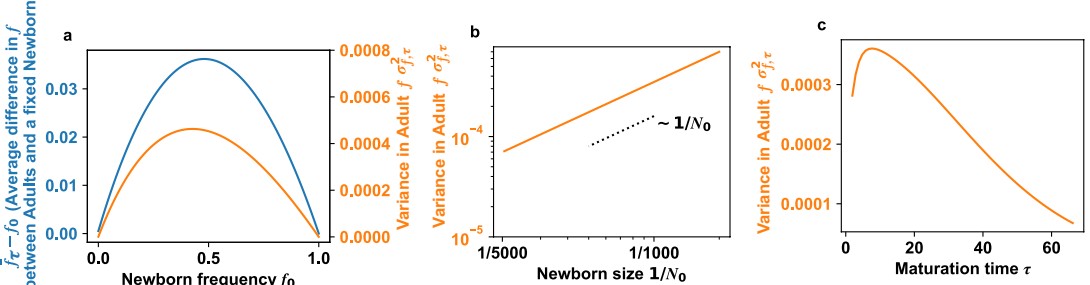

**Appendix 2—figure 2.** Effect of experimental parameters in the distribuiton of Adult's F frequency. (**a**) Mean (*Equation 41*) and variance (*Equation 42*) of $f$ values of Adult collectives with respect to the Newborn frequency $f_0$. (**b**) Scaling relation of F frequency variance (*Equation 49*) with Newborn collective size $N_0$. The initial F frequency is 0.5. The parameters are $r_S = 0.5$, $\omega = 0.03$, $\mu = 0.0001$, and $\tau \approx 4.8$. (**c**) Relation of F frequency variance (*Equation 49*) with maturation time $\tau$. Other parameters are the same as b.

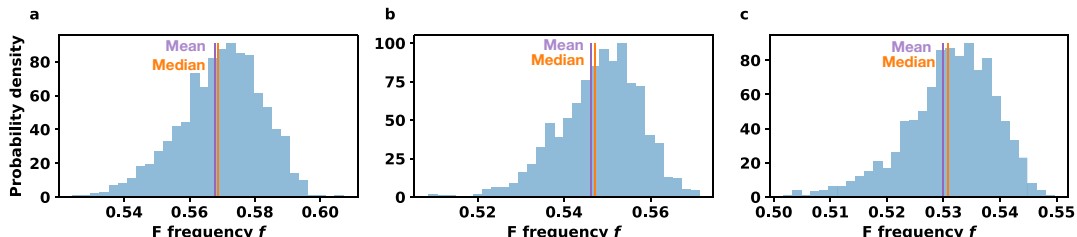

**Appendix 2—figure 3.** Median (orange) and mean (violet) have similar distributions. We performed 1000 simulations to get probability density. (**a**) $g = 10$, (**b**) $g = 100$, and (**c**) $g = 1000$. Initial F frequency is $f_k^* = 0.5$. The parameters are $r_S = 0.5$, $\omega = 0.03$, $\mu = 0.0001$ and $\tau = \ln[1000]/r_S$.

## Appendix 3

### Critical newborn size $\check{N}_0$ to allow all target frequencies

First, we note that $\sigma_f^2$ in *Equation 49* is proportional to $N_0^{-1}$ for the following reasons. Variance $\sigma_S^2(t)$ in *Equation 28* scales linearly with $N_0$ since both $\overline{S}_0$ and $\sigma_{S,0}^2$ scale linearly with $N_0$. Variance $\sigma_F^2(t)$ in *Equation 33* also scales linearly with $N_0$ because $\sigma_{F,0}^2$, $\overline{F}_0$, $\overline{S}_0$, and covariance $\overline{SF}_0 - \overline{S}_0\overline{F}_0$ all scale linearly with $N_0$. The mean adult size $\overline{N}(\tau) = \overline{S}(\tau) + \overline{F}(\tau)$ is also proportional to $N_0$ because the average cell numbers in *Equations 18; 19* are linear with respect to $N_0$. Thus, the scaling relation of *Equation 49* is given by $\sigma_f^2(\tau) = [(1 - \bar{f}(\tau))^2\sigma_F^2(\tau) + \bar{f}(\tau)^2\sigma_S^2(\tau)]/\overline{N}(t)^2 \sim N_0/N_0^2 \sim 1/N_0$.

Small $N_0$ makes all target frequencies achievable, as shown in *Figure 4a* in the main text. That is because small $N_0$ induces large $\sigma_f$, and thus $N_0$ smaller than a certain critical value $\check{N}_0$ makes the selection progress $\Delta f$ always negative, regardless of the value of $f_k^*$ (i.e. $\Delta f = \bar{f}(\tau) + \left[\Phi^{-1}(\frac{\ln 2}{g})\right]\sigma_f(\tau) - f_k^* < 0$). That means the inter-collective selection overcomes intra-collective selection in any target frequencies. To get an analytical approximation of the critical newborn size $\check{N}_0$, we simply assume that selection progress $\Delta f$ is maximum at $f_k^* = 1 - f_k^* = \frac{1}{2}$ where the changes in $\bar{f}$ and $\sigma_f^2$ are fastest. If the maximum value of $\Delta f$ is zero, all other values of *Equation 50* are negative, which naturally states that all targets are achievable. Putting $f_k^* = \frac{1}{2}$, *Equations 48; 49* become

$$\bar{f}(\tau)\Big|_{f_k^*=\frac{1}{2}} = \frac{\overline{F}(\tau)}{\overline{S}(\tau) + \overline{F}(\tau)} = \frac{W_\tau + \frac{\mu}{\omega}(W_\tau - 1)}{1 + W_\tau + \frac{\mu}{\omega}(W_\tau - 1)}, \tag{55}$$

and

$$\sigma_f^2(\tau)\Big|_{f_k^*=\frac{1}{2}} \approx \frac{2}{N_0 R_\tau \left(1 + W_\tau + \frac{\mu}{\omega}(W_\tau - 1)\right)^4} \times$$

$$\left[W_\tau\left(3R_t W_\tau - 1 - W_\tau\right)\right. \tag{56}$$

$$\left. + \frac{\mu}{\omega}\left\{\left(\left(\frac{2\omega}{r_S + 2\omega} - 1\right)R_t W_\tau^2 - W_t + \frac{r_S}{r_S + 2\omega} + R_t W_\tau\right) + (3R_t - 2)W_\tau(W_\tau - 1)\right\}\right]$$

So, by setting $\Delta f|_{f_k^*=\frac{1}{2}} = \bar{f}(\tau)|_{f_k^*=\frac{1}{2}} + \left[\Phi^{-1}(\frac{\ln 2}{g})\right]\sigma_f(\tau)|_{f_k^*=\frac{1}{2}} - \frac{1}{2} = 0$ with $N_0 = \check{N}_0$, we get a solution of

$$\check{N}_0 = \left[\Phi^{-1}\left(\frac{\ln 2}{g}\right)\right]^2 \frac{8}{R_\tau \left[1 + W_\tau + \frac{\mu}{\omega}(W_\tau - 1)\right]^2 \left[1 - W_\tau - \frac{\mu}{\omega}(W_\tau - 1)\right]^2} \times$$

$$\left[W_\tau\left(3R_\tau W_\tau - 1 - W_\tau\right)\right. \tag{57}$$

$$\left. + \frac{\mu}{\omega}\left\{\left(\left(\frac{2\omega}{r_S + 2\omega} - 1\right)R_\tau W_\tau^2 - W_\tau + \frac{r_S}{r_S + 2\omega} + R_\tau W_\tau\right) + (3R_\tau - 2)W_\tau(W_\tau - 1)\right\}\right]$$

Thus, all target frequencies are successfully selected with Newborn size $N_0$ smaller than $\check{N}_0$. If the mutation rate is zero, the critical value becomes

$$\check{N}_0 = \left[\Phi^{-1}\left(\frac{\ln 2}{g}\right)\right]^2 \frac{8W_\tau\left(3R_\tau W_\tau - W_\tau - 1\right)}{R_\tau\left(W_\tau^2 - 1\right)^2}. \tag{58}$$

## Appendix 4

### Selection without mutation $\mu = 0$

When the mutation rate is zero, two genotypes behave as two distinct species. The compositional change is provided by *Equation 50* with setting $\mu = 0$. Corresponding $\bar{f}$ in *Equation 48* and $\sigma_f^2$ in *Equation 49* become

$$\bar{f}(\tau) = \frac{f_k^* W_\tau}{1 - f_k^* + f_k^* W_\tau}, \tag{59}$$

$$\sigma_f^2(\tau) = \frac{f_k^*(1 - f_k^*)W_\tau \left[(2 - 2f_k^* + 2f_k^{*2})R_\tau W_\tau - (1 - f_k^*) - f_k^* W_\tau\right]}{N_0 R_\tau \left(1 - f_k^* + f_k^* W_\tau\right)^4}. \tag{60}$$

*Equations 59; 60* suggest that when a community consists of two competing species, we obtain similar conclusions on the accessible region for target composition. The stochastic simulation results are presented in *Appendix 4—figure 1*.

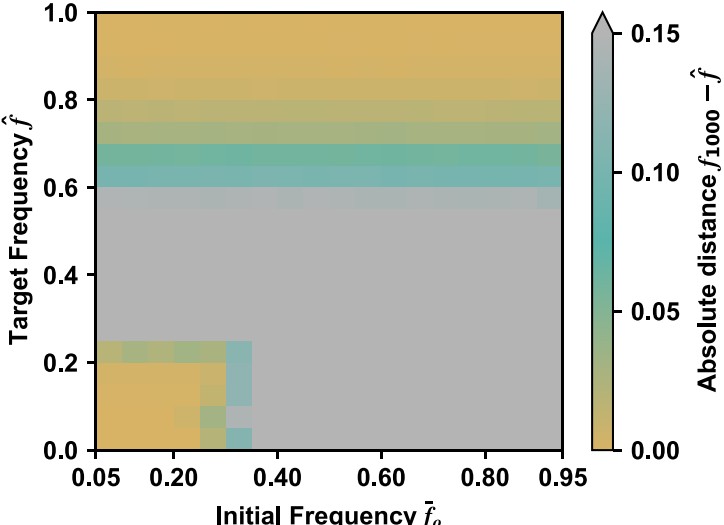

**Appendix 4—figure 1.** Simulation with zero mutation rate. Color map of the absolute error $d = |\langle f^* \rangle - \hat{f}|$ between frequency $\langle f^* \rangle$ of the averaged selected collectives at the end of simulations ($k = 1000$) and the target frequency $\hat{f}$. For parameters, we used $r_S = 0.5$, $\omega = 0.03$, $\mu = 0$, $N_0 = 1000$, $g = 10$, and $\tau \approx 4.8$.

## Appendix 5

### Stronger or weaker advantages $\omega$

The solution of *Equation (2)* in main text provides the boundary values with varying the $\omega$, the fitness advantage of F over S. We numerically calculate the solutions and plot in *Appendix 5—figure 1*.

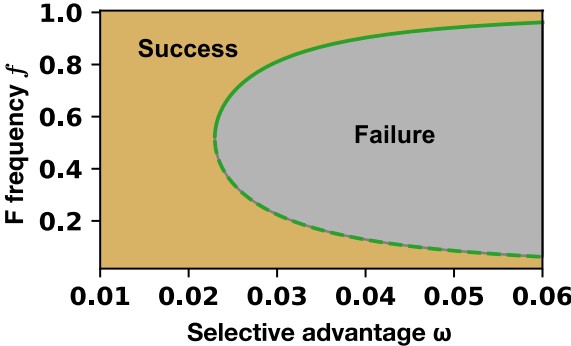

**Appendix 5—figure 1.** Change of success region in varying selective advantage $\omega.r_s$, $\omega = 0.03$, $\mu = 0.0001$, $N_0 = 1000$, $g = 10$, and $\tau \approx 4.8$.

## Appendix 6

### Deleterious mutation $\omega < 0$

In the main text, we show that the target composition can be achieved in some ranges of initial and target values when the mutation is beneficial to growth. The same analogy can be applied when the mutation is deleterious. Since the F cells grow slower than the S cells ($\omega < 0$), the F frequency naturally decreases in the maturation step. Then, the challenging case is selecting a larger F frequency against the intra-collective selection. So the conditional probability distribution $\Psi(f^*_{k+1}|f^*_k)$ that we consider now is a maximum value distribution of *Equation 44*. Thus, instead of *Equation 45*, we look for the cumulative distribution function of the maximum value $f^{\max}_{k+1}$ such that

$$
\begin{aligned}
C_{f^{\max}_{k+1}}(f|f^*_k) &= \text{Prob}\left[f^{\max}_{k+1} \geq f|f^*_k\right] \\
&= \text{Prob}\left[(f^{(1)}_{k+1}(\tau) \geq f) \wedge (f^{(2)}_{k+1}(\tau) \geq f) \wedge \cdots \wedge (f^{(g)}_{k+1}(\tau) \geq f)|f^*_k\right].
\end{aligned}
\tag{61}
$$

If all frequencies are independent and identically distributed random variables, the cumulative distribution function becomes

$$
C_{f^{\max}_{k+1}}(f|f^*_k) = \left(C_{f_{k+1,\tau}}(f|f^*_k)\right)^g.
\tag{62}
$$

Likewise in the previous section, we get the conditional probability density function by differentiating *Equation 62* with respect to $f$ and replacing $f \to f^*_{k+1}$ as

$$
\Psi(f^*_{k+1}|f^*_k) = g\left(C_{f_{k+1,\tau}}(f^*_{k+1}|f^*_k)\right)^{g-1} P_{f_{k+1,\tau}}(f^*_{k+1}|f^*_k).
\tag{63}
$$

The distribution in *Equation 63* is evaluated for various $f^*_k$ in *Appendix 6—figure 1a* with numerical simulations, and the median values of distributions are presented in *Appendix 6—figure 1b*. In the case of $\omega = -0.03$, the target frequency is lower than around 0.3 and larger than around 0.7 can be selected. Since the sign of $\omega$ is opposite to the result in the main text, the diagram is reversed from *Figure 2e* in the main text.

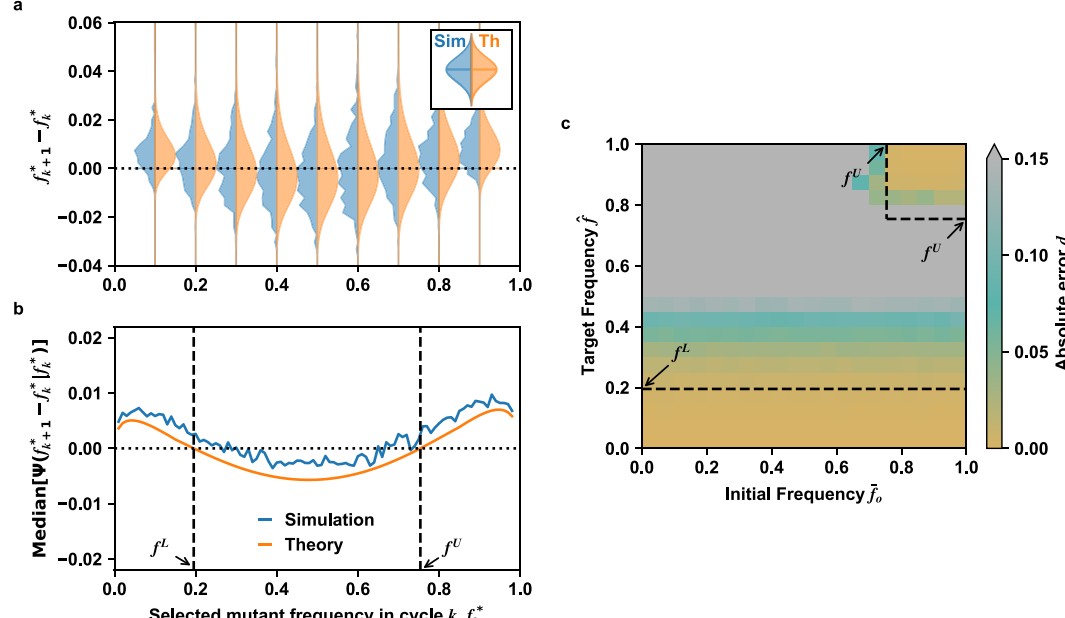

**Appendix 6—figure 1.** Artificial selection also works for deleterious mutation. (**a**) Conditional probability density functions of $f^*_{k+1} - f^*_k$ for various $f^*_k$ values. The left-hand side distribution is obtained from simulations and the right-hand side distribution is numerically obtained by evaluating *Equation 63*. Small triangles inside indicate the median values of the distributions. (**b**) The median value of distributions at a given $f^*_k$. The points where the shifted median becomes zero, $\text{Median}[\Psi(f^*_{k+1} - f^*_k|f^*_k)] = 0$ are denoted as $f^L$ and $f^U$, respectively. (**c**) The relative error between the target frequency $\hat{f}$ and the ensemble averaged selected frequency $\langle f^*_k \rangle$ is measured after 1000 cycles

starting from the initial frequency $\bar{f}_{1,0}$. Either the lower target frequencies or the higher target frequencies starting from the high initial frequencies can be achieved. The black dashed lines indicate the predicted boundary values $f^U$ and $f^L$ in **a**.

# Appendix 7

## Selecting more than one collective

In the main text, we choose one collective which has the closest frequency to the target among $g$ collectives. Such a 'top 1' strategy allows us to apply extreme value theory. However, 'top 1' may be too restrictive (*Xie et al., 2019*). Thus, we test the 'top-tier' strategy by choosing the top five among 100 Adults (*Appendix 7—figure 1*). The top-tier strategy is shown to be inefficient in our system. This is because in *Xie et al., 2019*, nonheritable variations – such as stochastic fluctuations in species composition introduced by pipetting – caused nonheritable variations in collective function. Nonheritable variations could potentially mask desired mutations if these mutations happened to occur in an 'unlucky' environment that yielded lower collective functions. Hence, lenient selection would allow the preservation of these mutations. In contrast here, stochastic fluctuations in genotype composition are heritable: a parent Newborn with lower F frequency $f$ will tend to have offspring Newborns with lower $f$ values. Hence, top-1 is more effective in this study.

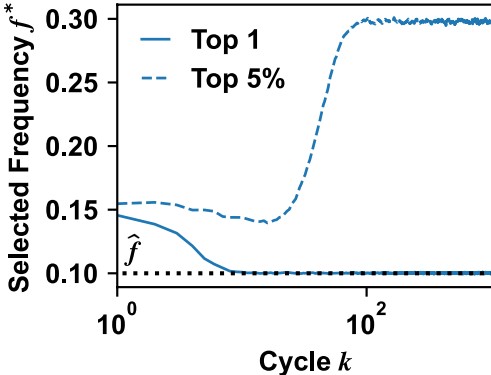

**Appendix 7—figure 1.** Selecting top 5% outperforms selecting top 1. We bred 100 collectives and chose either top-1 collective (solid line) or top-5 collectives (dashed line) with $f$ closest to the target value $\widehat{f}$ (black dotted line).

## Appendix 8

### Extension to three-population system

We assume that collectives consist of three genotypes with slow-growing (S), fast-growing (F), and faster-growing (FF) types. The growth rate of S is $r_S$. Each mutation adds $\omega$ to the growth rate. Thus, the F and FF types have growth rates $r_S + \omega$ and $r_S + 2\omega$, respectively. The mutation rate is $\mu$. So, the birth and mutation events are written by the chemical reactions:

$$S \xrightarrow{r_S} S + S, \tag{64}$$

$$F \xrightarrow[\omega > 0]{r_S + \omega} F + F, \tag{65}$$

$$FF \xrightarrow[\omega > 0]{r_S + 2\omega} FF + FF, \tag{66}$$

$$S \xrightarrow{\mu} F, \tag{67}$$

$$F \xrightarrow{\mu} FF \tag{68}$$

We write a master equation of the processes for $P(S, F, FF, t)$ which is the probability to have $S$, $F$, and $FF$ numbers of S, F, and FF cells at time $t$, respectively.

$$
\begin{aligned}
\frac{dP(S, F, FF, t)}{dt} &= r_S(S - 1)P(S - 1, F, FF, t) - r_S S P(S, F, FF, t) \\
&\quad + (r_S + \omega)(F - 1)P(S, F - 1, FF, t) - (r_S + \omega)F P(S, F, FF, t) \\
&\quad + (r_S + 2\omega)(FF - 1)P(S, F, FF - 1, t) - (r_S + 2\omega)FF P(S, F, FF, t) \\
&\quad + \mu(S + 1)P(S + 1, F - 1, FF, t) - \mu S P(S, F, FF, t) \\
&\quad + \mu(F + 1)P(S, F + 1, FF - 1, t) - \mu F P(S, F, FF, t).
\end{aligned}
\tag{69}
$$

The composition of collective $i$ in cycle $k$ is now represented with two frequencies $(f_k^{(i)}(t), h_k^{(i)}(t)) \equiv \mathbf{p}_k^{(i)}(t)$ where the F frequency is $f_k^{(i)}(t) = F_k^{(i)}(t)/(S_k^{(i)}(t) + F_k^{(i)}(t) + FF_k^{(i)}(t))$ and the FF frequency is $h_k^{(i)}(t) = FF_k^{(i)}(t)/(S_k^{(i)}(t) + F_k^{(i)}(t) + FF_k^{(i)}(t))$. Then, the target composition is set to be $(\hat{f}, \hat{h})$. The composition of the selected Adult in cycle $k$ is $(f_k^*, h_k^*) \equiv \mathbf{p}_k^*$. We apply the processes used in the above Appendix 2 to obtain the conditional probability $\Psi(\mathbf{p}_{k+1}^* | \mathbf{p}_k^*)$ by using the master *Equation 69*.

At the reproduction step in cycle $k$, we choose $N_0$ cells from the selected Adult whose composition is $(f_k^*, h_k^*) \equiv \mathbf{p}_k^*$. Then, newborn collectives are independently sampled from a multinomial distribution. For convenience, we drop the collective index $(i)$. Then, the conditional joint probability mass function of $F_{k+1,0}, FF_{k+1,0}$ cells is represented by

$$
\begin{aligned}
P(F_{k+1,0}, FF_{k+1,0} | f_k^*, h_k^*) &= \frac{N_0!}{(N_0 - F_{k+1,0} - FF_{k+1,0})! F_{k+1,0}! FF_{k+1,0}!} \times \\
&\quad (1 - f_k^* - h_k^*)^{N_0 - F_{k+1,0} - FF_{k+1,0}} (f_k^*)^{F_{k+1,0}} (h_k^*)^{FF_{k+1,0}},
\end{aligned}
\tag{70}
$$

where the number of S $S_{k+1,0}$ is automatically set to be $S_{k+1,0} = N_0 - F_{k+1,0} - FF_{k+1,0}$. Then, the approximated multivariate normal distribution is $\mathcal{N}(N_0 \mathbf{p}_k^*, N_0 \mathbf{M}_{k+1})$ where the mean distribution is $\mathbf{p}_k^* = (f_k^*, h_k^*)$ and covariance matrix is $\mathbf{M}_{k+1}$. The diagonal terms of $\mathbf{M}_{k+1}$ are variances $\sigma_X^2 = \overline{X^2} - [\overline{X}]^2$ and the off-diagonal terms are covariances $\sigma_{XY} = \overline{XY} - \overline{X}\,\overline{Y}$. The matrix is given by

$$
\mathbf{M}_{k+1} = 
\begin{bmatrix}
\dfrac{f_k^*(1 - f_k^*)}{N_0^2} & -\dfrac{f_k^* h_k^*}{N_0^2} \\[3mm]
-\dfrac{f_k^* h_k^*}{N_0^2} & \dfrac{h_k^*(1 - h_k^*)}{N_0^2}
\end{bmatrix}
\equiv
\begin{bmatrix}
\sigma_{f_{k+1,0}}^2 & \sigma_{f_{k+1,0} h_{k+1,0}} \\[2mm]
\sigma_{f_{k+1,0} h_{k+1,0}} & \sigma_{h_{k+1,0}}^2
\end{bmatrix}.
\tag{71}
$$

Then a Newborn's composition $\rho(\zeta, \eta)$ follows the multivariate Gaussian distribution $(\zeta, \eta) \sim \mathcal{N}(\mathbf{p}_k^*, \mathbf{M}_{k+1})$ whose joint probability distribution is given by

$$P_{\boldsymbol{\rho}_{k+1,0}}(\boldsymbol{\rho}|\mathbf{p}_k^*) = \frac{1}{\sqrt{2\pi}^2\sqrt{\det \boldsymbol{M}_{k+1}}} e^{-\frac{1}{2}(\boldsymbol{\rho}-\boldsymbol{p}_k^*)\boldsymbol{M}_{k+1}^{-1}(\boldsymbol{\rho}-\boldsymbol{p}_k^*)^T}. \tag{72}$$

At the beginning of cycle $k$, a newborn collective starts from $(S_0, F_0, FF_0)$ cells (for convenience, cycle index $k$ is dropped.) In terms of $(\zeta, \eta)$, each initial numbers are $S_0 = N_0(1 - \zeta - \eta)$, $F_0 = N_0\zeta$, and $FF_0 = N_0\eta$. Their initial covariance matrix is $N_0^2\mathbf{M}_{k+1}$. By using **Equation 69**, we can write ordinary differential equations up to the second moment.

$$\frac{d\overline{S}(t)}{dt} = r_S\overline{S}(t) - \mu\overline{S}(t), \tag{73}$$

$$\frac{d\overline{F}(t)}{dt} = (r_S + \omega)\overline{F}(t) + \mu(\overline{S}(t) - \overline{F}(t)), \tag{74}$$

$$\frac{d\overline{FF}(t)}{dt} = (r_S + 2\omega)\overline{FF}(t) + \mu\overline{F}(t), \tag{75}$$

$$\frac{d\overline{S^2}}{dt} = 2(r_S - \mu)\overline{S^2} + (r + \mu)\overline{S}, \tag{76}$$

$$\frac{d\overline{F^2}}{dt} = 2(r_S + \omega - \mu)\overline{F^2} + (r_S + \omega + \mu)\overline{F} + \mu(2\overline{SF} + \overline{F}), \tag{77}$$

$$\frac{d\overline{FF^2}}{dt} = 2(r_S + 2\omega)\overline{FF^2} + (r_S + 2\omega)\overline{FF} + \mu(2\overline{FFF} + \overline{F}), \tag{78}$$

$$\frac{d\overline{SF}}{dt} = (2r_S + \omega - 2\mu)\overline{SF} + \mu(\overline{S^2} - \overline{S}), \tag{79}$$

$$\frac{d\overline{FFF}}{dt} = (2r_S + 3\omega - \mu)\overline{FFF} + \mu(\overline{SFF} + \overline{F^2} - \overline{F}), \tag{80}$$

$$\frac{d\overline{SFF}}{dt} = (2r_S + 2\omega - \mu)\overline{SFF} + \mu\overline{SF}. \tag{81}$$

The initial conditions of the system in coupled **Equations 73–81** are obtained by the mean and (co)variances of **Equation 70**. By solving equations numerically, we obtain a set of mean cell numbers $(\overline{S}, \overline{F}, \overline{FF})$ and a set of variances $(\sigma_S^2, \sigma_F^2, \sigma_{FF}^2)$ as well as covariances $(\sigma_{SF}, \sigma_{FFF}, \sigma_{SFF})$. We assume that the covariances are smaller than the variances. We consider $S$, $F$, and $FF$ as Gaussian random variables

$$S(t) \approx \overline{S}(t) + \sigma_S(t)G_S(t), \tag{82}$$

$$F(t) \approx \overline{F}(t) + \sigma_F(t)G_F(t), \tag{83}$$

$$FF(t) \approx \overline{FF}(t) + \sigma_{FF}(t)G_{FF}(t). \tag{84}$$

Then, the F frequency becomes

$$f(t) \approx \frac{\overline{F}(t) + \sigma_F(t)G_F(t)}{\overline{S}(t) + \sigma_S(t)G_S(t) + \overline{F}(t) + \sigma_F(t)G_F(t) + \overline{FF}(t) + \sigma_{FF}(t)G_{FF}(t)}$$
$$\approx \bar{f}(t) + \sigma_f(t)G_f(t), \tag{85}$$

where $\bar{f} = \overline{F}/(\overline{S} + \overline{F} + \overline{FF})$ and $\sigma_f^2 = \left(\bar{f}^2(\sigma_S^2 + \sigma_{FF}^2) + (1 - \bar{f})^2\sigma_F^2\right)/(\overline{S} + \overline{F} + \overline{FF})$. Similarly, the FF frequency is

$$h(t) \approx \frac{\overline{FF}(t) + \sigma_{FF}(t)G_{FF}(t)}{\overline{S}(t) + \sigma_S(t)G_S(t) + \overline{F}(t) + \sigma_F(t)G_F(t) + \overline{FF}(t) + \sigma_{FF}(t)G_{FF}(t)}$$
$$\approx \overline{h}(t) + \sigma_h(t)G_h(t), \tag{86}$$

where $\bar{h} = \overline{FF}/(\bar{S} + \bar{F} + \overline{FF})$ and $\sigma_h^2 = \left(\bar{h}^2(\sigma_S^2 + \sigma_F^2) + (1 - \bar{h}^2)\sigma_{FF}^2\right)/(\bar{S} + \bar{F} + \overline{FF})$. The dynamic flow of F and FF frequencies during maturation is shown in **Appendix 8—figure 1a**. If the covariances are small enough, we can approximate the joint probability distribution of Adult's composition $(f_\tau, h_\tau) = \mathbf{p}_\tau$ as

$$P_{\mathbf{p}_\tau}(\mathbf{p}|\rho) = \frac{1}{\sqrt{2\pi}\sigma_{f_\tau}} e^{-\frac{(f - \bar{f}_\tau)^2}{2\sigma_{f_\tau}^2}} \frac{1}{\sqrt{2\pi}\sigma_{h_\tau}} e^{-\frac{(h - \bar{h}_\tau)^2}{2\sigma_{h_\tau}^2}} . \tag{87}$$

With cycle index $k$, we get the conditional probability of matured collectives $P_{\mathbf{p}_{k+1,\tau}}(\mathbf{p}|\mathbf{p}_k^*)$ by

$$P_{\mathbf{p}_{k+1,\tau}}(\mathbf{p}|\mathbf{p}_k^*) = \int_0^1 d\zeta \int_0^1 d\eta P_{\mathbf{p}_{k+1,\tau}}(\mathbf{p}|\zeta, \eta) P_{\mathbf{p}_{k+1,0}}(\zeta, \eta|\mathbf{p}_k^*). \tag{88}$$

We select the Adult collective among $g$ Adult collectives such that the change in frequencies during maturation could be compensated. During maturation, a frequency distribution moves in different directions in $(f, h)$ space depending on the initial composition $(f_k^*, h_k^*)$ So, we take different directions to obtain the extreme value distributions. Considering only the sign of the frequency changes in $f$ and $h$, we take either maximum or minimum. The mean change in $h$ is always positive in the whole $(f, h)$ space since $d\overline{FF}/dt$ is always positive in **Equation 75**. Thus, we choose the minimum value $h_{k+1}^{min}$ in every selection step.

If the mean $\bar{f}_{k+1,\tau}^{(i)} = \int_0^1 df' \int_0^1 dh' f P_{\mathbf{p}_{k+1,\tau}}(f', h'|\mathbf{p}_k^*)$ is larger than $f_k^*$, the minimum value among $f_{k+1,\tau}^{(1)}, f_{k+1,\tau}^{(2)}, \cdots, f_{k+1,\tau}^{(g)}$ will be chosen in the selection step to compensate for the frequency change in the maturation step. Let us denote the selected valued of $f$ and $h$ as $f_{k+1}^* = \min(f_{k+1,\tau}^{(1)}, f_{k+1,\tau}^{(2)}, \cdots, f_{k+1,\tau}^{(g)})$ and $h_{k+1}^* = \min(h_{k+1,\tau}^{(1)}, h_{k+1,\tau}^{(2)}, \cdots, h_{k+1,\tau}^{(g)})$. We temporarily drop the time index $\tau$ for simplicity. Then, the joint cumulative distribution function $C_{\mathbf{p}_{k+1}^*}(f, h|\mathbf{p}_k^*) = \Pr[f_{k+1}^* < f \wedge h_{k+1}^* < h|\mathbf{p}_k^*]$ is

$$\begin{aligned}
C_{\mathbf{p}_{k+1}^*}(f, h|\mathbf{p}_k^*) &= \int_0^f df' \int_0^h dh' P_{\mathbf{p}_{k+1}^*}(f', h'|\mathbf{p}_k^*) \\
&= \left(\int_0^1 - \int_f^1\right) df' \left(\int_0^1 - \int_h^1\right) dh' P_{\mathbf{p}_{k+1}^*}(f', h'|\mathbf{p}_k^*) \\
&= \int_0^1 df' \int_0^1 dh' P_{\mathbf{p}_{k+1}^*}(f', h'|\mathbf{p}_k^*) - \int_h^1 df' \int_0^1 dh' P_{\mathbf{p}_{k+1}^*}(f', h'|\mathbf{p}_k^*) \\
&\quad - \int_0^1 df' \int_h^1 dh' P_{\mathbf{p}_{k+1}^*}(f', h'|\mathbf{p}_k^*) + \int_f^1 df' \int_h^1 dh' P_{\mathbf{p}_{k+1}^*}(f', h'|\mathbf{p}_k^*) \\
&= 1 - \Pr[f_{k+1}^* \geq f|\mathbf{p}_k^*] - \Pr[h_{k+1}^* \geq h|\mathbf{p}_k^*] + \Pr[f_{k+1}^* \geq f \wedge h_{k+1}^* \geq h|\mathbf{p}_k^*]
\end{aligned} \tag{89}$$

The probability $\Pr[f_{k+1}^* \geq f \wedge h_{k+1}^* \geq h|\mathbf{p}_k^*]$ can be converted as

$$\begin{aligned}
&\Pr[f_{k+1}^* \geq f \wedge h_{k+1}^* \geq h|\mathbf{p}_k^*] \\
&= \Pr[f_{k+1}^{(1)} \geq f \wedge h_{k+1}^{(1)} \geq h \wedge f_{k+1}^{(2)} \geq f \wedge h_{k+1}^{(2)} \geq h \wedge \cdots \wedge f_{k+1}^{(g)} \geq f \wedge h_{k+1}^{(g)} \geq h|\mathbf{p}_k^*] \\
&= [\Pr[f_{k+1} \geq f \wedge h_{k+1} \geq h|\mathbf{p}_k^*]]^g \\
&= [1 - \Pr[f_{k+1} < f|\mathbf{p}_k^*] - \Pr[h_{k+1} < h|\mathbf{p}_k^*] + \Pr[f_{k+1} < f \wedge h_{k+1} < h|\mathbf{p}_k^*]]^g \\
&= [1 - C_{f_{k+1}}(f|p_k^*) - C_{h_{k+1}}(h|p_k^*) + C_{\mathbf{p}_{k+1}}(f, h|p_k^*)]^g,
\end{aligned} \tag{90}$$

where $C_{\mathbf{p}_{k+1}}(f, h|\mathbf{p}_k^*) = \int_0^f df' \int_0^h dh' P_{\mathbf{p}_{k+1}}(f', h'|\mathbf{p}_k^*)$ is a conditional joint cumulative distribution function of $(f^{(i)}, h^{(i)})$. The marginal cumulative distribution functions are

$$C_{f_{k+1}}(f|\mathbf{p}_k^*) = \int_0^f df' \int_0^1 dh' P_{\mathbf{p}_{k+1}}(f', h'|\mathbf{p}_k^*), \tag{91}$$

$$C_{h_{k+1}}(h|\boldsymbol{p}_k^*) = \int_0^1 df' \int_0^h dh' P_{\mathbf{p}_{k+1}}(f', h'|\boldsymbol{p}_k^*). \tag{92}$$

Similarly, the probabilities $\Pr[f_{k+1}^* \geq f|\boldsymbol{p}_k^*]$ and $\Pr[h_{k+1}^* \geq h|\boldsymbol{p}_k^*]$ are converted into $\Pr[f_{k+1}^* \geq f|\boldsymbol{p}_k^*] = \left[1 - C_{f_{k+1}}(f|\boldsymbol{p}_k^*)\right]^g$ and $\Pr[h_{k+1}^* \geq h|\boldsymbol{p}_k^*] = \left[1 - C_{h_{k+1}}(h|\boldsymbol{p}_k^*)\right]^g$. Thus, the joint cumulative distribution function is

$$\begin{aligned}
C_{\mathbf{p}_{k+1}^*}(f, h|\boldsymbol{p}_k^*) = {} & 1 - \left[1 - C_{f_{k+1}}(f|\boldsymbol{p}_k^*)\right]^g - \left[1 - C_{h_{k+1}}(h|\boldsymbol{p}_k^*)\right]^g \\
& + \left[1 - C_{f_{k+1}}(f|\boldsymbol{p}_k^*) - C_{h_{k+1}}(h|\boldsymbol{p}_k^*) + C_{\mathbf{p}_{k+1}}(f, h|\boldsymbol{p}_k^*)\right]^g.
\end{aligned} \tag{93}$$

Then, the conditional probability of the selected collective is given by

$$\begin{aligned}
P_{\mathbf{p}_{k+1}^*}(f, h|\boldsymbol{p}_k^*) = {} & \frac{\partial^2}{\partial f \partial h} C_{\mathbf{p}_{k+1}^*}(f, h|\boldsymbol{p}_k^*) \\
= {} & g(g-1) \left[1 - C_{f_{k+1}}(f|\boldsymbol{p}_k^*) - C_{h_{k+1}}(h|\boldsymbol{p}_{k-1}^*) + C_{\mathbf{p}_{k+1}}(f, h|\boldsymbol{p}_k^*)\right]^{g-2} \\
& \times (P_{f_{k+1}}(f|\boldsymbol{p}_{k-1}^*) - \partial_f C_{\mathbf{p}_{k+1}}(f, h|\boldsymbol{p}_k^*))(P_{h_{k+1}}(h|\boldsymbol{p}_{k-1}^*) - \partial_h C_{\mathbf{p}_{k+1}}(f, h|\boldsymbol{p}_k^*)) \\
& + g \left[1 - C_{f_{k+1}}(f|\boldsymbol{p}_k^*) - C_{h_{k+1}}(h|\boldsymbol{p}_{k-1}^*) + C_{\mathbf{p}_{k+1}}(f, h|\boldsymbol{p}_k^*)\right]^{g-1} P_{\mathbf{p}_{k+1}}(f, h|\boldsymbol{p}_k^*),
\end{aligned} \tag{94}$$

where $\partial_f C_{\mathbf{p}_{k+1}}(f, h|\boldsymbol{p}_k^*) = \frac{\partial}{\partial h} C_{\mathbf{p}_{k+1}}(f, h|\boldsymbol{p}_k^*) = \int_0^h dh' P_{\mathbf{p}_{k+1}}(f', h|\boldsymbol{p}_k^*)$ and $\partial_h C_{\mathbf{p}_{k+1}}(f, h|\boldsymbol{p}_k^*) = \frac{\partial}{\partial h} C_{\mathbf{p}_{k+1}}(f, h|\boldsymbol{p}_k^*) = \int_0^f df' P_{\mathbf{p}_{k+1}}(f', h|\boldsymbol{p}_k^*)$.

If the mean $\overline{f_{k+1,\tau}}$ is smaller than $f_k^*$, the chosen collective is likely to have maximum $f$ values among $g$ matured collectives. Then, the definition of $f^*$ is written by $f_{k+1}^* = \max(f_{k+1}^{(1)}, f_{k+1}^{(2)}, \cdots, f_{k+1}^{(g)})$. We rewrite the joint cumulative distribution function $C_{\mathbf{p}_{k+1}}(f, h|\boldsymbol{p}_k^*)$ to be a little different from **Equation 89** because now we have to utilize the condition $f^* < f$ instead of $f^* > f$,

$$\begin{aligned}
C_{\mathbf{p}_{k+1}}(f, h|\boldsymbol{p}_k^*) = {} & \int_0^f df' \int_0^h dh' P_{\mathbf{p}_{k+1}^*}(f', h'|\boldsymbol{p}_k^*) \\
= {} & \int_0^f df' \left(\int_0^1 - \int_h^1\right) dh' P_{\mathbf{p}_{k+1}^*}(f', h'|\boldsymbol{p}_k^*) \\
= {} & \int_0^f df' \int_0^1 dh' P_{\mathbf{p}_{k+1}^*}(f', h'|\boldsymbol{p}_k^*) - \int_0^f df' \int_h^1 dh' P_{\mathbf{p}_{k+1}^*}(f', h'|\boldsymbol{p}_k^*). \\
= {} & \Pr[f_{k+1}^* < f|\boldsymbol{p}_k^*] - \Pr[f_{k+1}^* < f \wedge h_{k+1}^* \geq h|\boldsymbol{p}_k^*].
\end{aligned} \tag{95}$$

The probability $\Pr[f_{k+1}^* < f \wedge h_{k+1}^* \geq h|\boldsymbol{p}_k^*]$ is converted as

$$\begin{aligned}
& \Pr[f_{k+1}^* < f \wedge h_{k+1}^* \geq h|\boldsymbol{p}_k^*] \\
& = \Pr[f_{k+1}^{(1)} < f \wedge h_{k+1}^{(1)} \geq h \wedge f_{k+1}^{(2)} < f \wedge h_{k+1}^{(2)} \geq h \wedge \cdots \wedge f_{k+1}^{(2)} < f \wedge h_{k+1}^{(2)} \geq h|\boldsymbol{p}_k^*] \\
& = \left[\Pr[f_{k+1} < f \wedge h_{k+1} \geq h|\boldsymbol{p}_k^*]\right]^g \\
& = \left[\Pr[f_{k+1} < f|\boldsymbol{p}_k^*] - \Pr[f_{k+1} < f \wedge h_{k+1} < h|\boldsymbol{p}_k^*]\right]^g.
\end{aligned} \tag{96}$$

$$= \left[C_{f_{k+1}}(f|\boldsymbol{p}_k^*) - C_{\mathbf{p}_{k+1}}(f, h|\boldsymbol{p}_k^*)\right]^g. \tag{97}$$

Thus, the joint cumulative distribution function is given by

$$C_{\mathbf{p}_{k+1}^*}(f, h|\boldsymbol{p}_k^*) = \left[C_{f_{k+1}}(f|\boldsymbol{p}_k^*)\right]^g - \left[C_{f_{k+1}}(f|\boldsymbol{p}_k^*) - C_{\mathbf{p}_{k+1}}(f, h|\boldsymbol{p}_k^*)\right]^g. \tag{98}$$

In this case, the conditional probability distribution function is given by

$$P_{\mathbf{p}^*_{k+1}}(f,h|\boldsymbol{p}^*_k) = \frac{\partial^2}{\partial f \partial h} C_{\mathbf{p}^*_{k+1}}(f,h|\boldsymbol{p}^*_k)$$

$$= g(g-1)\left[C_{f_{k+1}}(f|\boldsymbol{p}^*_k) - C_{\mathbf{p}_{k+1}}(f,h|\boldsymbol{p}^*_k)\right]^{g-2}$$

$$\times (P_{f_{k+1}}(f|\boldsymbol{p}^*_k) - \partial_f C_{\mathbf{p}_{k+1}}(f,h|\boldsymbol{p}^*_k))\partial_h C_{\mathbf{p}_{k+1}}(f,h|\boldsymbol{p}^*_k)$$

$$+ g\left[C_{f_{k+1}}(f|\boldsymbol{p}^*_k) - C_{\mathbf{p}_{k+1}}(f,h|\boldsymbol{p}^*_k)\right]^{g-1} P_{\mathbf{p}_{k+1}}(f,h|\boldsymbol{p}^*_k). \tag{99}$$

By replacing $(f,h)$ to $\mathbf{p}^*_{k+1}$, we finally obtain the conditional probability distribution $\Psi(\mathbf{p}^*_{k+1}|\mathbf{p}^*_k)$,

$$\Psi(\mathbf{p}^*_{k+1}|\mathbf{p}^*_k) = \begin{cases} g(g-1)\left[1 - C_{f_{k+1}}(f^*_{k+1}|\boldsymbol{p}^*_k) - C_{h_{k+1}}(h^*_{k+1}|\boldsymbol{p}^*_k) + C_{\mathbf{p}_{k+1}}(\mathbf{p}^*_{k+1}|\boldsymbol{p}^*_k)\right]^{g-2} \\ \times (P_{f_{k+1}}(f^*_{k+1}|\boldsymbol{p}^*_k) - \partial_f C_{\mathbf{p}_{k+1}}(\mathbf{p}^*_{k+1}|\boldsymbol{p}^*_k))(P_{h_{k+1}}(h) - \partial_h C_{\mathbf{p}_{k+1}}(\mathbf{p}^*_{k+1}|\boldsymbol{p}^*_k)) \\ + g\left[1 - C_{f_{k+1}}(f^*_{k+1}|\boldsymbol{p}^*_k) - C_{h_{k+1}}(h^*_{k+1}|\boldsymbol{p}^*_k) + C_{\mathbf{p}_{k+1}}(\mathbf{p}^*_{k+1}|\boldsymbol{p}^*_k)\right]^{g-1} P_{\mathbf{p}_{k+1}}(\mathbf{p}^*_{k+1}|\boldsymbol{p}^*_k) \\ \qquad\qquad\qquad \text{for } \overline{f_{k+1,\tau}} - f^*_k \geq 0 \text{ and } \overline{h_{k+1,\tau}} - h^*_k \geq 0 \\ g(g-1)\left[C_{f_{k+1}}(f^*_{k+1}|\boldsymbol{p}^*_k) - C_{\mathbf{p}_{k+1}}(\mathbf{p}^*_{k+1}|\boldsymbol{p}^*_k)\right]^{g-2} \\ (P_{f_{k+1}}(f^*_{k+1}|\boldsymbol{p}^*_k) - \partial_f C_{\mathbf{p}_{k+1}}(\mathbf{p}^*_{k+1}|\boldsymbol{p}^*_k))\partial_h C_{\mathbf{p}_{k+1}}(\mathbf{p}^*_{k+1}|\boldsymbol{p}^*_k) \\ + g\left[C_{f_{k+1}}(f^*_{k+1}|\boldsymbol{p}^*_k) - C_{\mathbf{p}_{k+1}}(\mathbf{p}^*_{k+1}|\boldsymbol{p}^*_k)\right]^{g-1} P_{\mathbf{p}_{k+1}}(\mathbf{p}^*_{k+1}|\boldsymbol{p}^*_k) \\ \qquad\qquad\qquad \text{for } \overline{f_{k+1,\tau}} - f^*_k < 0 \text{ and } \overline{h_{k+1,\tau}} - h^*_k \geq 0 \end{cases} \tag{100}$$

Using **Equation 100**, we get the mean values of $f^*$ and $h^*$ as

$$\overline{f^*_{k+1}} = \int_0^1 df' \int_0^1 dh'\, f' \times P_{f^*_{k+1}, h^*_{k+1}}(f', h'|f^*_k, h^*_k), \tag{101}$$

$$\overline{h^*_{k+1}} = \int_0^1 df' \int_0^1 dh'\, h' \times P_{f^*_{k+1}, h^*_{k+1}}(f', h'|f^*_k, h^*_k). \tag{102}$$

We define the accessible region in frequency space where the signs of the changes in both F frequency and FF frequency after a cycle are opposite to that of maturation (see **Appendix 8—figure 1**),

$$\text{sign}\left(\overline{f^*_{k+1}} - f^*_k\right) \times \text{sign}\left(\overline{f_{k+1,\tau}} - f^*_k\right) \leq 0 \text{ and } \text{sign}\left(\overline{h^*_{k+1}} - h^*_k\right) \times \text{sign}\left(\overline{h_{k+1,\tau}} - h^*_k\right) \leq 0, \tag{103}$$

where $\overline{f_{k+1,\tau}}$ and $\overline{f^*_{k+1}}$ are the mean values of F frequencies after the maturation step in cycle $k+1$ before and after selection, respectively, and $h$ values are defined similarly for FF. Or, if the condition is not met, the composition of the selected collective may diverge from the target composition after several cycles. The accessible regions are marked in the gold-colored area in **Appendix 8—figure 1b**. Similar to the two-population case, the accessible region is shaped by the flow velocity of the composition during the maturation step, as depicted in the flow diagram in **Appendix 8—figure 1a**. Both F and FF frequencies tend to increase, and the inter-collective selection can compensate for these changes if the composition changes slowly when the F and FF frequencies are small. However, if the changes occur too rapidly when the FF frequency is intermediate, the frequency cannot be stabilized. So the accessible region is limited to the regions where the composition changes slowly.

This is explainable by projecting the three-population problem into the two-population problem. The selective advantage of FF relative to the rest of the collective mainly determines the accessible region. The growth rate of the rest varies from $r_S$ to $r_S + \omega$ according to F frequency, so the mean growth rate of the rest is written by $\overline{r_S} = r_S + f'\omega$ where $f'$ is F frequency in S+F. Then, the corresponding selective advantage of FF is $\overline{\omega} = (2 - f')\omega$ which varies between $\omega$ to $2\omega$. Using $\overline{r_S}$ and $\overline{\omega}$ similar to Appendix 2, we get bounds of the accessible region (see dashed line in **Appendix 8—figure 1b**). The boundary from the projected problem agreed well with the original three-population problem.

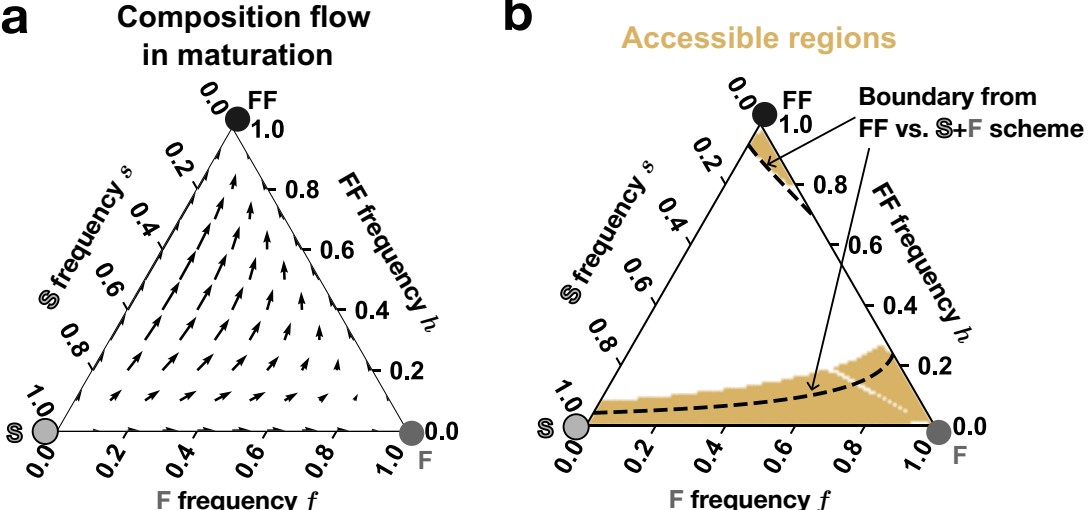

**Appendix 8—figure 1.** Accessible regions in the three-population system. (**a**) The flow of composition change in fast-growing (F) and faster-growing (FF) frequencies at each composition $(f, h)$. Top corner indicates that FF cells fix in the collective. Right bottom corner means collectives with only F cells, while collectives contain S cells only at left bottom corner. Arrow length means the speed of change. (**b**) The accessible regions are marked by the gold area. If the signs of changes in both F frequency and FF frequency after inter-collective selection are opposite to those during maturation, then the given composition is accessible. Otherwise, the composition is not accessible and will change after cycles. Dashed lines are the boundary of the accessible region by projecting the collective into a two-population problem (FF vs. S+F). The figures are drawn using the *mpltern* package (***Ikeda et al., 2019***).

# Appendix 9

## Derivation of equations

In this section, we go over the derivation of *Equations 18–42* for readers not equipped with advanced mathematics training.

Assumptions: $\mu \ll \omega, r_S$

## Equations 18 and 19

*Equation 18* is straightforwardly solved by integrating *Equation 16*. *Equation 19* is obtained from *Equation 17* using integration factor $e^{-(r_S+\omega)t}$:

$$\left[\frac{d\overline{F}(t)}{dt} - (r_S + \omega)\overline{F}(t)\right] e^{-(r_S+\omega)t} = \mu \overline{S}_o e^{(r_S-\mu)t} e^{-(r_S+\omega)t}, \tag{104}$$

$$\frac{d\left[\overline{F}(t)e^{-(r_S+\omega)t}\right]}{dt} = \mu \overline{S}_o e^{-(\omega+\mu)t}. \tag{105}$$

Integrating both sides, we get $\overline{F}(t)e^{-(r_S+\omega)t} - \overline{F}(0) = \frac{\mu \overline{S}_o}{\omega+\mu}\left(1 - e^{-(\omega+\mu)t}\right)$. Thus,

$$\overline{F}(t) = \overline{F}_o e^{(r_S+\omega)t} + \frac{\mu \overline{S}_o}{\omega+\mu}\left(e^{(r_S+\omega)t} - e^{(r_S-\mu)t}\right) \approx \overline{F}_o e^{(r_S+\omega)t} + \frac{\mu \overline{S}_o}{\omega}\left(e^{(r_S+\omega)t} - e^{r_S t}\right). \tag{106}$$

## Equations 24 and 25

Applying *Equation 12*, we have

$$\begin{aligned}
\frac{d\overline{S^2}}{dt} &= \sum_{S=0}^{\infty}\sum_{F=0}^{\infty} S^2 \frac{dP(S,F,t)}{dt} \\
&= \sum_{S=0}^{\infty}\sum_{F=0}^{\infty}[r_S S^2(S-1)P(S-1,F,t) + S^2(r_S+\omega)(F-1)P(S,F-1,t) \\
&\quad + \mu(S+1)S^2 P(S+1,F-1,t) - (rS + (r_S+\omega)F + \mu S)S^2 P(S,F,t)].
\end{aligned} \tag{107}$$

We collect the two purple-colored terms and change the order of summation. Note that the first purple-colored term does not change regardless of whether $S$ starts from 0 or 1 because the term is zero for $S = 0$. Thus, the first purple-colored term is equivalent to $\sum_{F=0}^{\infty}\sum_{S=1}^{\infty}\left[r_S S^2(S-1)P(S-1,F,t)\right]$. Let $\alpha = S-1$, and this becomes $\sum_{F=0}^{\infty}\sum_{\alpha=0}^{\infty}\left[r_S(\alpha+1)^2\alpha P(\alpha,F,t)\right]$. We reassign $\alpha$ as $S$, and obtain:

$$\begin{aligned}
&\sum_{F=0}^{\infty}\sum_{S=0}^{\infty}\left[r_S(S+1)^2 S P(S,F,t)\right] - \sum_{F=0}^{\infty}\sum_{S=0}^{\infty} r_S S^3 P(S,F,t) \\
&= r_S \sum_{F=0}^{\infty}\sum_{S=0}^{\infty}[(2S+1)S]P(S,F,t) \\
&= r_S\left(2\overline{S^2} + \overline{S}\right).
\end{aligned} \tag{108}$$

We collect the two blue terms, and similarly obtain:

$$\begin{aligned}
&\sum_{S=0}^{\infty}\sum_{F=0}^{\infty}\left[S^2(r_S+\omega)(F-1)P(S,F-1,t) - (r_S+\omega)FS^2 P(S,F,t)\right] \\
&= (r_S+\omega)\sum_{S=0}^{\infty}\sum_{F=0}^{\infty}\left[S^2 F P(S,F,t) - FS^2 P(S,F,t)\right] \\
&= 0.
\end{aligned} \tag{109}$$

Finally, we collect the two red terms. For the first red term, the sum is the same regardless of whether we start from $S = 0$ or -1. Let $S$ start from -1, and we have

$$\sum_{F=0}^{\infty} \sum_{S=-1}^{\infty} \left[ \mu(S+1)S^2 P(S+1, F-1, t) \right].$$

(110)

Let $\alpha = S + 1$, then the term becomes $\sum_{F=0}^{\infty} \sum_{\alpha=0}^{\infty} \left[ \mu\alpha(\alpha - 1)^2 P(\alpha, F-1, t) \right]$. We reassign $\alpha$ as $S$, and additionally apply index change on $F - 1$:

$$\mu \sum_{F=0}^{\infty} \sum_{S=0}^{\infty} \left[ S(S-1)^2 P(S, F, t) - S^3 P(S, F, t) \right]$$

$$= \mu \sum_{F=0}^{\infty} \sum_{S=0}^{\infty} \left[ (-2S^2 + S)P(S, F, t) \right]$$

(111)

$$= \mu \left( -2\overline{S^2} + \overline{S} \right).$$

Now, add the three parts together, and we have

$$\frac{d\overline{S^2}}{dt} = \mu \left( -2\overline{S^2} + \overline{S} \right) + r_S(2\overline{S^2} + \overline{S})$$

$$= 2\overline{S^2}(r_S - \mu) + \overline{S}(r_S + \mu),$$

(112)

which is *Equation 24*. Likewise,

$$\frac{d\overline{F^2}}{dt} = \sum_{S=0}^{\infty} \sum_{F=0}^{\infty} F^2 \frac{dP(S, F, t)}{dt}$$

$$= \sum_{S=0}^{\infty} \sum_{F=0}^{\infty} [r_S F^2 (S-1)P(S-1, F, t) + F^2(r_S + \omega)(F-1)P(S, F-1, t)$$

$$+ \mu(S+1)F^2 P(S+1, F-1, t) - (r_S S + (r_S + \omega)F + \mu S)F^2 P(S, F, t)]$$

$$= \sum_{S=0}^{\infty} \sum_{F=0}^{\infty} \left[ r_S F^2 S + (F+1)^2(r_S + \omega)F + \mu S(F+1)^2 - (r_S S + (r_S + \omega)F + \mu S)F^2 \right] P(S, F, t)$$

(113)

$$= \sum_{S=0}^{\infty} \sum_{F=0}^{\infty} \left[ (F+1)^2(r_S + \omega)F + \mu S(F+1)^2 - ((r_S + \omega)F + \mu S)F^2 \right] P(S, F, t)$$

$$= \sum_{S=0}^{\infty} \sum_{F=0}^{\infty} \left[ (r_S + \omega)F(2F+1) + \mu S(2F+1) \right] P(S, F, t)$$

$$= 2(r_S + \omega)\overline{F^2} + (r_S + \omega)\overline{F} + 2\mu\overline{SF} + \mu\overline{S}$$

(114)

which is *Equation 25*.

### Equation 26 and 28
Using integration factor $e^{-2(r_S - \mu)t}$ and *Equation 24*, we have:

$$\left[ \frac{d\overline{S^2}}{dt} - 2(r_S - \mu)\overline{S^2} \right] e^{-2(r_S - \mu)t} = (r_S + \mu)\overline{S}_o e^{(r_S - \mu)t} e^{-2(r_S - \mu)t},$$

(115)

$$\frac{d\left[ \overline{S^2} e^{-2(r_S - \mu)t} \right]}{dt} = (r_S + \mu)\overline{S}_o e^{(\mu - r_S)t},$$

(116)

$$\overline{S^2} e^{-2(r_S - \mu)t} = \frac{r_S + \mu}{\mu - r_S} \overline{S}_o \left[ e^{(\mu - r_S)t} - 1 \right] + \overline{S^2}_o.$$

(117)

Since $\mu \ll r_S$, we have

$$\overline{S^2} = \frac{r_S + \mu}{\mu - r_S}\overline{S}_o \left[e^{(\mu - r_S)t} - 1\right] e^{2(r_S - \mu)t} + \overline{S^2}_o e^{2(r_S - \mu)t}$$

$$\approx \overline{S}_o e^{2r_St} \left[1 - e^{-r_St}\right] + \overline{S^2}_o e^{2r_St}, \tag{118}$$

where $\overline{S^2}_o$ is the expected $S^2$ at time 0. For *Equation 28*,

$$\sigma_S^2(t) = \overline{S^2}(t) - [\overline{S}(t)]^2$$

$$\approx \overline{S}_o e^{2r_St} \left[1 - e^{-r_St}\right] + \overline{S^2}_o e^{2r_St} - \left(\overline{S}_o e^{r_St}\right)^2 \tag{119}$$

$$= \overline{S}_o e^{2r_St} \left[1 - e^{-r_St}\right] + e^{2r_St}\left(\overline{S^2}_o - \left(\overline{S}_o\right)^2\right) = \overline{S}_o e^{2r_St} \left[1 - e^{-r_St}\right] + e^{2r_St}\sigma_S^2(0).$$

## Equation 30 and 31

Since $\overline{S\,F} = \sum_{S=0}^{\infty}\sum_{F=0}^{\infty} S F P(S,F,t)$,

$$\frac{d\overline{S\,F}}{dt} = \sum_{S=0}^{\infty}\sum_{F=0}^{\infty} \left[S F r_S(S-1)P(S-1,F,t) + S F(r_S + \omega)(F-1)P(S,F-1,t)\right.$$

$$+ S F \mu(S+1)P(S+1,F-1,t)$$

$$\left. - S F(r_S S + (r_S + \omega)F + \mu S)P(S,F,t)\right]$$

$$= \sum_{S=0}^{\infty}\sum_{F=0}^{\infty} \left[(S+1)F r_S S + S(F+1)(r_S + \omega)F + (S-1)(F+1)\mu S\right. \tag{120}$$

$$\left. - S F(r_S S + (r_S + \omega)F + \mu S)\right]P(S,F,t)$$

$$= \sum_{S=0}^{\infty}\sum_{F=0}^{\infty} \left[2r_S S F + \omega S F - \mu S F + \mu S^2 - \mu S\right]P(S,F,t)$$

$$= (2r_S + \omega - \mu)\overline{S\,F} + \mu(\overline{S^2} - \overline{S}).$$

We can solve this, again using the integration factor technique above:

$$\frac{d[\overline{S\,F}e^{-(2r_S+\omega-\mu)t}]}{dt} = \mu(\overline{S^2} - \overline{S})e^{-(2r_S+\omega-\mu)t}. \tag{121}$$

Thus, we have

$$\overline{S\,F}e^{-(2r_S+\omega-\mu)t} - \overline{S\,F}_o = \mu \int (\overline{S^2} - \overline{S})e^{-(2r_S+\omega-\mu)t}dt$$

$$= \mu \int \left(\frac{r_S + \mu}{\mu - r_S}\overline{S}_o \left[e^{(\mu-r_S)t} - 1\right]e^{2(r_S-\mu)t}\right.$$

$$\left. + \overline{S^2}_o e^{2(r_S-\mu)t} - \overline{S}_o e^{(r_S-\mu)t}\right)e^{-(2r_S+\omega-\mu)t}dt$$

$$= \mu \int \left(\frac{r_S + \mu}{\mu - r_S}\overline{S}_o \left[e^{-(\mu-r_S)t} - e^{2(r_S-\mu)t}\right]e^{-(2r_S+\omega-\mu)t}\right.$$

$$\left. + \overline{S^2}_o e^{2(r_S-\mu)t}e^{-(2r_S+\omega-\mu)t} - \overline{S}_o e^{(r_S-\mu)t-(2r_S+\omega-\mu)t}\right)dt$$

$$= \mu \int \left(\frac{r_S + \mu}{\mu - r_S}\overline{S}_o e^{-(r_S+\omega)t} - \frac{r_S + \mu}{\mu - r_S}\overline{S}_o e^{-(\mu+\omega)t}\right.$$

$$\left. + \overline{S^2}_o e^{-(\mu+\omega)t} - \overline{S}_o e^{-(r_S+\omega)t}\right)dt$$

$$= \mu\left\{\frac{2r_S}{\mu - r_S}\overline{S}_o \frac{e^{-(r_S+s)t} - 1}{-(r_S + \omega)}\right.$$

$$\left. + \left(\overline{S^2}_o - \frac{r_S + \mu}{\mu - r_S}\overline{S}_o\right)\frac{e^{-(\mu+\omega)t} - 1}{-(\mu + \omega)}\right\}, \tag{122}$$

which results in

$$\overline{SF}(t) = \overline{SF}_o e^{(2r_S+\omega-\mu)t} + \mu \left\{ \frac{2r_S}{\mu-r_S}\overline{S}_o \frac{e^{-(r_S+\omega)t}-1}{-(r_S+\omega)} + (\overline{S^2}_o - \frac{r_S+\mu}{\mu-r_S}\overline{S}_o)\frac{e^{-(\mu+\omega)t}-1}{-(\mu+\omega)} \right\} e^{(2r_S+\omega-\mu)t}$$

$$= \overline{SF}_o e^{(2r_S+\omega-\mu)t} + \frac{2r_S\mu}{\mu-r_S}\overline{S}_o \frac{e^{(r_S-\mu)t}-e^{(2r_S+\omega-\mu)t}}{-(r_S+\omega)}$$

$$+ \mu\left(\overline{S^2}_o - \frac{r_S+\mu}{\mu-r_S}\overline{S}_o\right)\frac{e^{2(r_S-\mu)t}-e^{(2r_S+\omega-\mu)t}}{-(\mu+\omega)}$$

$$\approx \overline{SF}_o e^{(2r_S+\omega)t} - \frac{2\mu\overline{S}_o}{r_S+\omega}\left(e^{(2r_S+\omega)t}-e^{r_St}\right) + \frac{\mu}{\omega}(\overline{S^2}_o+\overline{S}_o)\left(e^{(2r_S+\omega)t}-e^{2r_St}\right).$$

(123)

## Equation 32

From *Equation 25*, we have

$$\left(\frac{d\overline{F^2}}{dt}-2(r_S+\omega)\overline{F^2}\right)e^{-2(r_S+\omega)t} = \frac{d\left(\overline{F^2}e^{-2(r_S+\omega)t}\right)}{dt} = \left((r_S+\omega)\overline{F}+2\mu\overline{SF}+\mu\overline{S}\right)e^{-2(r_S+\omega)t}.$$ (124)

The right-hand side becomes

$$\approx (r_S+\omega)\left(\overline{F}_o e^{(r_S+\omega)t} + \frac{\mu\overline{S}_o}{\omega+\mu}\left(e^{(r_S+\omega)t}-e^{(r_S-\mu)t}\right)\right)e^{-2(r_S+\omega)t}$$

$$+ 2\mu\overline{SF}_o e^{-(\omega+\mu)t} + \mu\overline{S}_o e^{-(r_S+2\omega+\mu)t} + o(\mu^2)$$

$$= (r_S+\omega)\left(\overline{F}_o e^{-(r_S+\omega)t} + \frac{\mu\overline{S}_o}{\omega+\mu}\left(e^{-(r_S+\omega)t}-e^{-(r_S+2\omega+\mu)t}\right)\right)$$

$$+ 2\mu\overline{SF}_o e^{-(\omega+\mu)t} + \mu\overline{S}_o e^{-(r_S+2\omega+\mu)t} + o(\mu^2)$$

$$= (r_S+\omega)\left(\frac{\overline{F}_o\omega+N_0\mu}{\omega+\mu}e^{-(r_S+\omega)t} - \frac{\mu\overline{S}_o}{\omega+\mu}e^{-(r_S+2\omega+\mu)t}\right)$$

$$+ 2\mu\overline{SF}_o e^{-(\omega+\mu)t} + \mu\overline{S}_o e^{-(r_S+2\omega+\mu)t} + o(\mu^2).$$

(125)

Note that we have checked that the second and third terms of $\overline{SF}$ can be ignored after we compare the full calculation with this simpler version. Integrate both sides:

$$\overline{F^2}e^{-2(r_S+\omega)t} - \overline{F^2}_o \approx (r_S+\omega)\left(\frac{\overline{F}_o\omega+N_0\mu}{(\omega+\mu)(r_S+\omega)}\left(1-e^{-(r_S+\omega)t}\right)\right.$$

$$\left.+ \frac{\mu\overline{S}_o}{(\omega+\mu)(r_S+2\omega+\mu)}\left(e^{-(r_S+2\omega+\mu)t}-1\right)\right)$$

$$+ \frac{2\mu\overline{SF}_o}{\omega+\mu}(1-e^{-(\omega+\mu)t}) + \frac{\mu\overline{S}_o}{r_S+2\omega+\mu}\left(1-e^{-(r_S+2\omega+\mu)t}\right) + o(\mu^2)$$

(126)

Then, we have

$$\overline{F^2}(t) \approx \overline{F^2}_o e^{2(r_S+\omega)t} + (r_S+\omega)\left(\frac{\overline{F}_o\omega + N_0\mu}{(\omega+\mu)(r_S+\omega)}(e^{2(r_S+\omega)t} - e^{(r_S+\omega)t})\right.$$

$$\left. + \frac{\mu\overline{S}_o}{(\omega+\mu)(r_S+2\omega+\mu)}(e^{(r_S-\mu)t} - e^{2(r_S+\omega)t})\right)$$

$$+ \frac{2\mu\overline{S\,F}_o}{\omega+\mu}(e^{2(r_S+\omega)t} - e^{(2r_S+\omega-\mu)t}) + \frac{\mu\overline{S}_o}{r_S+2\omega+\mu}(e^{2(r_S+\omega)t} - e^{(r_S-\mu)t}) + o(\mu^2)$$

$$= \overline{F^2}_o e^{2(r_S+\omega)t} + \left(\frac{\overline{F}_o\omega + N_0\mu}{\omega}(e^{2(r_S+\omega)t} - e^{(r_S+\omega)t}) + \left(\frac{1}{\omega} - \frac{1}{r_S+2\omega}\right)\mu\overline{S}_o(e^{r_S t} - e^{2(r_S+\omega)t})\right)$$

$$+ \frac{2\mu\overline{S\,F}_o}{\omega}e^{(r_S+\omega)t}(e^{(r_S+\omega)t} - e^{r_S t}) + \frac{\mu\overline{S}_o}{r_S+2\omega}(e^{2(r_S+\omega)t} - e^{r_S t}) + o(\mu^2)$$

$$= \overline{F^2}_o e^{2(r_S+\omega)t} + \frac{2\mu\overline{S\,F}_o}{\omega}e^{(r_S+\omega)t}(e^{(r_S+\omega)t} - e^{r_S t}) + \overline{F}_o e^{(r_S+\omega)t}(e^{(r_S+\omega)t} - 1)$$

$$+ \frac{\mu\overline{S}_o}{\omega}\left[(-e^{(r_S+\omega)t}) + \frac{\overline{F}_o}{\overline{S}_o}(e^{2(r_S+\omega)t} - e^{(r_S+\omega)t}) + \frac{2\omega}{r_S+2\omega}e^{2(r_S+\omega)t} + \frac{r_S}{r_S+2\omega}e^{r_S t}\right] + o(\mu^2)$$

$$= \overline{F^2}_o e^{2(r_S+\omega)t} + \frac{2\mu\overline{S\,F}_o}{\omega}e^{(r_S+\omega)t}(e^{(r_S+\omega)t} - e^{r_S t}) + \overline{F}_o e^{(r_S+\omega)t}(e^{(r_S+\omega)t} - 1)$$

$$+ \frac{\mu\overline{S}_o}{\omega}\left[(-e^{(r_S+\omega)t}) + \frac{2\omega}{r_S+2\omega}e^{2(r_S+\omega)t} + \frac{r_S}{r_S+2\omega}e^{r_S t}\right] + \frac{\mu\overline{F}_o}{\omega}(e^{2(r_S+\omega)t} - e^{(r_S+\omega)t}) + o(\mu^2)$$

$$= \overline{F^2}_o e^{2(r_S+\omega)t} + \frac{2\mu\overline{S\,F}_o}{\omega}e^{(r_S+\omega)t}(e^{(r_S+\omega)t} - e^{r_S t}) + \overline{F}_o(1+\frac{\mu}{\omega})e^{(r_S+\omega)t}(e^{(r_S+\omega)t} - 1)$$

$$+ \frac{\mu\overline{S}_o}{\omega}\left[(-e^{(r_S+\omega)t}) + \frac{2\omega}{r_S+2\omega}e^{2(r_S+\omega)t} + \frac{r_S}{r_S+2\omega}e^{r_S t}\right] + o(\mu^2)$$

$$= \overline{F^2}_o e^{2(r_S+\omega)t} + \frac{2\mu\overline{S\,F}_o}{\omega}e^{(r_S+\omega)t}(e^{(r_S+\omega)t} - e^{r_S t}) + \overline{F}_o e^{(r_S+\omega)t}(e^{(r_S+\omega)t} - 1)$$

$$+ \frac{\mu\overline{S}_o}{\omega}\left[(-e^{(r_S+\omega)t}) + \frac{2\omega}{r_S+2\omega}e^{2(r_S+\omega)t} + \frac{r_S}{r_S+2\omega}e^{r_S t}\right] + o(\mu^2). \tag{127}$$

### Equation 33

$$\sigma_F^2(t) = \overline{F^2}(t) - [\overline{F}(t)]^2$$

$$\approx \overline{F^2}_o e^{2(r_S+\omega)t} + \frac{2\mu\overline{S\,F}_o}{\omega}e^{(r_S+\omega)t}(e^{(r_S+\omega)t} - e^{r_S t}) + \overline{F}_o e^{(r_S+\omega)t}(e^{(r_S+\omega)t} - 1)$$

$$+ \frac{\mu\overline{S}_o}{\omega}\left[(-e^{(r_S+\omega)t}) + \frac{2\omega}{r_S+2\omega}e^{2(r_S+\omega)t} + \frac{r_S}{r_S+2\omega}e^{r_S t}\right] + o(\mu^2)$$

$$- \left[\overline{F}_o e^{(r_S+\omega)t} + \frac{\mu\overline{S}_o}{\omega}(e^{(r_S+\omega)t} - e^{r_S t})\right]^2$$

$$= \overline{F^2}_o e^{2(r_S+\omega)t} + \frac{2\mu\overline{SF}_o}{\omega}e^{(r_S+\omega)t}(e^{(r_S+\omega)t} - e^{r_S t}) + \overline{F}_o e^{(r_S+\omega)t}(e^{(r_S+\omega)t} - 1) \tag{128}$$

$$+ \frac{\mu\overline{S}_o}{\omega}\left[(-e^{(r_S+\omega)t}) + \frac{2\omega}{r_S+2\omega}e^{2(r_S+\omega)t} + \frac{r_S}{r_S+2\omega}e^{r_S t}\right] + o(\mu^2)$$

$$- \overline{F}_o^2 e^{2(r_S+\omega)t} - 2\overline{F}_o e^{(r_S+\omega)t}\frac{\mu\overline{S}_o}{\omega}(e^{(r_S+\omega)t} - e^{r_S t}) - \left[\frac{\mu\overline{S}_o}{\omega}(e^{(r_S+\omega)t} - e^{r_S t})\right]^2$$

$$= \sigma_{F,0}^2 e^{2(r_S+\omega)t} + \frac{2\mu(\overline{S\,F}_o - \overline{S}_o\overline{F}_o)}{\omega}e^{(r_S+\omega)t}(e^{(r_S+\omega)t} - e^{r_S t})$$

$$+ \overline{F}_o e^{(r_S+\omega)t}(e^{(r_S+\omega)t} - 1) + \frac{\mu\overline{S}_o}{\omega}\left[-e^{(r_S+\omega)t} + \frac{2\omega}{r_S+2\omega}e^{2(r_S+\omega)t} + \frac{r_S}{r_S+2\omega}e^{r_S t}\right] + o(\mu^2).$$

### Equations 40-42

To derive this equation, we use the fact that $1/(1+x) \sim 1-x$ for small $x$. We will omit $(t)$ for simplicity. Also, note that we are considering relatively large populations so that the standard deviation is much smaller than the mean.

$$f \approx \frac{\overline{F} + \sigma_F G_F}{\overline{S} + \sigma_S G_S + \overline{F} + \sigma_F G_F}$$

$$= \frac{\overline{F}\left(1 + \frac{\sigma_F}{\overline{F}} G_F\right)}{(\overline{S} + \overline{F})(1 + \frac{\sigma_S G_S + \sigma_F G_F}{\overline{S} + \overline{F}})}$$

$$\approx \frac{\overline{F}}{\overline{S} + \overline{F}}\left(1 + \frac{\sigma_F}{\overline{F}} G_F\right)\left(1 - \frac{\sigma_S G_S + \sigma_F G_F}{\overline{S} + \overline{F}}\right)$$

$$\approx \bar{f}\left(1 + \frac{\sigma_F}{\overline{F}} G_F - \frac{\sigma_S G_S + \sigma_F G_F}{\overline{S} + \overline{F}}\right)$$

$$= \bar{f}\left(1 + \left(\frac{\sigma_F}{\overline{F}} - \frac{\sigma_F}{\overline{S} + \overline{F}}\right) G_F - \frac{\sigma_S}{\overline{S} + \overline{F}} G_S\right). \tag{129}$$

Recall that if $A \sim \mathcal{N}(\mu_A, \sigma_A)$, $B \sim \mathcal{N}(\mu_B, \sigma_B)$, then $A - B \sim \mathcal{N}(\mu_A - \mu_B, \sqrt{\sigma_A^2 + \sigma_B^2})$. Thus, $f$ is distributed as a Gaussian with the mean of

$$\bar{f}(t) = \frac{\overline{F}(t)}{\overline{S}(t) + \overline{F}(t)} \approx \frac{\overline{F}_o e^{(r_S + \omega)t} + \frac{\mu \overline{S}_o}{\omega}(e^{(r_S + \omega)t} - e^{r_S t})}{\overline{S}_o e^{r_S t} + \overline{F}_o e^{(r_S + \omega)t} + \frac{\mu \overline{S}_o}{\omega}(e^{(r_S + \omega)t} - e^{r_S t})}$$

$$= \frac{\overline{F}_o e^{\omega t} + \frac{\mu \overline{S}_o}{\omega}(e^{\omega t} - 1)}{\overline{S}_o + \overline{F}_o e^{\omega t} + \frac{\mu \overline{S}_o}{\omega}(e^{\omega t} - 1)} \tag{130}$$

$$= \frac{\bar{f}_o e^{\omega t} + \frac{\mu}{\omega}(1 - \bar{f}_o)(e^{\omega t} - 1)}{(1 - \bar{f}_o) + \bar{f}_o e^{\omega t} + \frac{\mu}{\omega}(1 - \bar{f}_o)(e^{\omega t} - 1)}.$$

Note that the initial value of mean $\bar{f}(0)$ is equal to the mean of the binomial distribution **Equation 37**, $f_k^*$. The variance is

$$\sigma_f^2(t) = \bar{f}^2 \left(\left(\frac{\overline{S}}{(\overline{S} + \overline{F})\overline{F}}\right)^2 \sigma_F^2(t) + \left(\frac{1}{\overline{S} + \overline{F}}\right)^2 \sigma_S^2(t)\right)$$

$$= \bar{f}^2 \left(\frac{(1 - \bar{f}(t))^2}{\overline{N}(t)^2 (\bar{f}(t))^2} \sigma_F^2(t) + \frac{1}{\overline{N}(t)^2} \sigma_S^2(t)\right)$$

$$= \frac{(1 - \bar{f}(t))^2 \sigma_F^2(t) + \bar{f}(t)^2 \sigma_S^2(t)}{\overline{N}(t)^2}. \tag{131}$$

