## [Editor Report · eLife Assessment]

This **important** study of artificial selection in microbial communities shows that the possibility of selecting a desired fraction of slow and fast-growing types is impacted by their initial fractions. The evidence, which relies on mathematical analysis and simulations of a stochastic model, is **compelling**. It highlights the tension between selection at the strain and the community level. This study should be of interest to researchers interested in ecology, both theoretical and experimental.

---

## [Referee Report · Reviewer #1 (Public review)]

Summary:

The authors demonstrate with a simple stochastic model that the initial composition of the community is important in achieving a target frequency during the artificial selection of a community.

Strengths:

To my knowledge, the intra-collective selection during artificial selection has not been seriously theoretically considered. However, in many cases, the species dynamics during the incubation of each selection cycle is important and relevant to the outcome of the artificial selection experiment. Stochasticity from birth and death (demographic stochasticity) plays a big role in these species' abundance dynamics. This work uses a simple framework to tackle this idea meticulously.

This work may or may not be related to hysteresis (path dependency). If this is true, maybe it would be nice to have a discussion paragraph talking about how this may be the case. Then, this work would even attract the interest of people studying dynamical systems.

Weaknesses:

(1) Connecting structure and function.

In typical artificial selection literature, most of them select the community based on collective function. Here in this paper, the authors are selecting a target composition. Although there is a schematic cartoon illustrating the relationship between collective function (y-axis) and the community composition in the main figure 1, there is no explicit explanation or justification of what may be the origin of this relationship. I think giving the readers a naïve idea about how this structure-function relationship arises in the introduction section would help. This is because the conclusion of this paper is that the intra-collective selection makes it hard to artificially select for a community that has an intermediate frequency of f (or s). If there is really evidence or theoretical derivation from this framework that indeed the highest function comes from the intermediate frequency of f, then the impact of this paper would increase because the conclusions of this stochastic model could allude to the reasons for the prevalent failures of artificial selection in literature.

(2) Explain intra-collective and inter-collective selection better for readers.

The abstract, the introduction, and the result section use these terms or intra-collective and inter-collective selection without much explanation. For the wide readership of eLife, a clear definition in the beginning would help the audience grasp the importance of this paper, because these concepts are at the core of this work.

(3) Achievable target frequency strongly depending on the degree of demographic stochasticity.

I would expect that the experimentalists would find these results interesting and would want to consider these results during their artificial selection experiments. The main figure 4 indicates that the Newborn size N0 is a very important factor to consider during the artificial selection experiment. This would be equivalent to how much bottleneck you impose on the artificial selection process in every iteration step (i.e., the ratio of serial dilution experiment). However, with a low population size, all target frequencies can be achieved, and therefore in these regimes, the initial frequency now does not matter much. It would be great for the authors to provide what the N0 parameter actually means during the artificial selection experiments. Maybe relative to some other parameter in the model. I know this could be very hard. But without this, the main result of this paper (initial frequency matters) cannot be taken advantage of by the experimentalists.

(4) Consideration of environmental stochasticity.

The success (gold area of Figure 2d) in this framework mainly depends on the size of the demographic stochasticity (birth-only model) during the intra-collective selection. However, during experiments, a lot of environmental stochasticity appears to be occurring during artificial selection. This may be out of the scope of this study. But it would definitely be exciting to see how much environmental stochasticity relative to the demographic stochasticity (variation in the Gaussian distribution of F and S) matters in succeeding in achieving the target composition from artificial selection.

(5) Assumption about mutation rates

If setting the mutation rates to zero does not change the result of the simulations and the conclusion, what is the purpose of having the mutation rates \mu? Also, is the unidirectional (S -> F -> FF) mutation realistic? I didn't quite understand how the mutations could fit into the story of this paper.

(6) Minor points

In Figure 3b, it is not clear to me how the frequency difference for the Intra-collective and the Inter-collective selection is computed.

In Figure 5b, the gold region (success) near the FF is not visible. Maybe increase the size of the figure or have an inset for zoom-in. Why is the region not as big as the bottom gold region?

Comments on revisions:

I thank the authors for addressing many points raised by the reviewers. Overall, the readability of the manuscript has improved with more context provided around why they were solving this specific problem. However, I've found many of the responses to be too terse. It would have been nicer if there had been more discussion and description of the thought process that led up to the conclusions they made for each comment or question. Instead, many of the responses only showed the screenshot of the text they added.

Most of my comments or questions were answered. Below are my comments on some of the authors' responses.

(2) Explain intra-collective and inter-collective selection better for readers.

In the Abstract and Introduction, you've added more sentences about the intra-collective or inter-collective selection. However, these are either making analogies to the waterfall or just describing the result of the intra/inter-collective selection. I would still appreciate a proper definition of those terms, which is paramount for readers to understand the entire paper.

(4) Consideration of environmental stochasticity.

I think providing the reason 'why' the paper focuses on demographic stochasticity and not environmental stochasticity will greatly justify the paper's work. For example, citing papers that actually performed artificial selection and pointing out that your model captures the stochasticity from those kinds of experiments would be great.

(5) Assumption about mutation rates.

It would be great if you could add a citation in the added sentence to support your claim: "This scenario is encountered in biotechnology: .....".

---

## [Referee Report · Reviewer #3 (Public review)]

The authors address the process of community evolution under collective-level selection for a prescribed community composition. They mostly consider communities composed of two types that reproduce at different rates, and that can mutate one into the other. Due to such difference in 'fitness' and to the absence of density dependence, within-collective selection is expected to always favour the fastest grower, but collective-level selection can oppose this tendency, to a certain extent at least. By approximating the stochastic within-generation dynamics and solving it analytically, the authors show that not only high frequencies of fast growers can be reproducibly achieved, aligned with their fitness advantage. Small target frequencies can also be maintained, provided that the initial proportion of fast growers is sufficiently small. In this regime, similar to the 'stochastic corrector' model, variation upon which selection acts is maintained by a combination of demographic stochasticity and of sampling at reproduction. These two regions of achievable target compositions are separated by a gap, encompassing intermediate frequencies that are only achievable when the bottleneck size is small enough or the number of communities is (disproportionately) large.

A similar conclusion, that stochastic fluctuations can maintain the system over evolutionary time far from the prevalence of the faster-growing type, is then confirmed by analyzing a three-species community, suggesting that the qualitative conclusions of this study are generalizable to more complex communities.

I expect that these results will be of broad interest to the community of researchers who strive to improve community-level selection but are often limited to numerical explorations, with prohibitive costs for a full characterization of the parameter space of such embedded populations. The realization that not all target collective functions can be as easily achieved and that they should be adapted to the initial conditions and the selection protocol is also a sobering message for designing concrete applications.

A major strength of this work is that the qualitative behaviour of the system is captured by an analytically solvable approximation so that the extent of the 'forbidden region' can be directly and generically related to the parameters of the selection protocol.

The phenomenon the authors characterize is ecological in nature, though it is maintained even when switching between types is possible. Calling this dynamics community evolution reflects a widespread ambiguity in the field, not ascribable just to this work.

Although different types compete for being represented in the next generation's propagules, within-generation ecology is here representative of exponential growth. As species interactions are commonly manifest in lab serial dilution experiments, it would be interesting if future work explores the extent of the robustness of these results to density-dependent demography.

---

## [Author Response]

The following is the authors’ response to the original reviews.

Common comments

(1) Significance of zero mutation rate

Reviewers asked why we included mutation rate even though setting mutation rate to zero doesn’t change results. We think that including non-zero mutation rate makes our results more generalisable, and thus is a strength rather than weakness. To better motivate this choice, we have added a sentence to the beginning of Results:

(2) Writing the mu=0 case first

Reviewers suggested that we should first focus on the mu=0 case, and then generalize the result. The suggestions are certainly good. However, given the large amount of work involved in a re-organization, we have decided to adhere to our current narrative. However, we now only include equations where mu=0 in the main text, and have moved the case of nonzero mutation rate to Supplementary Information.

(3) Making equations more accessible

We have taken three steps to make equations more readable.

Equations in the main text correspond to the case of zero-mutation rate.The original section on equation derivation is now in a box in the main text so that readers have the choice of skipping it but interested readers can still get a gist of where equations came from.We have provided a much more detailed interpretation of the equation.

(4) Validity of the Gaussian approximation

Reviewers raised concerns about the validity of Gaussian approximation on F frequency 𝑓(𝜏) . The fact that our calculations closely match simulations suggest thatthis approximation is reasonable. Still, we added a discussion frequency about the validity of this approximation in Box 1.

We also added to SI with various cases of initial S and F sizes. This figure not normal. However, if initial S and F are both on the order of hundreds,𝑓(𝜏) then shows that when either initial S or initial F is small, the distribution of is the distribution of 𝑓(𝜏) is approximately Gaussian.

**Public Reviews:**
Summary:The authors demonstrate with a simple stochastic model that the initial composition of the community is important in achieving a target frequency during the artificial selection of a community.Strengths:To my knowledge, the intra-collective selection during artificial selection has not been seriously theoretically considered. However, in many cases, the species dynamics during the incubation of each selection cycle are important and relevant to the outcome of the artificial selection experiment. Stochasticity from birth and death (demographic stochasticity) plays a big role in these species' abundance dynamics. This work uses a simple framework to tackle this idea meticulously.This work may or may not be hysteresis (path dependency). If this is true, maybe it would be nice to have a discussion paragraph talking about how this may be the case. Then, this work would even attract the interest of people studying dynamic systems.

We have added this clarification in the main text:

“Note that here, selection outcome is path-dependent in the sense of being sensitive to initial conditions. This phenomenon is distinct from hysteresis where path-dependence results from whether a tuning parameter is increased or decreased.”

Weaknesses:(1) Connecting structure and functionIn typical artificial selection literature, most of them select the community based on collective function. Here in this paper, the authors are selecting a target composition. Although there is a schematic cartoon illustrating the relationship between collective function (y-axis) and the community composition in the main Figure 1, there is no explicit explanation or justification of what may be the origin of this relationship. I think giving the readers a naïve idea about how this structure-function relationship arises in the introduction section would help. This is because the conclusion of this paper is that the intra-collective selection makes it hard to artificially select a community that has an intermediate frequency of f (or s). If there is really evidence or theoretical derivation from this framework that indeed the highest function comes from the intermediate frequency of f, then the impact of this paper would increase because the conclusions of this stochastic model could allude to the reasons for the prevalent failures of artificial selection in literature.

We have added this to introduction:

“This is a common quest: whenever a collective function depends on both populations, collective function is maximised, by definition, at an intermediate frequency (e.g. too little of either population will hamper function [23]).”

(2) Explain intra-collective and inter-collective selection better for readers.The abstract, the introduction, and the result section use these terms or intra-collective and inter-collective selection without much explanation. For the wide readership of eLife, a clear definition in the beginning would help the audience grasp the importance of this paper, because these concepts are at the core of this work.

This is a great point. We have added in Abstract:

“Such collective selection is dictated by two opposing forces: during collective maturation, intra-collective selection acts like a waterfall, relentlessly driving the S-frequency to lower values, while during collective reproduction, inter-collective selection resembles a rafter striving to reach the target frequency. Due to this model structure, maintaining a target frequency requires the continued action of inter-collective selection.”

and in Introduction:

“A selection cycle consists of three stages (Fig. 1). During collective maturation, intra-collective selection favors fast-growing individuals within a collective. At the end of maturation, inter-collective selection acts on collectives and favors those achieving the target composition. Finally during collective reproduction, offspring collectives sample stochastically from the parents, a process dominated by genetic drift.”

(3) Achievable target frequency strongly depending on the degree of demographic stochasticity.I would expect that the experimentalists would find these results interesting and would want to consider these results during their artificial selection experiments. The main Figure 4 indicates that the Newborn size N0 is a very important factor to consider during the artificial selection experiment. This would be equivalent to how much bottleneck is imposed on the artificial selection process in every iteration step (i.e., the ratio of serial dilution experiment). However, with a low population size, all target frequencies can be achieved, and therefore in these regimes, the initial frequency now does not matter much. It would be great for the authors to provide what the N0 parameter actually means during the artificial selection experiments. Maybe relative to some other parameter in the model. I know this could be very hard. But without this, the main result of this paper (initial frequency matters) cannot be taken advantage of by the experimentalists.

We have added an analytical approximation for \begin{document}$\breve{N} _0$\end{document}, the Newborn size below which all target frequencies can be achieved in SI.

Also, we have added lines indicating \begin{document}$\breve{N} _0$\end{document} in Figure 4a.

(4) Consideration of environmental stochasticity.The success (gold area of Figure 2d) in this framework mainly depends on the size of the demographic stochasticity (birth-only model) during the intra-collective selection. However, during experiments, a lot of environmental stochasticity appears to be occurring during artificial selection. This may be out of the scope of this study. But it would definitely be exciting to see how much environmental stochasticity relative to the demographic stochasticity (variation in the Gaussian distribution of F and S) matters in succeeding in achieving the target composition from artificial selection.

You are correct that our work considers only demographic stochasticity. Indeed, considering other types of stochasticity will be an exciting future research direction. We added in the main text:

“Overall our model considers mutational stochasticity, as well as demographic stochasticity in terms of stochastic birth and stochastic sampling of a parent collective by offspring collectives. Other types of stochasticity, such as environmental stochasticity and measurement noise, are not considered and require future research.”

(5) Assumption about mutation ratesIf setting the mutation rates to zero does not change the result of the simulations and the conclusion, what is the purpose of having the mutation rates \mu? Also, is the unidirectional (S -> F -> FF) mutation realistic? I didn't quite understand how the mutations could fit into the story of this paper.

This is a great point. We have added this to the beginning of Results to better motivate our study:

“We will start with a complete model where S mutates to F at a nonzero mutation rate µ. We made this choice because it is more challenging to attain or maintain the target frequency when the abundance of fast-growing *F* is further increased via mutations. This scenario is encountered in biotechnology: an engineered pathway will slow down growth, and breaking the pathway (and thus faster growth) is much easier than the other way around. When the mutation rate is set to zero, the same model can be used to capture collectives of two species with different growth rates.

See answer on common question 1.

(6) Minor pointsIn Figure 3b, it is not clear to me how the frequency difference for the Intra-collective and the Inter-collective selection is computed.

We added a description in caption 3b.

In Figure 5b, the gold region (success) near the FF is not visible. Maybe increase the size of the figure or have an inset for zoom-in. Why is the region not as big as the bottom gold region?

We increased the resolution of Figure 5b so that the gold region near FF is more visible.

We have added Figure 5c and the following explanation to the main text:

“From numerical simulations, we identified two accessible regions: a small region near FF and a band region spanning from S to F (gold in Fig. 5b i). Intuitively, the rate at which FF grows faster than S+F is greater than the rate at which F grows faster than S (see section VIII in Supplementary Information). Thus, the problem can initially be reduced to a two-population problem (i.e. FF versus F+S; Fig. 5c left), and then expanded to a three-population problem (Fig. 5c right).”

**Recommendations For The Authors**
Since the conclusion of the model greatly depends on the noise (variation) of F and S in the Gaussian distribution, it would be nice to have a plot where the y-axis is the variation in terms of frequency and the x-axis is the s_0 or f_0 (frequency). In the plot, I would love to see how the variation in the frequency depends on the initial frequency of S and F. Maybe this is just trivial.

In the SI, we added Fig6a, as per your request. Previous Figure 6 became Figure 6b.

**Reviewer #2 (Public review):**
The authors provide an analytical framework to model the artificial selection of the composition of communities composed of strains growing at different rates. Their approach takes into account the competition between the targeted selection at the level of the meta-community and the selection that automatically favors fast-growing cells within each replicate community. Their main finding is a tipping point or path-dependence effect, whereby compositions dominated by slow-growing types can only be reached by community-level selection if the community does not start and never crosses into a range of compositions dominated by fast growers during the dynamics.These results seem to us both technically correct and interesting. We commend the authors on their efforts to make their work reproducible even when it comes to calculations via extensive appendices, though perhaps a table of contents and a short description of these appendices at the start of SI would help navigate them.

Thank you for the suggestion. We have added a paragraph at the beginning of SI.

The main limitation in the current form of the article is that it could clarify how its assumptions and findings differ from and improve upon the rest of the literature:- Many studies discuss the interplay between community-level evolution and species- or strain-level evolution. But "evolution" can be a mix of various forces, including selection, drift/randomness, and mutation/innovation.- This work's specificity is that it focuses strictly on constant community-level selection versus constant strain-level selection, all other forces being negligible (neither stochasticity nor innovation/mutation matter at either level, as we try to clarify now).

Note that intra-collective selection is not strictly “constant” in the sense that selection favoring F is the strongest at intermediate F frequency (Figure 3). However, we think that you mean that intra- and inter-collective selection are present in every cycle, and this is correct for our case, and for community selection in general.

- Regarding constant community-level selection, it is only briefly noted that "once a target frequency is achieved, inter-collective selection is always required to maintain that frequency due to the fitness difference between the two types" [pg. 3 {section sign}2]. In other words, action from the selector is required indefinitely to maintain the community in the desired state. This assumption is found in a fraction of the literature, but is still worth clarifying from the start as it can inform the practical applicability of the results.

This is a good point. We have added to abstract:

“Such collective selection is dictated by two opposing forces: during collective maturation, intra-collective selection acts like a waterfall, relentlessly driving the S-frequency to lower values, while during collective reproduction, inter-collective selection resembles a rafter striving to reach the target frequency. Due to this model structure, maintaining a target frequency requires the continued action of inter-collective selection.”

- More importantly, strain-level evolution also boils down here to pure selection with a constant target, which is less usual in the relevant literature. Here, (1) drift from limited population sizes is very small, with no meaningful counterbalancing of selection, (2) pure exponential regime with constant fitness, no interactions, no density- or frequency-dependence, (3) there is no innovation in the sense that available types are unchanging through time (no evolution of traits such as growth rate or interactions) and (4) all the results presented seem unchanged when mutation rate mu = 0 (as noted in Appendix III), meaning that the conclusions are not "about" mutation in any meaningful way.

With regard to point (1), Figure 4a (reproduced below) shows how Newborn size affects the region of achievable targets. Indeed at large Newborn size (e.g. 5000 and above), no target frequency is achievable (since drift is too small to generate sufficient inter-community variation and consequently all communities are dominated by fast-growing F). However at Newborn size of for example 1000, there are two regions of accessible target frequencies. At smaller Newborn size, all target frequencies become achievable due to drift becoming sufficiently strong.

With regard to points (2) and (3), we have added to Introduction

“To enable the derivation of an analytical expression, we have made the following simplifications.

First, growth is always exponential, without complications such as resource limitation, ecological interactions between the two populations, or density-dependent growth. Thus, the exponential growth equation can be used. Second, we consider only two populations (genotypes or species): the fast-growing *F* population with size F and the slow-growing S population with size S. We do not consider a spectrum of mutants or species, since with more than two populations, an analytical solution becomes very difficult.”

With regard to point (4), we view this as a strength rather than weakness. We have added the following to the beginning of Results and Discussions:

“We will start with a complete model where S mutates to F at a nonzero mutation rate µ. We made this choice because it is more challenging to attain or maintain the target frequency when the abundance of fast-growing F is further increased via mutations.”

“When the mutation rate is set to zero, the same model can be used to capture collectives of two species with different growth rates.”

See Point 1 of Common comments.

- Furthermore, the choice of mutation mechanism is peculiar, as it happens only from slow to fast grower: more commonly, one assumes random non-directional mutations, rather than purely directional ones from less fit to fitter (which is more of a "Lamarckian" idea). Given that mutation does not seem to matter here, this choice might create unnecessary opposition from some readers or could be considered as just one possibility among others.

We have added the following justification:

“This scenario is encountered in biotechnology: an engineered pathway will slow down growth, and breaking the pathway (and thus faster growth) is much easier than the other way around.”

It would be helpful to have all these points stated clearly so that it becomes easy to see where this article stands in an abundant literature and contributes to our understanding of multi-level evolution, and why it may have different conclusions or focus than others tackling very similar questions.Finally, a microbial context is given to the study, but the assumptions and results are in no way truly tied to that context, so it should be clear that this is just for flavor.

We have deleted “microbial” from the title, and revised our abstract:

**Recommendations For The Authors**
(1) More details concerning our main remark above:- The paragraph discussing refs [24, 33] is not very clear in how they most importantly differ from this study. Our impression is that the resource aspect is not very important for instance, and the main difference is that these other works assume that strains can change in their traits.

We are fairly sure that resource depletion is important in Rainey group’s study, as the attractor only evolved after both strains grew fast enough to deplete resources by the end of maturation. Indeed, evolution occurred in interaction coefficients which dictate the competition between strains for resources.

Regardless, you raised an excellent point. As discussed earlier, we have added the following:

“To enable the derivation of an analytical expression, we have made the following simplifications.

First, growth is always exponential, without complications such as resource limitation, ecological interactions between the two populations, or density-dependent growth. Thus, the exponential growth equation can be used. Second, we consider only two populations (genotypes or species): the fast-growing *F* population with size F and the slow-growing S population with size S. We do not consider a spectrum of mutants or species, since with more than two populations, an analytical solution becomes very difficult.”

- We would advise the main text to focus on mu = 0, and only say in discussion that results can be generalized.

Your suggestion is certainly good. However, given the large amount of work involved in a reorganisation, we have decided to adhere to our current narrative. However, as discussed earlier, we have added this at the beginning of Results to help orient readers:

“We will start with a complete model where S mutates to F at a nonzero mutation rate µ. We made this choice because it is more challenging to attain or maintain the target frequency when the abundance of fast-growing F is further increased via mutations.”

“When the mutation rate is set to zero, the same model can be used to capture collectives of two species with different growth rates.”

(2) We think the material on pg. 5 "Intra-collective evolution is the fastest at intermediate F frequencies, creating the "waterfall" phenomenon", although interesting, could be presented in a different way. The mathematical details on how to find the probability distribution of the maximum of independent random variables (including Equation 1) will probably be skipped by most of the readers (for experienced theoreticians, it is standard content; for experimentalists, it is not the most relevant), as such I would recommend displacing them to SM and report only the important results.

This is an excellent suggestion. We have put a sketch of our calculations in a box in the main text to help orient interested readers. As before, details are in SI.

Similarly, Equations 2, 3, and 4 are hard to read given the large amount of parameters and the low amount of simplification. Although exploring the effect of the different parameters through Figures 3 and 4 is useful, I think the role of the equations should be reconsidered:i. Is it possible to rewrite them in terms of effective variables in a more concise way?

See Point 3 of Common comments.

ii. Is it possible to present extreme/particular cases in which they are easier to interpret?

We have focused on the case where the mutation rate is zero. This makes the mathematical expressions much simpler (see above).

(3) Is it possible to explain more in detail why the distribution of f_k+1 conditional to f_k^* is well approximated by a Gaussian? Also, have you explored to what extent the results would change if this were not true (in light of the few universal classes for the maximum of independent variables)?

Despite the appeal to the CLT and the histograms in the Appendix suggesting that the distribution looks a bit like a Gaussian at a certain scale, fluctuations on that scale are not necessarily what is relevant for the results - a rapid (and maybe wrong) attempt at a characteristic function calculation suggests that in your case, one does not obtain convergence to Gaussians unless we renormalize by S(t=0) and F(t=0), so it seems there is a justification missing in the text as is for the validity of this approximation (or that it is simply assumed).

See point 4 of Common comments.

**Reviewer #3 (Public Reviews):**
The authors address the process of community evolution under collective-level selection for a prescribed community composition. They mostly consider communities composed of two types that reproduce at different rates, and that can mutate one into the other. Due to such differences in 'fitness' and to the absence of density dependence, within-collective selection is expected to always favour the fastest grower, but the collective-level selection can oppose this tendency, to a certain extent at least. By approximating the stochastic within-generation dynamics and solving it analytically, the authors show that not only high frequencies of fast growers can be reproducibly achieved, aligned with their fitness advantage. Small target frequencies can also be maintained, provided that the initial proportion of fast growers is sufficiently small. In this regime, similar to the 'stochastic corrector' model, variation upon which selection acts is maintained by a combination of demographic stochasticity and of sampling at reproduction. These two regions of achievable target compositions are separated by a gap, encompassing intermediate frequencies that are only achievable when the bottleneck size is small enough or the number of communities is (disproportionately) larger.A similar conclusion, that stochastic fluctuations can maintain the system over evolutionary time far from the prevalence of the faster-growing type, is then confirmed by analyzing a three-species community, suggesting that the qualitative conclusions of this study are generalizable to more complex communities.I expect that these results will be of broad interest to the community of researchers who strive to improve community-level selection, but are often limited to numerical explorations, with prohibitive costs for a full characterization of the parameter space of such embedded populations. The realization that not all target collective functions can be as easily achieved and that they should be adapted to the initial conditions and the selection protocol is also a sobering message for designing concrete applications.A major strength of this work is that the qualitative behaviour of the system is captured by an analytically solvable approximation so that the extent of the 'forbidden region' can be directly and generically related to the parameters of the selection protocol.

Thanks so much for these positive comments.

I however found the description of the results too succinct and I think that more could be done to unpack the mathematical results in a way that is understandable to a broader audience. Moreover, the phenomenon the authors characterize is of purely ecological nature. Here, mutations of the growth rate are, in my understanding, neither necessary (non-trivial equilibria can be maintained also when \mu = 0) nor sufficient (community-level selection is necessary to keep the system far from the absorbing state) for the phenomenon described. Calling this dynamics community evolution reflects a widespread ambiguity, and is not ascribable just to this work. I find that here the authors have the opportunity to make their message clearer by focusing on the case where the 'mutation' rate \mu vanishes (Equations 39 & 40 of the SI) - which is more easily interpretable, at least in some limits - while they may leave the more general equations 3 & 4 in the SI.

See points 1-4 of Common comments.

Combined with an analysis of the deterministic equations, that capture the possibility of maintaining high frequencies of fast growers, the authors could elucidate the dynamics that are induced by the presence of a second level of selection, and speculate on what would be the result of real open-ended evolution (not encompassed by the simple 'switch mutations' generally considered in evolutionary game theory), for instance discussing the invasibility (or not) of mutant types with slightly different growth rates.

Indeed, evolution is not restricted to two types. However, our main goal here is to derive an analytical expression, and it was difficult for even two types. For three-type collectives, we had to resort to simulations. Investigating the case where fitness effects of mutations are continuously distributed is beyond the scope of this study.

The single most important model hypothesis that I would have liked to be discussed further is that the two types do not interact. Species interactions are not only essential to achieve inheritance of composition in the course of evolution but are generally expected to play a key role even on ecological time scales. I hope the authors plan to look at this in future work.

In our system, the S and F do interact in a competitive fashion: even though S and F are not competing for nutrients (which are always in excess), they are competing for space. This is because a fixed number of cells are transferred to the next cycle. Thus, the presence of F will for example reduce the chance of S being propagated. We have added this clarification to our main text:

“Note that even though S and F do not compete for nutrients, they compete for space: because the total number of cells transferred to the next cycle is fixed, an overabundance of one population will reduce the likelihood of the other being propagated.”

**Recommendations For The Authors**
I felt the authors could put some additional effort into making their theoretical results meaningful for a population of readers who, though not as highly mathematically educated as they are, can nonetheless appreciate the implications of simple relations or scaling. Below, you find some suggestions:(1) In order to make it clear that there is a 'natural' high-frequency equilibrium that can be reached even in the absence of selection, the authors could examine first the dynamics of the deterministic system in the absence of mutations, and use its equilibria to elucidate the combined role of the 'fitness' difference \omega and of the generation duration \tau in setting its value. The fact that these parameters always occur in combination (when there are no mutations) is a general and notable feature of the stochastic model as well. Moreover, this model would justify why you only focus on decreasing the frequency in the new generation.

Note that the ‘natural’ high-frequency equilibrium in the absence of collective selection is when fast grower F becomes fixed in the population. Following your suggestion, we have introduced two parameters 𝑅τ and 𝑊τ to reflect the coupling between ‘fitness’ and ‘generation duration’:

(2) Since the phenomenon described in the paper is essentially ecological in nature (as the author states, it does not change significantly if the 'mutation rate' \mu is set to zero), I would put in the main text Equations 39 & 40 of the SI in order to improve intelligibility.

See Point 2 at the beginning of this letter.

These equations can be discussed in some detail, especially in the limit of small f^*_k, where I think it is worth discussing the different dependence of the mean and the variance of the frequency distribution on the system's parameters.

This is a great suggestion. We have added the following:

“In the limit of small \begin{document}$f_{k}^{*}$\end{document}, Equation (3) becomes f \begin{document}$\left.\overline{f}(\tau)\right|_{f} 1 \approx f_{k}^{*} W_{\tau}$\end{document} while Equation (4) becomes *N0*) and fold-change in F/S during maturation (\begin{document}$\left.\sigma_{f}^{2}(\tau)\right|_{\mathscr{L}} \psi_{1}=\left(2-\frac{1}{R W}\right) f_{k}^{*} W_{\tau}^{2} / N_{0}$\end{document}. Thus, both Newborn size (*Wτ*) are important determinants of selection progress.

(3) I would have appreciated an explanation in words of what are the main conceptual steps involved in attaining Equation 2, the underlying hypotheses (notably on community size and distributions), and the expected limits of validity of the approximation.

See points 3 and 4 at the beginning of this letter.

(4) I think that some care needs to be put into explaining where extreme value statistics is used, and why is the median of the conditional distribution the most appropriate statistics to look at for characterizing the evolutionary trajectory (which seems to me mostly reliant on extreme values).

Great point! We added an explanation of using median value in Box 1.

and also added figure 7 to explaining it in SI.

Showing in a figure the different distributions you are considering (for instance, plotting the conditional distribution for one generation in the trajectories displayed in Figure 2) would be useful to understand what information \bar f provides on a sequence of collective generations, where in principle there may be memory effects.

Thanks for this suggestion. We have added to Fig 2d panel to illustrate the shape and position of F frequency distributions in each step in the first two selection cycles.

(5) Similarly, I do not understand why selecting the 5% best communities should push the system's evolution towards the high-frequency solution, instead of just slowing down the improvement (unless you are considering the average composition of the top best communities - which should be justified). I think that such sensitivity to the selection intensity should be appropriately referenced and discussed in the main text, as it is a parameter that experimenters are naturally led to manipulate.

In the main text, we have added this explanation:

“In contrast with findings from an earlier study [23], choosing top 1 is more effective than the less stringent “choosing top 5%”. In the earlier study, variation in the collective trait is partly due to nonheritable factors such as random fluctuations in Newborn biomass. In that context, a less stringent selection criterion proved more effective, as it helped retain collectives with favorable genotypes that might have exhibited suboptimal collective traits due to unfavorable nonheritable factors. However, since this study excludes nonheritable variations in collective traits, selecting the top 1 collective is more effective than selecting the top 5% (see Fig. 11 in Supplementary Information).”

(6) Equation 1 could be explained in simpler terms as the product between the probability that one collective reaches the transmitted value times the probability that all others do worse than that. The current formulation is unclear, perhaps just a matter of English formulation.

We have revised our description to state:\begin{document}$$\displaystyle \Psi\left(f_{k+1}^{*}=f \mid f_{k}^{*}\right)=g P_{f_{k+1, \tau}}\left(f \mid f_{k}\right)\left[1-\int_{0}^{f} d f^{\prime} P_{f_{k+1, \tau}}\left(f^{\prime} \mid f_{k}^{*}\right)\right]^{g-1}$$\end{document}

“Equation (1) can be described as the product between two terms related to probability: (i) \begin{document}$g P_{f_{k+1,7}}\left(f \mid f_{k}^{*}\right)$\end{document} describes the probability density that any one of the *g* Adult collectives achieves *f* given \begin{document}$f_{k}^{*}$\end{document}, and (ii) \begin{document}$\left[1-\int_{0}^{f} d f^{\prime} P_{f_{k+1,7}}\left(f^{\prime} \mid f_{k}\right)\right]^{g-1}$\end{document} describes the probability that all other g – 1 collectives achieve frequencies above *f* and thus not selected.”

(7) I think that the discussion of the dependence of the boundaries of the 'waterfall' region with the difference in growth rate \omega is important and missing, especially if one wants to consider open-ended evolution of the growth rate - which can occur at steps of different magnitude.

We added a new chapter and figure in supplementary information on the threshold values when \omega varies. As expected, smaller \omega enlarges the success area.

We have also added a new figure panel to show how maturation time affects selection efficacy.

(8) Notations are a bit confusing and could be improved. First of all, in most equations in the main text and SI, what is initially introduced as \omega appears as s. This is confusing because the letter s is also used for the frequency of the slow type.The letter S is used to denote an attribute of cells (S cells), the type of cells (Equations 1-3 of the SI) and the number of these cells in the population, sometimes with different meanings in the same sentence. This is confusing, and I suggest referring to slow cells or fast cells instead (or at least to S-cells and F-cells), and keeping S and F as variables for the number of cells of the two types.

All typos related to the notation have been fixed. We use S and F as types, and *S* and *F* (italic) and population numbers.

(9) On page 3, when introducing the sampling of newborns as ruled by a binomial distribution, the information that you are just transmitting one collective is needed, while it is conveyed later.

We have added this emphasis:

“At the end of a cycle, a single Adult with the highest function (with F frequency *f* closest to the target frequency \begin{document}$\hat{f}$\end{document}) is chosen to reproduce *g* Newborn collectives each with *N0* cells (‘Selection’ and ’Reproduction’ in Fig. 1).”

(10) I found that the abstract talks too early about the 'waterfall' phenomenon. As this is a concept introduced here, I suggest the authors first explain what it is, then use the term. It is a useful metaphor, but it should not obscure the more formal achievements of the paper.

We feel that the “waterfall” analogy offers a gentle helping hand to orient those who have not thought much about the phenomenon. We view abstract as an opportunity to attract readership, and thus the more accessible the better.

(11) In the SI there are numerous typos and English language issues. I suggest the authors read carefully through it, and add line numbers to the next version so that more detailed feedback is possible.

Thank you for going through SI. We have gone through the SI, and fixed problems.